# Heart-nosed bat alphacoronaviruses use human CEACAM6 to enter cells

Giulia Gallo[1,2], Antonello Di Nardo[1], Doreen Lugano[3], Adam J. Roberts[4], Bernadette Ataku Kutima[3], Moses Okombo[5], Aghnianditya Kresno Dewantari[1,6], Florence M. M. Buckley[2], Gavin J. Wright[4], James Nyagwange[3], Bernard Agwanda[7], Stephen C. Graham[2✉] & Dalan Bailey[1✉]

Identifying viruses with zoonotic potential on the basis of their ability to enter human cells is a critical component of pandemic prediction, prevention and preparedness. Here using a computational approach that retains maximum phylogenetic diversity, we selected an optimal subset of alphacoronavirus spike proteins to screen against broad coronavirus receptor libraries. Most of the selected spike proteins did not use any of the established coronavirus receptors. However, the pseudotyped spike protein of *Cardioderma cor* (heart-nosed bat) coronavirus KY43 (CcCoV-KY43) could enter human cells. Using a recombinant CcCoV receptor-binding domain (RBD) and a human receptor screening platform, we identified direct interactions with the human CEACAM proteins CEACAM3, CEACAM5 and CEACAM6. Overexpression of human CEACAM6—a protein widely expressed in the human lung—conferred permissivity to otherwise refractory human cells. A crystal structure showed that the RBD binds the amino-terminal IgV-like domain of human CEACAM6. Immune surveillance studies using sera of individuals from the Taveta region of Kenya, where CcCoV-KY43 was identified, did not show significant evidence of recent spillover. Wider characterization of alphacoronaviruses related to CcCoV-KY43 showed that human CEACAM6 is used by two other CcCoVs collected in Kenya. Moreover, there was more restricted nonhuman CEACAM6 tropism for viruses isolated from *Rhinolophus* bats from Russia and China. Thus, alphacoronaviruses that use CEACAM6 are probably geographically widespread, and viruses from East Africa show potential for transmission to humans.

Following the COVID-19 pandemic, there has been a renewed focus on the characterization and prediction of viruses with zoonotic potential, particularly among the coronaviruses. Between 60% and 75% of human pathogens are thought to have a zoonotic origin[1]. However, acquiring comprehensive information on the zoonotic potential of all viruses is technically challenging because of their extensive diversity. Detailed data are usually available for only a few species of human viruses, or their close relatives, which belies the fact that the next pandemic is likely to be a previously undescribed virus, as SARS-CoV-2 was to virologists in 2019.

The first barrier for any cross-species viral jump is cell entry, a process that relies on the binding of viral attachment proteins to cellular receptors. To attempt wider characterization and identification of viruses with zoonotic potential, we focused on alphacoronavirus entry as a model, owing to the lack of detailed genus-wide information on the host range of these predominantly bat-borne viruses. So far, only two cellular receptors have been characterized for alphacoronaviruses. Human, porcine, canine and feline aminopeptidase N (APN) receptors provide entry to group I viruses, which include human coronavirus 229E (HCoV-229E)[2,3], transmissible gastroenteritis virus and the related porcine respiratory coronavirus (PRCV)[4], as well as some canine and feline coronaviruses[5]. Human coronavirus NL63 (HCoV-NL63) is the only known alphacoronavirus to use ACE2 as a receptor[6]. Apart from ACE2 (used by SARS-CoV-1 and SARS-CoV-2), human betacoronaviruses use the receptors DPP4 (used by MERS-CoV)[7], O-acetylated sialic acids (used by HKU1 (ref. 8) and OC43 (ref. 9)) and TMPRSS2 (used by HKU1)[10]. Recently, porcine and human DPEP1 were both shown to be efficient receptors for porcine haemagglutinating encephalomyelitis virus, although there is no evidence for zoonotic infection[11]. Entry of the betacoronavirus mouse hepatitis virus (MHV) is facilitated by mouse CEACAM1 (ref. 12); although in this case, the human orthologue is not efficiently used[13]. How receptor usage partitions across the entire alphacoronavirus genus and whether there are other human tropic viruses and/or other permissive receptors are unclear.

## APN- and ACE2-dependent entry is rare for alphacoronaviruses

For all coronaviruses, the spike protein is the major determinant of viral entry, and cleavage by cellular proteases produces two subunits: S1 and S2. S1 is involved in binding to the cellular receptor, whereas

[1]The Pirbright Institute, Woking, UK. [2]Department of Pathology, University of Cambridge, Cambridge, UK. [3]KEMRI–Wellcome Trust Research Programme, Kilifi, Kenya. [4]Hull York Medical School, Department of Biology, York Biomedical Research Institute, University of York, York, UK. [5]Loisaba Conservancy, Nanyuki, Kenya. [6]Department of Infectious Disease, Faculty of Medicine, Imperial College London, London, UK. [7]Department of Zoology, National Museum of Kenya, Nairobi, Kenya. ✉e-mail: scg34@cam.ac.uk; dalan.bailey@pirbright.ac.uk

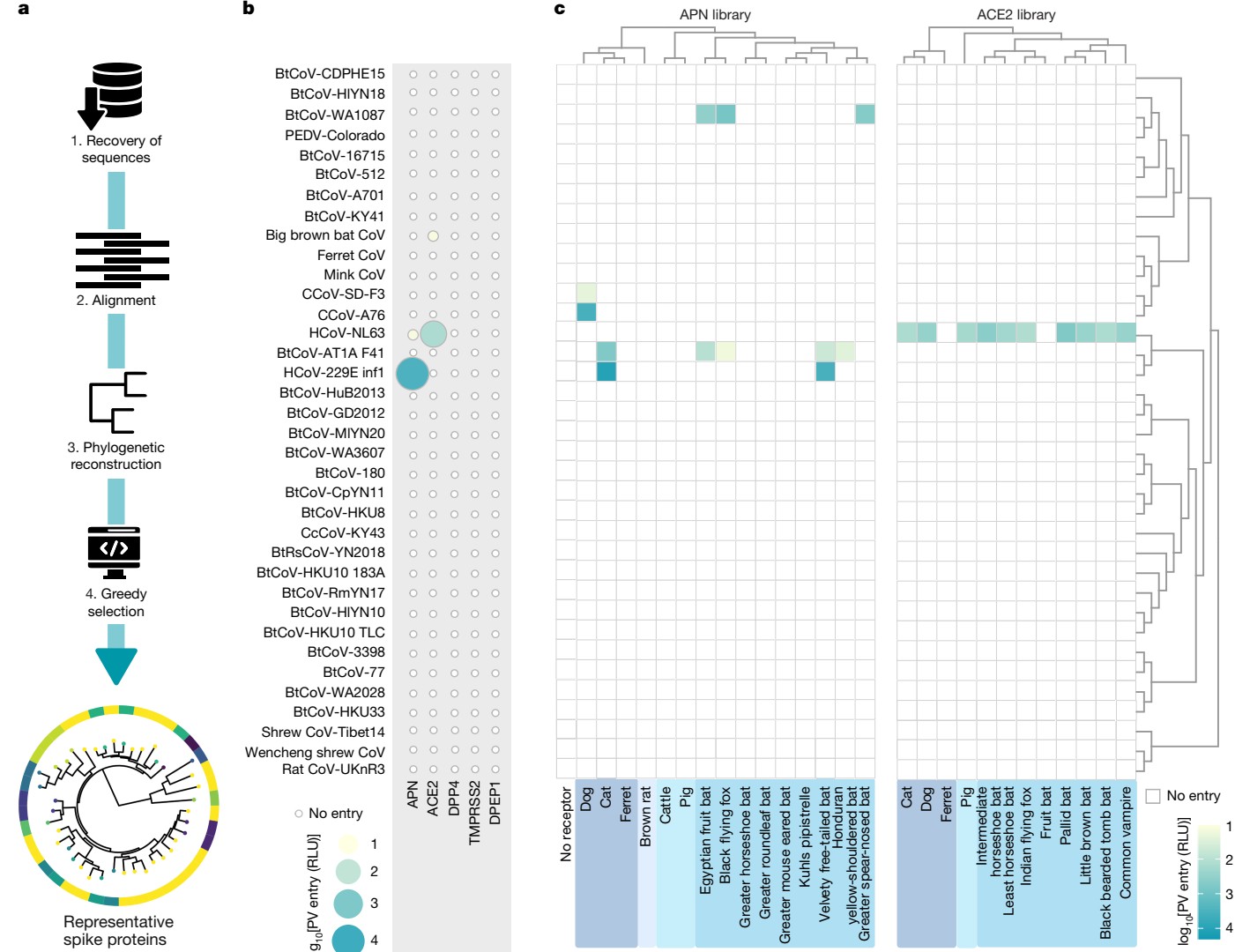

**Fig. 1 | Construction of the alphacoronavirus spike protein library and screening of the receptor libraries. a**, Schematic of the selection process used to create the spike protein library. Full-length amino acid sequences of alphacoronaviruses were recovered from public repositories (1) and aligned (2) to infer the alphacoronavirus genus phylogeny (3), from which cophenetic distances between all taxa were estimated. In total, 40 representatives spike protein sequences were selected using an optimal greedy-based algorithm (4). **b**, Screening of known human coronavirus receptors using the alphacoronavirus spike protein library pseudotypes. As expected, entry was observed for HCoV-229E inf1 and HCoV-NL63 with APN and ACE2, respectively. Entry of other

viruses in the library was not supported by previously described human coronavirus receptors. **c**, Wider receptor usage by the alphacoronavirus library. To assess whether APN or ACE2 proteins from nonhuman species could be used as receptors for spike proteins in the alphacoronavirus library, we examined receptors from different mammalian species. Positive results were reproduced at least three times using independent biological replicates. All experiments were performed in technical triplicate with pseudotype (PV) entry indicated as $\log_{10}$ relative light units (RLU). The average of the repeats, minus background, is shown.

S2 is involved in fusion of the viral and cellular membranes. To select alphacoronavirus spike proteins that accurately represent the known diversity, we used a greedy algorithm[14] on all full-length spike protein sequences ($n = 2,714$) retrieved from the Virus Pathogen Database and Analysis Resource database[15] and deposited into GenBank (as of May 2021). Our goal was to scale down the number of spike proteins tested without significantly losing the richness and heterogeneity of diversity in this genus. The final number chosen ($n = 40$) was based on our capacity for pseudotyping and receptor screening (Fig. 1a and Supplementary Table 1). Among these sequences, which included representatives from taxonomically classified subgenera such as *Colacovirus*, *Pedacovirus*, *Minacovirus*, *Tegacovirus*, *Nyctacovirus* and *Luchacovirus*, the overall level of amino acid conservation was low, especially in the predicted RBD (Supplementary Fig. 1). The selected 40-sequence spike protein panel (around 1.5% of the full dataset) captured 53.4% of the total phylogenetic

diversity, which substantially exceeded size-matched random panels (13.7 ± 3.2%, 10,000 permutations, around 3.9-fold enrichment, empirical $P < 0.0001$). Most selected spike proteins were from poorly characterized alphacoronaviruses isolated from bats (27 out of 40); however, the two endemic human viruses HCoV-229E and HCoV-NL63 were selected. To confirm that the commercially synthesized, cloned, codon-optimized spike protein open-reading frames (ORFs) could efficiently produce spike proteins that are pseudotyped onto HIV-1-based lentiviruses, pseudoviruses were purified and spike protein incorporation was confirmed by immunoblotting (Extended Data Fig. 1a). Concurrent to the establishment and validation of this alphacoronavirus spike protein pseudotype library, we developed equivalent plasmid-based APN and ACE2 expression libraries that represented 25 and 34 mammalian species, respectively (Supplementary Tables 2 and 3). Expression of these tagged receptors was verified by flow cytometry (Supplementary Fig. 2).

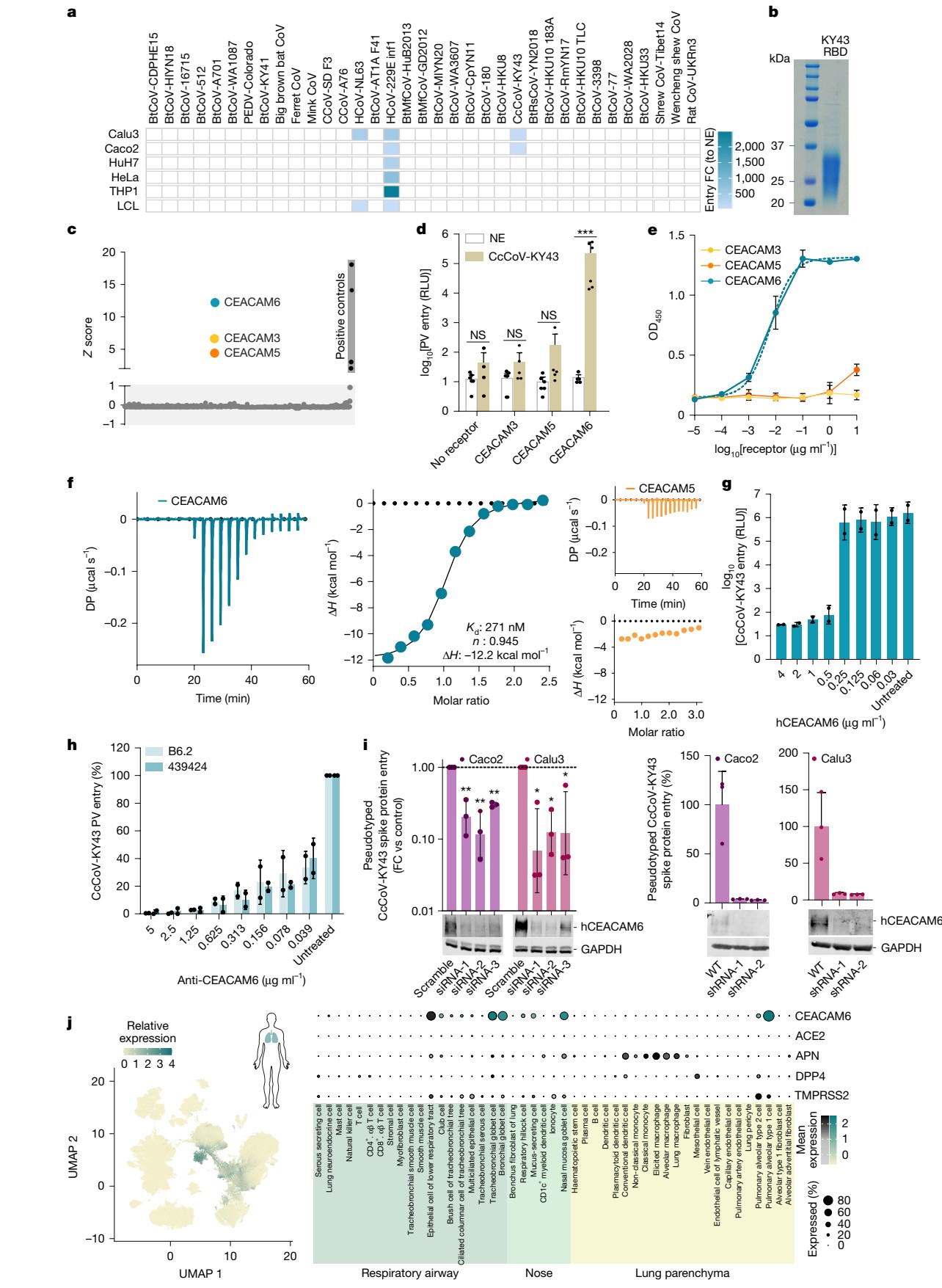

**Fig. 2** | See next page for caption.

**Fig. 2 | Human CEACAM6 supports entry of CcCoV-KY43. a**, Different human cell lines were assessed for their permissivity to alphacoronavirus pseudotyped spike proteins. CcCoV-KY43 could enter Calu3 (lung) and Caco2 (intestinal) cells. The mean entry (fold change (FC) compared with non-enveloped (NE) controls) from technical triplicates is shown. **b**, Recombinantly produced CcCoV-KY43 RBD shows potentially high levels of glycosylation. **c**, Screening an array of human receptor ectodomains identified CEACAM6, CEACAM3 and CEACAM5 as interactors of the CcCoV-KY43 RBD. **d**, CEACAM proteins were overexpressed in refractory HEK293T cells, and the assays showed that pseudotyped CcCoV-KY43 spike protein could only enter in the presence of CEACAM6. Two biological replicates (technical triplicates) along with their s.d. values, are shown. ***$P = 0.0003$ using two-way analysis of variance (ANOVA). NS, not significant. **e**, Interaction of the CcCoV-KY43 RBD and CEACAM6 was confirmed by ELISA, whereas no binding was observed with CEACAM3 or CEACAM5. The mean of two experiments (technical triplicate) is shown with s.d. values. **f**, ITC (showing differential power (DP) values) of CEACAM6 and CEACAM5 binding the CcCoV-KY3 RBD. Data are representative of two (CEACAM5) or three (CEACAM6) independent experiments. For CEACAM6, mean $K_d$, number of binding sites ($n$) and enthalpy change ($\Delta H$) are shown. **g**, Pseudotyped CcCoV-KY43 spike protein was incubated for 1 h with recombinant CEACAM6, followed by titration on HEK293T cells expressing

human CEACAM6 (hCEACAM6). Raw data of the mean of two independent experiments (technical triplicate) are shown with s.d. values. **h**, Monoclonal antibodies against CEACAM6 (B6.2 and clone 439424) were used to neutralize CcCoV-KY43 entry into HEK293T cells expressing hCEACAM6. The mean of two independent experiments (technical triplicate) was normalized to the untreated condition and plotted with s.d. values. **i**, Left, CEACAM6-specific siRNAs, or negative control, were electroporated into permissive cells and then infected with CcCoV-KY43. To confirm CEACAM6 reduction, cell lysates were analysed by immunoblotting (bottom). Mean entry reduction relative to the scrambled siRNA control from three independent experiments (technical triplicates) along with s.d. values, is shown. Significance of $\log_{10}$ fold change was determined using a one-sample $t$-test and $P$ values were adjusted for multiple comparisons (Caco2: $P = 0.004$ for siRNA-1, $P = 0.001$ for siRNA-2 and $P = 0.009$ for siRNA-3; Calu3: $P = 0.022$ for siRNA-1, $P = 0.04$ for siRNA-2 and siRNA-3). Right, using lentivirus expressing shRNA, stable knockdown of *CEACAM6* expression in Caco2 and Calu3 cell lines was induced and validated by immunoblotting. One biological replicate (technical triplicates) is shown, along with s.d. values. WT, wild type. **j**, Left, lung-specific single-cell transcriptomic data show the expression of CEACAM6. Right, comparison of the expression of coronavirus receptors in lung cells: CEACAM6, ACE2, APN, DDP4 and TMPRSS2 are shown with the dot plot indicating both average and per cent expression.

First, we assessed use of known human receptors (APN, ACE2, DPP4, TMPRSS2 and DPEP1), which identified only HCoV-229E and HCoV-NL63 as human-tropic (Fig. 1b). The algorithm-selected HCoV-229E spike protein was not functional; therefore, we replaced it with the reference sequence for the virus (Supplementary Fig. 3a). Next, we assayed whether the pseudotyped spike protein from our alphacoronavirus library could use any APN or ACE2 from our receptor libraries (Fig. 1c (selected data) and Supplementary Figs. 4a and 5a (all data)). Canine coronaviruses (CCoVs) could enter cells that express APN, whereas despite a feline coronavirus being included in the library, it did not pseudotype. Moreover, two different porcine epidemic diarrhoea virus (PEDV) spike proteins did not use porcine APN in our experimental settings, which could be explained by the dynamics of PEDV spike protein internalization in non-susceptible cells[16]. However, another porcine alphacoronavirus, PRCV, was able to use the same receptor (Supplementary Fig. 3b). Where possible, we included the species for which individual viruses were identified or the closest relative available (Supplementary Figs. 4b and 5b). This selection included genetically divergent *Chiroptera* species to account for the reservoir biology of coronaviruses and domesticated and peridomestic animals that might bridge the gap between unknown reservoirs and humans. In general, APN-using viruses seemed to be restricted to lower numbers of host species than the ACE2-using HCoV-NL63. We suggest that this difference is determined by the spike protein–receptor interaction surface area, affinity and physicochemical nature (for example, charge and hydrophobicity), combined with receptor sequence variability across hosts. Notably, two bat-origin alphacoronaviruses, BtCoV-WA1087 and BtCoV-AT1A-F41, used APN receptors to enter cells, albeit not human APN (Fig. 1c). To our knowledge, this is the first study to confirm APN usage by alphacoronaviruses isolated from bats. Most of the alphacoronaviruses tested did not use any of the expressed APN or ACE2 receptors to enter cells. Some coronaviruses are only infectious after protease treatment, which facilitates transition of the RBD from a closed to open conformation and/or exposure of the fusion peptide at the S2′ site. To address this possibility, we repeated experiments in the presence or absence of human serine protease TMPRSS2 or trypsin (Extended Data Fig. 2a (TMPRSS2) and 2b (trypsin)). Co-expression or pretreatment did not affect host range, although there was some evidence of digestion of spike proteins by trypsin (Extended Data Fig. 1b). To examine whether closed RBD conformations were leading to false negatives in our screens, we purified a selection of RBDs and performed flow cytometry to assess receptor binding at the cell surface (Extended Data Fig. 3a,b). The observed correlation between entry and receptor

binding indicated that an absence of pseudotype entry reflects a genuine inability to bind APN and ACE2. Together, these results suggest that use of the known receptors ACE2 and APN may be a relatively rare phenomenon in the alphacoronavirus genus.

## Human CEACAM6 supports entry of CcCoV-KY43

Recent high-throughput analyses of a diverse range of pseudotyped viruses (screened against the NCI-60 panel from the National Institutes of Health) concluded that coronavirus entry is a key determinant of host range[17]. As most alphacoronavirus spike proteins did not use human APN, ACE2, DPP4, TMPRSS2 or DPEP1 for entry, the next step in our zoonotic risk assessment was to examine pseudotype infection of a wider range of human cell lines. Across our screen (Fig. 2a), the only examples of infection we detected were with pseudotyped HCoV-NL63 and HCoV-229E, a finding consistent with our receptor screening results and their known human tropism (Fig. 1b), and with BtCoV-KY43 (Fig. 2a). The spike protein sequence of BtCoV-KY43 (hereafter referred to as CcCoV-KY43) was originally isolated from *C. cor* bats in Kenya in 2006 (ref. 18), with a partial genome sequence published in 2012 (GenBank accession: HQ728480)[19]. We speculated that successful infection of Calu3 and Caco2 cells by CcCoV-KY43 spike protein pseudotypes is facilitated by an unknown human receptor. To identify this receptor, we used an avidity-based method designed to systematically identify direct extracellular interactions with a library of 759 human receptor ectodomains[20], using the CcCoV-KY43 spike protein RBD (CcCoV-KY43 RBD) as prey (Fig. 2b). We identified three interactions, all of which were paralogues of the human carcinoembryonic antigen cell adhesion molecule (CEACAM) protein: CEACAM3, CEACAM5 and CEACAM6 (Fig. 2c and Supplementary Fig. 6). These cell surface glycoproteins contain immunoglobulin-like domains, with both CEACAM5 and CEACAM6 having a glycosylphosphatidylinositol anchor. Subsequent overexpression of a panel of full-length human CEACAM proteins in HEK293 cells, which are normally refractory to CcCoV-KY43 spike protein pseudotype infection, demonstrated that only human CEACAM6 expression resulted in a significant increase in permissivity (Fig. 2d). A split luciferase-based cell–cell fusion reporter assay subsequently confirmed this receptor usage profile in a separate cellular context (Extended Data Fig. 4). ELISA experiments with recombinant CcCoV-KY43 RBD and the extracellular domains of human CEACAM3, CEACAM5 and CEACAM6 enabled us to attribute this entry specificity to a higher affinity for CEACAM6 (Fig. 2e). Thermodynamic analyses using isothermal titration calorimetry (ITC) confirmed a 1:1 interaction between the CcCoV-KY43

RBD and CEACAM6, with a dissociation constant ($K_d$) of $271 \pm 68$ nM (mean ± s.d., $n = 3$), whereas the RBD did not measurably bind CEACAM5 (Fig. 2f and Supplementary Table 4). Dose-dependent neutralization of CcCoV-KY43 spike protein pseudotypes was demonstrated using soluble, recombinant CEACAM6 (Fig. 2g), and cell entry was blocked when using monoclonal antibodies against human CEACAM6 (Fig. 2h and Extended Data Fig. 5a). To further confirm the role of human CEACAM6 in CcCoV-KY43 entry, we used CEACAM6-specific short interfering RNAs (siRNAs) to knockdown endogenous protein expression in the susceptible cell lines Caco2 and Calu3. These experiments revealed reduced virus entry for CcCoV-KY43 (Fig. 2i, left) but not HCoV-229E inf1 (Supplementary Fig. 7). The same results were obtained in Calu3 and Caco2 cells treated with short hairpin RNA (shRNA) that were selected to stably downregulate the expression of human *CEACAM6* (Fig. 2i, right). To assess whether human CEACAM6 could support entry of other greedy-selected alphacoronaviruses, we re-screened our library but did not find other examples of entry (Extended Data Fig. 6).

In humans, pathogenic coronaviruses often have tropism for the respiratory and/or gastrointestinal tract. Analyses of datasets from The Human Cell Atlas[21], ranked by the number of cells expressing CEACAM6 in individual tissues, identified the lung, colon and bronchus as the three tissues with the most numerous CEACAM6-expressing cells (Supplementary Fig. 8). Stratification of single-cell transcriptomic data for the lung[22] by cell type identified pulmonary alveolar type 1, goblet (bronchial, tracheobronchial and nasal) and epithelial cells of the lower respiratory tract as having the most prevalent CEACAM6 expression (Fig. 2j and Supplementary Fig. 9a). These cell types are frequent targets for respiratory virus infections. The only precedents for CEACAM6 being used by pathogens for infection is as an adhesion factor for Gram-negative bacteria[23,24], for example by *Escherichia coli* in patients with Crohn's disease who have aberrant CEACAM6 expression in their ileal epithelium[25], and by the pathogenic yeast *Candida albicans*[26]. Of note, expression of CEACAM6 in the human lung is both higher and more widespread than any of the established human coronavirus proteinaceous receptors APN, ACE2 or DPP4 (Fig. 2j and Supplementary Fig. 9b). Together, these data demonstrate that CcCoV-KY43 spike protein uses human CEACAM6 as a receptor to infect cells, and this receptor is expressed in the human lung.

## KY43 RBD binds CEACAM6 N-terminal immunoglobulin domain

We used crystallography to solve the structure of a complex comprising the CcCoV-KY43 RBD and the three immunoglobulin-like domains that form the ectodomain of CEACAM6 to define the molecular architecture of their interaction (Fig. 3a). The structure was refined ($R/R_{free} = 0.256/0.317$) from crystals with anisotropic diffraction (3.0 and 5.1 Å in the best and worst direction, respectively) that contained one heterodimer of the CcCoV-KY43 RBD plus CEACAM6 per asymmetric unit (Supplementary Table 5 and Supplementary Fig. 10a). The RBD binds the tip of the N-terminal V-set immunoglobulin-like domain of CEACAM6, with receptor binding loops 1–3 of the RBD engaging the surface formed by CEACAM6 strands G-F-C-C′-C″ plus the F-G and C-C′ loops (Fig. 3b and Supplementary Table 6). This region is the same surface that mediates both homodimerization of CEACAM6 and its heterodimerization with CEACAM8 (ref. 27). Moreover, the N-terminal domain of CEACAM6 is highly similar between the complexes with CcCoV-KY43 RBD and CEACAM6 and CEACAM8, with 0.51–0.54 Å root-mean-squared deviation across 99–103 Cα atoms (Supplementary Fig. 11). The interaction buries 1,300 Å² of the surface area, less extensive than CEACAM6 homodimerization or heterodimerization (1,600 Å²) but more than the interaction between the HCoV-229E RBD and human APN (1,000 Å²)[28] and comparable with that of other coronavirus–receptor interactions (Supplementary Fig. 12). The interaction is dominated by the hydrophobic interaction

between CcCoV-KY43 RBD loop 3, especially residue W600, and the hydrophobic surface formed by strands C, C′ and F of CEACAM6, centred on residue L129 (Fig. 3c and Supplementary Table 6).

Elucidation of the structure of the binding interface enabled us to dissect the genetic determinants of CEACAM6 receptor specificity. Loops 1–3 in the RBDs from alphacoronavirus species included in our library differed markedly from CcCoV-KY43 in both their length (Supplementary Fig. 13a) and amino acid composition (Supplementary Fig. 13b). This result is consistent with the observation that only CcCoV-KY43 uses CEACAM6 for entry (Extended Data Fig. 6). Although human CEACAM5 was identified as an entry receptor in our sensitive avidity-based receptor discovery experiment, the ELISA and ITC experiments showed that it has reduced affinity for the CcCoV-KY43 RBD (Fig. 2e,f). Moreover, CEACAM5 overexpression was not sufficient for pseudotype entry (Fig. 2d). Alignment of the surface contact interface identified two important amino acid substitutions, I63F and Q123H, in CEACAM5 relative to CEACAM6 (Fig. 3d). A F63I, but not H123Q, substitution in CEACAM5 partially overcame the entry restriction of CcCoV-KY43 pseudotypes, with the corresponding substitutions in CEACAM6–I63F, Q123H or both–reducing pseudotype entry (Fig. 3d and Extended Data Fig. 7a). Note that the mutated receptors were expressed at comparable levels to the wild type for these experiments (Extended Data Fig. 7b). Detailed structural mechanics of coronavirus receptor binding have been extensively characterized, which have highlighted a fascinating pattern of either convergent or continued evolution. The architecture of the CcCoV-KY43 RBD is similar to those of the other two human alphacoronaviruses HCoV-229E and HCoV-NL63, even though their cellular receptors are structurally different (Fig. 3e). Although our data provide evidence that CEACAM6 can be used as a direct proteinaceous receptor by viruses, we note that mouse CEACAM1 is used as a receptor by the betacoronavirus MHV[29]. MHV receptor binding is mediated through the N-terminal domain of S1, whereas most protein-binding coronavirus RBDs are found in the S1 C-terminal domain, as is the case for CcCoV-KY43. Nevertheless, the target region on the CEACAM receptor remains orthologous: the N-terminal immunoglobulin domain (Supplementary Fig. 14). As the human CEACAM N-terminal immunoglobulin domains are relatively conserved (Supplementary Fig. 15, binding region in teal), we also examined CEACAM1, CEACAM4, CEACAM8 and CEACAM20 in CcCoV-KY43 entry assays (Extended Data Fig. 8 and Supplementary Table 7). None of these CEACAM proteins was able to facilitate entry of CcCoV-KY43 pseudotypes. Moreover, CEACAM1, CEACAM4 and CEACAM8 did not show appreciable binding in biolayer interferometry assays, nor did the I63F mutant of human CEACAM6 (Extended Data Fig. 9). These data highlight another example of the diversity of receptor recognition modalities among coronaviruses, with two coronaviruses from distinct genera evolving to bind the same family of receptors (in the same orthologous domain on those receptors), but through different S1 domains. These results further confirm that CEACAM usage across the rich coronavirus diversity may be underexplored.

## CEACAM6-using coronaviruses are globally distributed

CcCoV-KY43 was isolated from a bat collected in a rural setting: an artisanal mine in southeastern Kenya[18] (Fig. 4a). *C. cor* is naturally distributed across Eastern Africa[30]. However, more granular population distribution data for Kenya are currently lacking. To provide an interim assessment of sites where bat–human spillover might occur, we summarized our bat sampling data from across Kenya over the past 20 years (Fig. 4a, derived geographical range of *C. cor* coloured in teal). This analysis revealed that coastal regions in the southeast are human population centres at increased risk. Focusing on this region, we examined human sera from our biobank and assessed for the presence of CcCoV-KY43 RBD-specific antibodies (Fig. 4b). We analysed a total

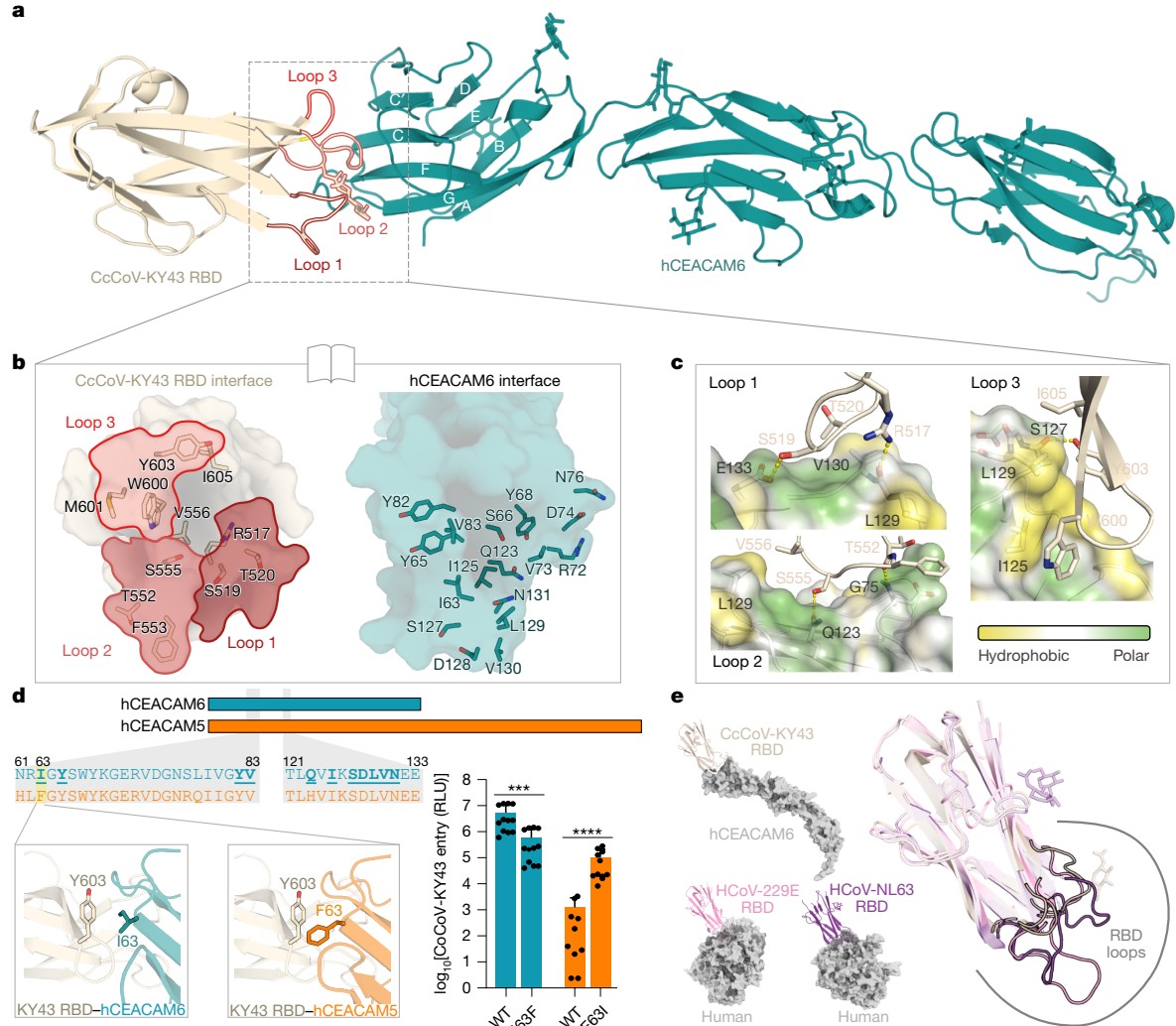

**Fig. 3 | CcCoV-KY43 RBD interacts with the IgV domain of CEACAM6.**
**a**, Structure of the CcCoV-KY43 RBD interacting with the human receptor CEACAM6. The three loops protruding from the RBD that are responsible for the interaction with CEACAM6 are highlighted. **b**, Open-book representation of the interfaces of interaction between the two proteins. The surface formed by residues constituting the three loops of the RBD are coloured as in **a**. **c**, Zoom-in to key residues involved in binding between the CcCoV-KY43 RBD and human CEACAM6. Residues of loop 1 (R517 and S519) and loop 2 (T552 and S555) establish hydrogen bonds with the surface polar regions of CEACAM6, whereas W600 and I605 on loop 3 of the RBD interact with the hydrophobic pocket of the receptor. **d**, Alignment of human CEACAM5 and CEACAM6, showing the residues interacting with CcCOV-KY43 RBD (in bold). An I63F substitution was introduced in CEACAM6, and the complementary substitution F63I in CEACAM5, followed by entry assays. The average of three independent experiments, performed in technical triplicates, is shown with s.d. values. ***$P = 0.0001$, ****$P < 0.0001$ statistical analysis using two-way ANOVA. **e**, Structure of alphacoronavirus RBDs in complex with human receptors used for entry: CcCoV-KY43 with CEACAM6, HCoV-229E with APN (Protein Data Bank (PDB) ID: 6ATK)[28] and HCoV-NL63 with ACE2 (PDB ID: 3KBH)[48]. Despite binding architecturally different proteins, the RBDs share similar overall folds and all bind their receptors using the same three surface loops.

of 368 blood donor samples collected in 2020 and 2021 from mainly male individuals (95%) under 35 years of age (72%) living in the Tana River and Taita-Taveta counties where CcCoV-KY43 was identified (Supplementary Table 8). The samples were examined for CcCoV-KY43, HCoV-229E, HCoV-NL63 and SARS-CoV-2 S-protein-specific responses by ELISA (Fig. 4b, inset, Tana River and Taita-Taveta counties highlighted in pale red). We also included two RBDs from more distantly related alphacoronaviruses (BtCoV-HlYN18 and BtCoV-A701, both isolated from bats in China) as comparators. Across the full dataset, Spearman's rank correlation analysis revealed weak monotonic associations between CcCoV-KY43 and HCoV-229E, SARS-CoV-2, BtCov-HlYN18 and BtCoV-A701 ($\rho = 0.14$–$0.30$), whereas no association was observed with HCoV-NL63 ($\rho = 0.06$) (Fig. 4b). Although ELISA reactivity against the human alphacoronaviruses HCoV-229E and HCoV-NL63 was commonly observed, increased CcCoV-KY43 ELISA signals were detected only sporadically, with optical density (OD) values exceeding one in nine samples. Using this OD threshold as a descriptive marker of high reactivity rather than evidence of seropositivity, similarly infrequent high OD signals were observed for BtCoV-HlYN18 ($n = 5$) and BtCoV-A701 ($n = 2$). Restricting the analysis to samples comprising the top 10% of CcCoV-KY43 ELISA signals, Pearson's correlation analysis of matched sera revealed moderate correlations with HCoV-NL63 and SARS-CoV-2, but only weak correlation with HCoV-229E, BtCoV-HlYN18 and BtCoV-A701 (Supplementary Fig. 16). Together, these data suggest that widespread CcCoV-KY43 spillover is unlikely in these populations. Alternatively, sporadic CcCoV-KY43 reactivity may reflect cross-reactive antibody responses induced by exposure to antigenically related coronaviruses.

Recently, two more CcCoV sequences from Meru National Park in central Kenya, about 600 km from the site of the isolation of CcCoV-KY43,

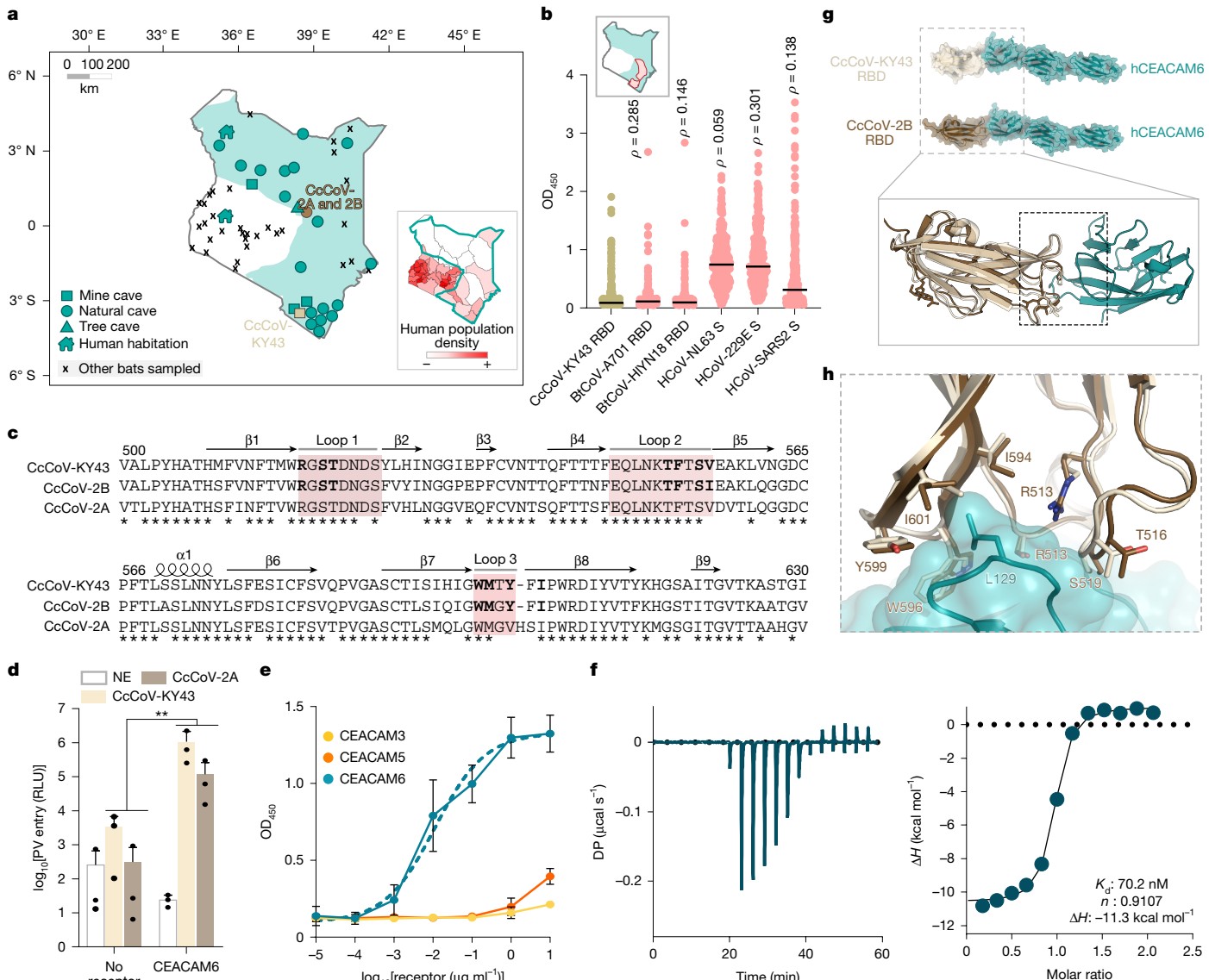

**Fig. 4 | Human CEACAM6 is the receptor for other CcCoVs identified in Kenya.**
**a**, Sites of bat sampling in Kenya and the distribution of *C. cor* (highlighted in teal). *C. cor* is found in mines (squares), natural (circles) and tree (triangles) caves and in two instances in human habitations (houses). Other sampling locations where *C. cor* was not observed are labelled with a cross. Inset, human population-dense areas of Kenya do not generally overlap with *C. cor* roosting sites. The individual sites where CcCoV were identified are shown in pale brown (CcCoV-KY43) and dark brown (CcCoV-2A and CcCoV-2B). **b**, Human sera from individuals (*n* = 368) from Tana River and Taita-Taveta counties (highlighted in pale red in the inset map; teal indicates distribution of *C. cor*) were analysed for their reactivity to different human coronavirus glycoproteins. Individual Spearman's rank correlations (*ρ*) for each dataset (compared with CcCoV-KY43) are provided. **c**, Alignment of *C. cor*-derived alphacoronavirus RBD amino acid sequences, showing high identity in the loops interacting with CEACAM6 (loops are highlighted as in Fig. 3a). The secondary structure of the RBD, based on the CcCoV-KY43 RBD, is depicted above the sequences. Residues that

interact with CEACAM6 are in bold. **d**, Entry assays with a pseudotyped spike protein of CcCoV-2A showed that human CEACAM6 confers permissivity to HEK293T cells at levels similar to CcCoV-KY43. Average of the raw data of three independent experiments, performed in technical triplicate, are shown, with s.d. values. **P* = 0.0036 for CcCoV-KY43; **P* = 0.0014 for CcCoV-2A. Statistical analysis used two-way ANOVA. **e**, Recombinant CcCoV-2B spike protein RBD was purified and used to assess binding to human CEACAM3, CEACAM5 and CEACAM6 by ELISA. Raw data of three independent experiments, performed in technical triplicate, are shown with s.d. values. **f**, ITC showed that the CcCoV-2B RBD binds CEACAM6 with high affinity. **g**, Crystal structure of CcCoV-2B in complex with human CEACAM6. Inset shows the CcCoV-2B and CcCoV-KY43 RBDs from the two complexes superposed, which highlights the similar folds of the RBDs and conserved interaction with the receptor. **h**, Zoom-in of key residues at the CEACAM6-binding interfaces of CcCoV-2B and CcCoV-KY43, highlighting the similarity of the interactions.

were published[31]: CcCoV 2A/Kenya/BAT2621/2015 (CcCoV-2A; GenBank identifier: PP273172.1) and CcCoV 2B/Kenya/BAT2618/2015 (CcCoV-2B; GenBank identifier: PP273173.1) (Fig. 4a). Notably, both viruses are relatively divergent. CcCoV-2B is more related to CcCoV-KY43; however, it shares only 79% and 83% nucleotide and amino acid identity, respectively, in the spike protein (79% and 85%, respectively, in the RBD). The more distantly related isolate, CcCoV-2A, shares only 70% and 77% nucleotide and amino acid identity, respectively, across the

spike protein (72% and 76%, respectively, for the RBD) with CcCoV-KY43 (Fig. 4c). Despite this variability, entry assays using CcCoV-2A spike protein pseudotypes showed human CEACAM6-dependent entry (Fig. 4d). In accordance with the CcCoV-KY43 data, human CEACAM5 did not support pseudotype entry. Similarly, the F63I mutant in CEACAM5 led to increased entry to levels seen with human CEACAM6 (Extended Data Fig. 10a), whereas the I63F substitution in CEACAM6 reduced entry (Extended Data Fig. 10b). In parallel, an ELISA with a

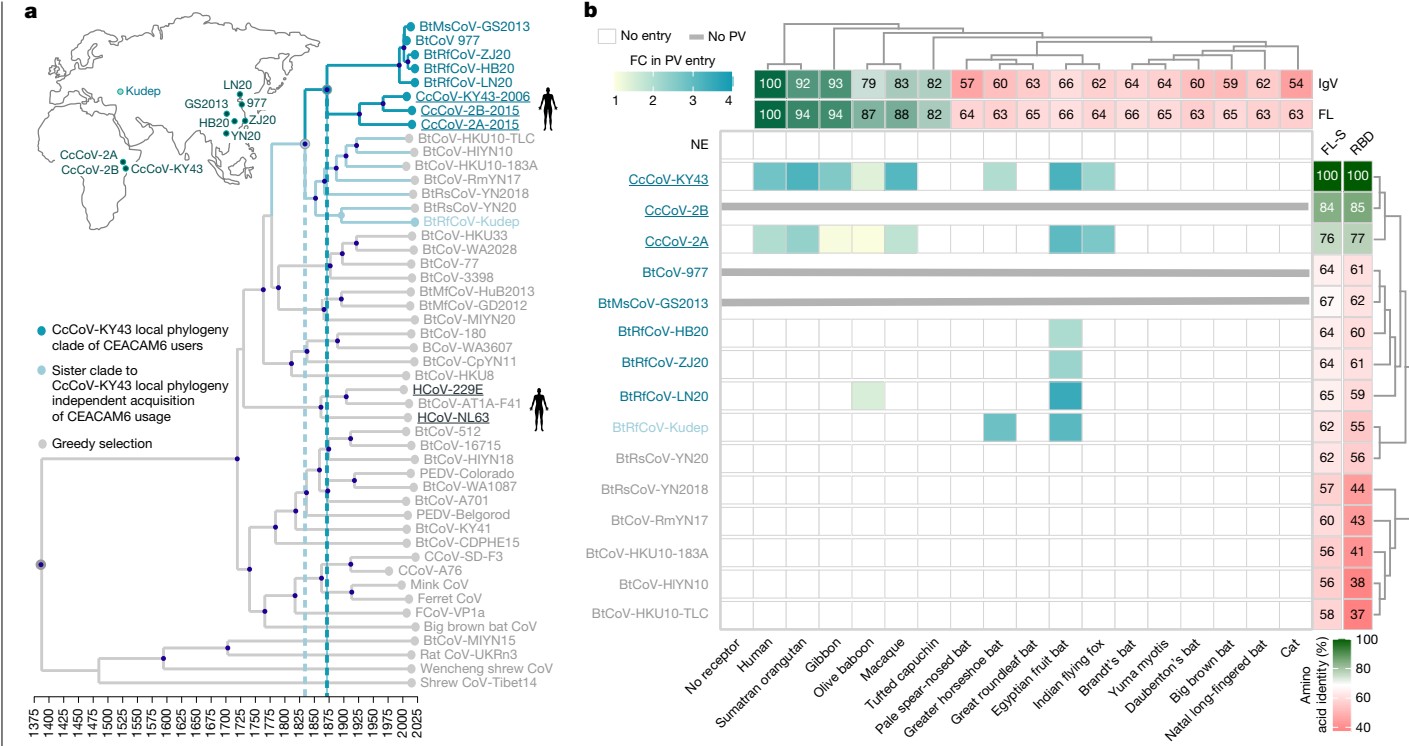

**Fig. 5 | CEACAM6 is the receptor for CcCoV-related viruses identified in Asia.**
**a**, Evolutionary reconstruction of the spike protein-encoding gene for selected alphacoronaviruses demonstrates the relatively recent acquisition of CEACAM6 usage. Viruses reported to utilize human receptors are underlined. Inset map shows locations of isolation of indicated CcCoV-related viruses. **b**, Pseudotype entry assays for CcCoV and related viruses with CEACAM6-like proteins from different mammalian species (percentage amino acid identity compared with

human CEACAM6 or CcCoV-KY43 for the full-length spike protein (FL-S) or the IgV RBD domain (RBD), respectively, is shown). Viruses closely related to CcCoV-2A, CcCoV-2B and CcCoV-KY43 users were included in the analysis, which informed the evolutionary acquisition of receptor usage. Positive results were reproduced three times, using independent biological replicates. All experiments were performed in technical triplicate with PV entry indicated as the fold change above background.

CcCoV-2B recombinant RBD confirmed binding to human CEACAM6 (Fig. 4e), and ITC showed that it had higher affinity for CEACAM6 than CcCoV-KY43 (70 ± 16 nM; Fig. 4f and Supplementary Table 4). The structure of the CcCoV-2B RBD in complex with human CEACAM6 was solved from crystals with an anisotropic diffraction limit of 3 Å (Supplementary Table 5). The structures showed an identical quaternary organization to the structure of human CEACAM6 in complex with the CcCoV-KY43 RBD (Fig. 4g) and highly similar residues at the interaction interface (Fig. 4h and Supplementary Fig. 10b).

The use of human CEACAM6 by genetically divergent CcCoVs strongly suggest that CEACAM6 usage might be a conserved ancestral trait of a larger group of uncharacterized alphacoronaviruses. To explore the evolutionary history of CEACAM6 receptor usage, we assembled a library of spike proteins from related viruses and reconstructed the 'local phylogeny' of CcCoVs on the alphacoronavirus tree (Supplementary Table 9 and Supplementary Fig. 17a). Overall, CcCoVs were phylogenetically placed in a sister clade to alphacoronaviruses isolated from *Rhinolophus ferrumequinum* bats in South China (Fig. 5a). The CcCoV clade was returned as monophyletic in all sampled posterior trees (*n* = 9,000, 95% binomial confidence interval (CI) of 0.999–1.000), with its most recent common ancestor (MRCA) estimated to have evolved around 1833 (95% highest posterior density (HPD) of 1794–1884). Time-dependent rate effects in viral molecular evolution suggest that our date estimates for the CEACAM6-adapted lineage are likely to be conservative (that is, biased towards more recent times). Across diverse viruses, substitution rate estimates are systematically higher over short time scales and decay as the measurement interval increases, consistent with a general time-dependent rate phenomenon[32]. Mechanistic work further implicated purifying

selection and multiple-hit saturation as key drivers of this bias, which causes long-term evolutionary change to accumulate more slowly than expected under a simple, constant-rate clock[32]. For coronaviruses specifically, models that explicitly accommodated variable selection pressure and substitution saturation pushed the inferred origin of coronavirus radiation from around 10^4 years to an ancient time scale of millions of years, a result in closer agreement with the diversification of their hosts[33]. A full investigation of the time-dependent rate phenomenon (for example, using epoch or heterogenous clock models[34]) is beyond the scope of this study. However, we acknowledge that estimates for the time to MRCA reported here should be interpreted as lower bounds and that the acquisition of CEACAM6 usage may predate our point estimates. To understand the potential host range of these viruses in more detail, entry through additional CEACAM6-like proteins from primates, bats and other mammals was examined (Fig. 5b, Supplementary Table 10 and Supplementary Fig. 17b). *C. cor* are taxonomically classified in the family Megadermatidae (African false vampire bat) and the superfamily Rhinolophoidea, which also contains the common coronavirus reservoir bat species *Hipposideros* and *Rhinolophus*. Typically, nonhuman primates and bats in the Yinpterochiroptera suborder (Megadermatidae, Rhinolophidae, Hipossideridae, Rhinopomatidae, Craseonycteridae and all fruit bats) have four to six *CEACAM* genes that produce CEACAM6-like proteins[35]. Notably, this is not the case for Yangochiroptera bats, which have undergone large expansions in their *CEACAM* gene repertoire[36]. As there are currently no Megadermatidae *CEACAM6* sequences publicly available, we screened a wide diversity of other bat species. Our screens identified that human CEACAM6 tropism is exclusive to the Kenyan CcCoV isolates (Fig. 5b). The CcCoV-KY43 and CcCoV-2A pseudotypes showed broad use of primate CEACAM6

receptors as well as various Yinpterochiroptera bats, including Indian flying fox, Egyptian fruit bat, large flying fox and greater horseshoe bat. Moreover, pseudotyped spike proteins from alphacoronaviruses isolated from *R. ferrumequinum* bats in South China[37] and Southern European Russia[38] were able to use CEACAM6 as receptors to enter cells, but only from Egyptian fruit bat and/or greater horseshoe bat (Fig. 5b). ELISA experiments showed that the recombinant RBD from a related virus, BtCoV-977, which did not pseudotype, can bind Egyptian fruit bat, but not human, CEACAM6 (Extended Data Fig. 11). No consistent amino acid substitution pattern in these viral RBDs distinguished CEACAM6-like proteins that permit CcCoV entry from those that do not (Extended Data Fig. 12a). However, viruses that use human CEACAM6 conserve multiple RBD sequence features that are absent from other RBDs, for example, the length and amino acid compositions of loops 1 and 2 (Extended Data Fig. 12a). We introduced point mutations into the BtCoV-LN20 RBD, which uses various CEACAM6-like proteins but not the human orthologue, to better define the genetic determinants of human receptor usage and zoonotic spillover (Extended Data Fig. 12b). Although we could expand the host range phenotype of BtCoV-LN20, individual changes did not confer human CEACAM6 tropism, which indicated that multiple changes are needed to induce this characteristic and that the risk of rapid adaptation of CcCoV-related viruses to zoonotic spillover is low. To investigate the potential acquisition (or loss) of CEACAM6 usage along the full evolutionary history of alphacoronavirus, we further explicitly modelled CEACAM6 usage as a discrete trait in a Markov jump framework to assess whether the data support single versus multiple gains (or losses). The posterior distribution of CEACAM6 transitions supports ≥2 independent gains (median number of gains = 2; 95% HPD of 1–3), whereas losses are rare and are not mapped on the resulting tree generated using the method highest independent posterior subtree reconstruction (median number of gains = 0, 95% HPD of 0–1). Consistent with this result, the posterior probability that the 'focal MRCA' used CEACAM6 is 0.97. In the reconstructed history, CEACAM6 usage in *Rhinolophus ferrumequinum* bat (BtRf) CoV-Kudep arises on a branch leading to the MRCA of a sister clade to the local phylogeny, whereas the MRCA of these two sister clades itself is reconstructed as not using CEACAM6 (Supplementary Fig. 18). This pattern is most parsimoniously explained by two independent acquisitions of CEACAM6 usage: one along the lineage giving rise to the local phylogeny and one along the lineage leading to the *Rhinolophus sinicus* (BtRs) CoV-YN20 and BtRfCoV-Kudep strains. Collectively, these data provide evidence that CEACAM6 is a receptor for a broad range of geographically divergent alphacoronaviruses found across East Africa, European Russia and China.

## Discussion

As a consequence of the COVID-19 pandemic, there has been a significant increase in research on coronavirus discovery, reservoir characterization and spillover, as well as the development of broad-acting antivirals and therapeutics[39–42]. However, most studies have focused on betacoronaviruses, and the zoonotic and pandemic potential of alphacoronaviruses has remained relatively uncharacterized. Indeed, sequencing efforts to understand the origin of alphacoronaviruses has identified a rich diversity of virus genotypes in reservoir species such as rodents and bats[31,43]. Here we used an approach aimed at analysing and capitalizing on viral heterogeneity at the genus-wide level to gain a broad understanding of receptor usage and host tropism. We confirmed the importance of APN and ACE2 for human, livestock and companion animal alphacoronavirus infections. However, the broader trend was that use of these two receptors is poorly conserved across the genus. This was especially true for spike proteins for which ORFs were sequenced after sampling of bats. Indeed, only 2 out of 25 functionally pseudotyped bat coronavirus spike proteins use a recognized alphacoronavirus receptor. One limitation of our study

is that only 28 out of the 36 pseudotyped viruses have a single species assigned as a host, and for these 28 species, the corresponding receptor sequences are known in only 50% (APN) and 79% (ACE2) of cases. We directly matched 10 out of 28 and 9 out of 28 of viruses to their cognate host's APN and ACE2 receptors, respectively. These numbers increased to 13 out of 28 and 18 out of 28, respectively, when the identity threshold for the receptors was reduced to >85%; however, only 5 pseudotypes used APN and only 1 used ACE2 (Supplementary Figs. 4 and 5). Although we cannot formally discount the hypothesis that some bat alphacoronaviruses are hyperspecialized to only one cognate bat APN or ACE2 that was not included in our libraries, our results strongly indicated the presence of other receptors. The subsequent discovery that one of these viruses—CcCoV-KY43 from Kenya—has at least partial tropism for human cells provides a critical risk assessment for regional and global health communities to prepare for any potential spillover of these viruses. This assessment is strengthened by the following findings: identification of the CcCoV receptor CEACAM6; examination of host reservoir distribution in Kenya; sero-surveillance in human populations; and finally, wider elucidation of receptor use by related bat viruses from outside Africa. These results are crucial when considering that bat coronaviruses closely related to the endemic human alphacoronaviruses HCoV-229E and HCoV-NL63 have been found across Sub-Saharan Africa (Supplementary Fig. 19), consistent with previous alphacoronavirus zoonotic spillover events in this region[44,45].

Many viruses use the N-terminal V-set immunoglobulin domain of cellular adhesion molecules for cell attachment[46], including MHV, which binds CEACAM1 (ref. 29). The high abundance of CEACAM family proteins on the apical surfaces of mucosal epithelia make them highly suitable for pathogen attachment. Indeed, *C. albicans*[26] and several Gram-negative bacteria[24] use CEACAM proteins for adhesion. Given their immunoglobulin-domain architecture and abundance on barrier membranes, we propose that additional coronaviruses use CEACAM family proteins for cell entry. The ultimate goal of pandemic preparedness is to be able to predict and risk assess the zoonotic potential of viruses from their genome sequence alone. Various approaches are being leveraged to do this, one such being the application of machine-learning algorithms to predict zoonotic potential[47]. By integrating computational, unbiased selection of sequences with high-throughput screening, receptor identification, structural characterization, field epidemiology and sero-surveillance, we identified a previously uncharacterized receptor used by alphacoronaviruses from Africa, Europe and East Asia. This finding both identifies a potential threat to human health and provides the underpinning characterization to enhance pandemic preparedness and prevention efforts.

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

## Methods

### Ethics statement

The use of human sera for this study was approved by the Scientific and Ethics Review Unit of the Kenya Medical Research Institute (protocol SSC 3426). Before the blood draw, donors gave individual consent for the use of their samples for research.

### Construction of gene libraries

A schematic of the downstream analysis pipeline used for retrieving alphacoronavirus data and for generating the final spike protein library used in this study ($n = 40$) is provided in Fig. 1a. All publicly available alphacoronavirus genome sequences were retrieved from the Virus Pathogen Database and Analysis Resource platform hosted by the Bioinformatics Resource Center at the National Institute of Allergy and Infectious Diseases[15]. As of May 2021, the full database consists of 19,082 alphacoronavirus genomes, from which we extracted sequences of the whole spike protein-coding region to obtain a final database of 2,714 sequences. We constructed the spike protein-coding DNA sequence alignment using MAFFT (v.7.526)[49,50] by integrating structural alignments of homologous spike protein structures queried from the UniProt Reference Clusters[51]. Maximum-likelihood phylogenetic reconstruction was performed using IQTREE (v.2.3.4)[52] with 1,000 ultrafast bootstrap replicates (UFBoot) and 1,000 SH-like approximate likelihood ratio tests[53] (Supplementary Fig. 20). We performed codon model reconstruction by determining the best-fit model using the model selection procedure implemented in IQTREE. Patristic distances (the sum of branch lengths along the shortest path) were then computed between all pairs of tips in R (v.4.4.1) using the ape package[54], which informed the unbiased selection of the $n = 40$ spike protein-coding sequences by applying a greedy algorithm. In brief, let $\delta(i,j)$ denote the patristic distance between tips $i$ and $j$ on the reconstructed maximum-likelihood tree, with the branch length estimated in substitutions or sites, and let $k$ be a previous number of tips to be selected. The greedy algorithm identifies the farthest pair $\arg\max_{i,j}\delta(i,j)$ and initializes the final selection set $S$ with these two tips. Next, for each subsequent selection step, it adds the tip $x \notin S$ that maximize its nearest-neighbour distance to the current set $x = \arg\max_{x \notin S}\min_{y \in S}\delta(i,j)$ and repeats the process until $|S| = k$. Although heuristic, this approach ensures that an optimal subset of tips (evolutionary units) is returned under the assumption of maximizing both minimum phylogenetic distance and phylogenetic diversity, as previously theoretically proposed and discussed[14]. We report Faith's phylogenetic diversity[55] of the induced minimal subtree and benchmarked against a 10,000 random panels of matching size. For the most divergent alphacoronaviruses, a RBD could not be readily identified for two unclassified viruses and viruses classified in the *Soracivirus* and *Luchacovirus* subgenera.

### Plasmids used for pseudotyping

Selected alphacoronavirus spike protein-coding sequences were ordered from Biobasic as codon-optimized synthetic genes and subcloned in pcDNA3.1 with a HA tag in the C-terminal cytoplasmic tail. For the APN library, genes were ordered from GenScript and subcloned with a N-terminal V5 tag in the vector pCAGGS. For the ACE2 library and human DPP4 (GenBank: NP_001926), the ectodomains of the ORFs were subcloned with a N-terminal HA tag in pDisplay. CEACAM libraries were obtained from GenScript and subcloned in pcDNA3.1(+) with a C-terminal 8×His tag. Flag-tagged human TMPRSS2 (GenBank: NP_001128571.1) and human DPEP1 (GenBank: NP_001121613.1) were cloned into pcDNA3.1. GenBank accession numbers of the sequences used for the study are provided in Supplementary Table 1 for the alphacoronavirus spike protein library, Supplementary Table 2 for APN, Supplementary Table 3 for ACE2, Supplementary Table 7 for human CEACAM receptors, Supplementary Table 9 for the CcCoV local phylogeny spike protein library and Supplementary Table 10 for CEACAM6-like proteins of different mammalian species.

### Cells

Mycoplasma-free HEK293T (human kidney), Calu3 (human lung), Caco2 (human colorectal adenocarcinoma) and HuH7 (human hepatoma) were cultured in Dulbecco's modified Eagle medium (DMEM) supplemented with 10% FBS, 100 U ml$^{-1}$ penicillin plus 100 µg ml$^{-1}$ streptomycin (PenStrep) and 1 mM sodium pyruvate. Mycoplasma-free THP1 (human monocytes) and LCL (human B lymphocytes) cell lines were maintained in suspension in RPMI1640 medium supplemented with 10% FBS and PenStrep. All reagents for cell culture were purchased from Gibco. All cells were cultured in a humidified atmosphere at 37 °C, 5% $CO_2$.

### Pseudotype virus production

To pseudotype alphacoronavirus spike proteins, plasmids encoding their ORF were transfected in confluent HEK293T cells seeded in a 6-well plate using polyethylenimine (PEI, 5 µg ml$^{-1}$). HIV-1-based lentiviral vectors coding for viral structural proteins (p8.91), particle packaging signals and a luciferase reporter gene (pCSFLW) were also included in the transfection mix[56]. The following day, the medium was replaced and pseudoviruses in the supernatant were collected 48 and 72 h after transfection and pooled together. Following the final collection, the supernatant was centrifuged for 10 min at 4,000 rpm to remove cellular debris. Finally, pseudoviruses were aliquoted and stored at −80 °C until further use. To verify spike protein incorporation, pseudoparticles were purified by ultracentrifugation at 23,000 rpm, 4 °C for 2 h using a 20% sucrose gradient. Supernatants were discarded and pellets were resuspended in PBS. Concentrated pseudoparticles were lysed by boiling with Laemmli (Bio-Rad) and spike protein expression was analysed by SDS–PAGE. Separated proteins were transferred onto a 0.45 µm nitrocellulose membrane (Cytiva), blocked in PBS supplemented with 0.05% Tween-20 and 5% (w/v) unskimmed milk powder and incubated with mouse monoclonal anti-HA (clone 6E2, Cell Signaling Technology, 1:5,000) and anti-p24 (clone 5, Abcam, 1:2,000) overnight at 4 °C. The following day, goat anti-mouse DyLight 680 (Invitrogen, 1:10,000) was used to probe primary antibodies, and signals were detected with an Odyssey DLx imaging system (Li-Cor Biosciences). Of note, for those spike proteins for which we saw entry (Fig. 1; HCoV-NL63, CcCoV-SD F3, CcCoV-A76, HCoV-229E, BtCoV-AT1A F41 and BtCoV-WA1087), protein expression in the purified pseudoparticle immunoblots (Extended Data Fig. 1) did not quantitatively correlate with entry signals, which indicated that it is difficult to establish a minimum threshold of spike protein incorporation needed for membrane fusion. For the six spike proteins that did not pseudotype (Supplementary Fig. 1), we attempted to substitute them for spike proteins that have been reported to pseudotype and that share at least 95% amino acid similarity (Supplementary Fig. 21). However, the only match was for PEDV-KDJ, which was replaced by PEDV-Colorado[57]. The selected HCoV-229E spike protein sequence pseudotyped to low levels and was not functional in our downstream applications (Supplementary Fig. 3a); therefore, we replaced it with a different strain of the same species. Original uncropped immunoblots are provided in the Zenodo repository (https://doi.org/10.5281/zenodo.17951484)[58].

### Pseudovirus entry assay

For receptor usage screening, HEK293T cells were transfected with plasmids coding for the receptors of interest. The following day, the medium was removed and replaced with fresh DMEM supplemented with 10% FBS to a final concentration of $2 \times 10^4$ cells per ml. Next, 100 µl of cell preparation was aliquoted in each well of a 96-well plate and treated with 100 µl pseudovirus preparation, diluted 1:1 with fresh medium. Two days later, the supernatant was removed and cells were treated with Bright-Glo (Promega), diluted 1:1 with PBS. Luciferase signals were acquired using a Glomax Discover luminometer (Promega). To assess cell line permissivity to pseudovirus infection, confluent

human cell lines were transduced with undiluted pseudoparticle supernatant in a 96-well plate. Two days later, plates were spun down, the medium replaced with 50 µl Bright-Glo and luciferase signals were acquired on a luminometer. To inhibit human CEACAM6-dependent entry, pseudotyped CcCoV-KY43 spike protein was incubated with commercial monoclonal antibodies against human CEACAM6, clone B6.2 (ThermoFisher) and clone 439424 (ThermoFisher), or recombinant C-terminally Fc-tagged human CEACAM6, for 1 h at room temperature in a 96-well plate. HEK293T cells transiently transfected with pcDNA-human CEACAM6 was then added to the plate at a concentration of $2 \times 10^4$ cells per well. Two days later, luciferase signals were measured as described above.

### Flow cytometry for expression of the receptor libraries
Plasmids were transfected in HEK293T cells using Trans-IT X2 (Mirus Bio). The following day, cells were resuspended in PBS, fixed using 2% paraformaldehyde for 20 min at 4 °C and permeabilized using PBS supplemented with 0.5% Triton X-100 for 5 min at 4 °C. After washing, cells were incubated for 1 h with anti-tag antibodies conjugated with PE for APN (V5 tag, Invitrogen, clone TCM5, 1:500) and APC for ACE2 (Flag-tag, Miltenyi Biotec, clone REA216, 1:500) and CEACAM (His-tag, Miltenyi Biotec, clone GG11-8F3.5.1, 1:500) libraries. Samples were run on a MACSQuant Analyzer 10 cytometer (Miltenyi Biotec), and data analysis was performed using FlowJo (BD Biosciences). To assess the specificity of anti-human CEACAM6 monoclonal antibodies, HEK293T cells were transfected with a library of different human CEACAM proteins. The following day, cells were washed in PBS and fixed using 2% paraformaldehyde for 20 min at 4 °C. After washing, cells were incubated with clone B6.2 (1:500) or clone 439424 (1:500) for 1 h on ice. Cells were washed three times before incubation with the secondary antibody conjugated with FITC and recognizing mouse IgG(H+L) (Invitrogen, 1:5,000). Samples were run on a MACSQuant Analyzer 10 cytometer and data were analysed using FlowJo.

### Human virus receptor discovery
Human receptor ectodomains were expressed as enzymatically monobiotinylated soluble proteins as previously described[20] by co-transfecting HEK293 cells with a secreted BirA protein biotin[59], as previously described[60]. Protein-containing supernatants were collected after 5 days of expression at 37 °C with 5% $CO_2$ and 70% humidity. Cells were removed by centrifugation 2,000g for 20 mins, before filtering away particulates (Acrodisc PF PES 0.8/0.2 micron filters, 4658). The imidazole concentration was adjusted to 20 mM and applied to a 96-well HisTrapHP plate as previously described[61]. Non-captured proteins were removed using 3 washes with buffer (20 mM sodium phosphate buffer, 400 mM NaCl and 40 mM imidazole, pH 7.4). Captured proteins were eluted after 15 min of incubation with elution buffer (20 mM sodium phosphate, 400 mM NaCl and 400 mM imidazole, pH adjusted to 7.4). Protein concentrations were determined using a Pierce Bradford protein assay kit, comparing the absorbance values for the purified proteins against values obtained for a dilution series of bovine serum albumin (BSA). Protein purity was assessed by separating protein preparations under reducing conditions (NuPAGE sample reducing agent and LDS sample loading buffer) by SDS–PAGE in MOPS–SDS buffer (NuPAGE) and visualized using Coomassie dye (InstantBlue). Human receptor screening assays were carried out as previously described[20]. In brief, horseradish-peroxidase-labelled preys were produced by complexing monobiotinylated proteins (1.785 ml at 17.5 nM) with streptavidin HRP (714 µl of a 1 in 1,000 dilution, Pierce: 21130) for 1 h at 23 °C, diluted in HEPES buffered saline (HBS) containing 2% (w/v) BSA. Preys were further diluted 20-fold in the same buffer and applied to $2 \times 384$-well plates containing 759 immobilized human receptor ectodomains and incubated for 1 h at 23 °C. Plates were washed twice with HBS containing Tween-20 (0.1% w/v), followed by a final wash in HBS. Protein interactions were visualized using TMB/E solution (Merck ES001), stopping

the reaction by the addition of NaF to a final concentration 0.15% (w/v). Absorbance readings at 652 nm were processed using a median polished $Z$ score with the significance threshold set to $Z > 2$.

### Site-directed mutagenesis
Substitutions in human CEACAM5 and CEACAM6 were introduced using a Quikchange Lightning Site-Directed Mutagenesis kit (Agilent) following the manufacturer's instructions. Primers were designed using the Agilent online tool (https://www.agilent.com/store/primerDesignProgram.jsp).

### Cell–cell fusion assay
HEK293T cells were transfected with either the human CEACAM receptor constructs and rLuc-GFP 1-7 plasmid[62] or with plasmids encoding CcCoV-KY43 spike protein and rLuc-GFP 8-11 using Transit-X2 transfection reagent (Mirus) according to the manufacturer's instructions. The following day, cells were resuspended in fresh medium and co-cultured at a ratio of 1:1 to a final density of $4 \times 10^4$ cells per well in a 96-well plate. Two days later, medium was removed and cells were lysed in Passive Lysis buffer (Promega). Renilla luciferase substrate coelenterazine-H (Promega) was added and signals were read using a Glomax Discover Reader (Promega).

### Knockdown of *CEACAM6* in human cells
siRNA targeting human *CEACAM6* (Dharmacon) was introduced in Calu3 and Caco2 cells by electroporation (Neon NxT, ThermoFisher). In brief, cells were washed, resuspended in electroporation buffer at a final concentration of $1 \times 10^8$ cells per ml and mixed with siRNA at 100 mM. Electroporation was carried out for 2 pulses of 20 ms at 1,400 V. Afterwards, cells were grown for 24 h in DMEM supplemented with 20% FBS before transduction with pseudotyped CcCoV-KY43 spike protein. Entry signals using firefly luciferase were detected as described above. To evaluate reduction in protein expression, cellular lysates were obtained at the same time. Western blot membranes were incubated with mouse anti-human CEACAM6 (Fisher Scientific, clone B6.2, 1:1,000) and rabbit anti-human GAPDH (Proteintech, 1:10,000) as a loading control. The following day, goat anti-mouse DyLight 680 (Invitrogen, 1:10,000) was used to probe primary antibodies, and signals were detected using an Odyssey DLx imaging system (Li-Cor Biosciences). In parallel, stable Caco2 and Calu3 cell lines for which expression of human *CEACAM6* was knocked down using shRNA were obtained. Hairpin sequences of 21-mers targeting the gene were obtained from the web portal of the Genetic Perturbation Consortium (Broad Institute, https://portals.broadinstitute.org/gpp/public/seq/search). The sequences TRCN0000424513 and TRCN000062298 were purchased as oligomers from IDT and cloned into the pKLO.1C vector (Addgene, 139470; PMID 16564017). The final constructs were transfected into HEK293T cells, along with the lentiviral packaging vector p8.91 and a plasmid encoding for VSV glycoprotein G. Pseudoviruses were collected as described above. Caco2 and Calu3 cells were plated in a 6-well plate and transduced when 80% confluency was reached. Pseudoviruses were added for 3 days, then removed and replaced with DMEM 20% FBS containing 8 µg ml$^{-1}$ puromycin. Functional entry assays were performed as described above, and reduced expression of CEACAM6 was again confirmed by immunoblotting. Original uncropped immunoblots are provided in the Zenodo repository (https://doi.org/10.5281/zenodo.17951484)[58].

### Transcriptomic analysis of available datasets
Publicly available single-cell RNA-sequencing data from the Human Protein Atlas were downloaded (https://www.proteinatlas.org/human-proteome/single+cell/single+cell+type/data#datasets). The dataset included gene read counts per cell for 31 human organs. Count matrices per organ were imported into R (v.4.3.3) and processed using Seurat (v.5.3.0) in RStudio (v.2025.05.1). To further zoom in on lung-specific

single-cell transcriptomic data, we downloaded the Human Lung Cell Atlas (HLCA v.1.0) dataset for normal human lungs samples (https://data.humancellatlas.org/hca-bio-networks/lung/atlases/lung-v1-0). Processed cell-type annotations and expression matrices were loaded into Seurat (v.4.3.3.) for downstream analyses.

## Recombinant protein production

Receptors (human CEACAM6, residues 1–326; human CEACAM5, residues 1–684; human CEACAM3, residues 1–215; human CEACAM1, residues 1–428; human CEACAM8, residues 1–327; and Egyptian fruit bat CEACAM6, residues 1–388) were cloned into pOPINTT vectors, upstream of the human rhinovirus 3C protease site and human IgG-Fc tag. Full-length spike proteins (SARS-CoV-2, HCoV-NL63 and HCoV-229E) were cloned into pCDNA3.1 vectors. RBDs (CcCoV-KY43, residues 500–630; CcCoV-2B, residues 496–625; BtCoV-HlYN18, residues 490–627; BtCoV-A701, residues 488–622; BtMfCoV-HuB2013, residues 497–641; BtCoV-WA3607, residues 497–652; BtCoV-CpYN11, residues 491–641; BtCoV-180, residues 477–633; BtCoV-HKU8, residues 497–644; BtCoV-RmYN17, residues 506–632; BtCoV-77, residues 495–634, BtCoV-WA2028, residues 497–648; BtCoV-977, residues 508-638; mink coronavirus, residues 518–660; and CCoV-SD-F3, residues 525–679) were cloned into pOPINTT vectors, which encode a 6×His tag at the C terminus of the protein. Mycoplasma-free Expi293 cells were cultured in Expi293 Expression medium (Gibco) at 37 °C, 8% $CO_2$. Expression plasmids were transfected using Polyethylenimine Max (PEI 40K, Polyscience) at a mass ratio of 1:1.5 according to the manufacturer's instructions. After 18 h, valproic acid (Merck), sodium propionate (Merck) and glucose (Merck) were added to the cellular suspension at a final concentration of 5 mM, 6.7 mM and 46 mM, respectively. After 3 days, supernatant was collected by centrifugation (3,800g for 10 min) and sterile filtered before storage at 4 °C. Fc-tagged proteins were purified using HiTrap protein G HP (Cytiva) prepacked affinity columns, washing with 10 mM sodium phosphate pH 7 buffer, eluting with 0.1 M glycine pH 2.7 that was immediately neutralized with 1 M Tris pH 8.0. His-tagged proteins were purified using HisTrap FF (Cytiva) prepacked affinity columns, washing with 10 mM sodium phosphate pH 7.5, 150 mM NaCl, 20 mM imidazole and eluting with 10 mM sodium phosphate pH 7.5, 150 mM NaCl and 1 M imidazole. For all samples, eluted fractions were analysed by SDS–PAGE, and fractions containing the relevant protein were pooled. Pooled protein was exchanged into 10 mM Tris pH 7.5, 150 mM NaCl via repeated concentration and dilution using Amicon Ultra Centrifugal filters (Merck). For crystallization, the CEACAM6 Fc tag was removed by overnight incubation at 4 °C with human rhinovirus 3C protease, and both the Fc tag plus uncleaved CEACAM6-Fc were depleted using a HiTrap protein G HP affinity column. For crystallography and biophysics (ITC and biolayer interferometry assays), proteins were further purified by size-exclusion chromatography using a Superdex 200 10/300 GL column (Cytiva) equilibrated in 10 mM Tris pH 7.5, 150 mM NaCl. Proteins were stored at 4 °C (<2 weeks) or snap-frozen and stored at –80 °C (long term).

## Protein binding determined by ELISA

Recombinant RBDs were coated onto Maxisorp NUNC-immuno flat-bottomed 96-well plates (Thermo Scientific) at 1 μg ml⁻¹ overnight at 4 °C in carbonate–bicarbonate solution (0.6 M, pH 9.6). Plates were blocked for 1 h at room temperature with PBS Tween-20 containing 2% BSA (Merck), after which a dilution series of CEACAM proteins in PBS BSA 2% was added for 1 h at room temperature. Plates were washed 3 times with PBS Tween-20 0.1% (Sigma), followed by the addition of anti-human-Fc HRP conjugate diluted 1:10,000 for 1 h at room temperature. 1-step Ultra TMB (Merck) was added to each well, incubated for 5–10 min at room temperature and the reactions stopped with an equivalent volume of 2 M sulfuric acid solution. The OD at 450 nm was measured using a Glomax Discover luminometer (Promega).

The binding IgG ELISAs with human sera were performed according to previously published protocols[63]. In brief, Maxisorp NUNC-immuno flat-bottomed 96-well plates were coated with 2 μg ml⁻¹ coronavirus spike antigens and CcCoV-KY43, BtCoV-HlYN18 and BtCoV-A701 RBDs at 37 °C for 1 h, then washed 3 times in 0.1% Tween-20 and blocked with blocker casein (Thermo Fisher) for 1 h. As full-length spike protein ectodomains could not be produced in sufficient quantities for CcCoV-KY43, ELISAs used stabilized full-length spike proteins for HCoV-229E, HCoV-NL63 and SARS-CoV-2 and RBDs for CcCoV-KY43, BtCoV-HlYN18 and BtCoV-A701; this modification represents a limitation of the study and precludes direct quantitative comparison of ELISA signal magnitudes across antigens. Samples were diluted 1:800 in blocker casein and added to both receptor binding domain and spike protein-coated plates, and incubated for 2 h at room temperature. After washing with 0.1% Tween-20, a 1:10,000 dilution of horseradish peroxidase-conjugated goat antihuman IgA antibody (Sigma) in wash buffer was added to plates, incubated for 1 h at room temperature, washed and $O$-phenylenediamine dihydrochloride substrate (Sigma) was added for colour development for 10 min. Absorbance was measured at 492 nm. Spearman's rank correlations were applied to assess associations across the full dataset, whereas Pearson's correlations were applied to the top 10% of CcCoV-KY43 ELISA signals to explore co-variation among high responders. As true negative control sera are unavailable in this emergent-virus setting, OD thresholds were used solely as descriptive markers of high reactivity (in each antigen-specific ELISA) and not to infer seropositivity or to compare absolute signal magnitudes across assays.

## Flow cytometry for receptor–viral protein binding

HEK293T cells were transiently transfected with plasmids encoding APN and ACE2 from different species, as well as human TMPRSS2, human DPP4 and human DPEP1. The following day, cells were washed in PBS and fixed using 2% formaldehyde in PBS for 20 min on ice. After incubation, cells were permeabilized with Triton 1% in PBS, for 5 min on ice. Cells were washed twice in PBS and incubated with 10 mg ml⁻¹ viral-predicted RBD diluted in PBS BSA 1%, for 1 h on ice. After washing, the secondary antibodies were incubated with the cells for 1 h on ice. To label the receptors, PE-conjugated anti-V5 (Invitrogen, clone TCM5, 1:500) was used for the APN library, PE-conjugated anti-HA (Miltenyi Biotec, clone GG8-1F3.3.1, 1:500) was used for the ACE2 library and human DPP4 and PE conjugated anti-Flag (BioLegend, clone L5, 1:500) was used for human TMPRSS2 and human DPEP1. To recognize the His-tagged viral predicted RBDs, APC-conjugated anti-6×His was used (Miltenyi Biotec, 1:500). Cells were run on a MACSQuant flow cytometer (Miltenyi Biotec), and at least 25,000 events were acquired per sample. Data were analysed using FlowJo (BD Biosciences). The gating strategy is shown in Supplementary Fig. 22.

## Crystallization

Complexes of CEACAM6 with RBDs from CcCoV-KY43 and CcCoV-2B were formed by mixing the proteins at a 1:1.2 molar ratio and incubating for 60 min at 22 °C. Proteins were crystallized in 96-well nanolitre-scale sitting drops (200 nl protein plus 200 nl of reservoir) equilibrated at 293 K against 80 μl reservoir solution as follows: 2 mg ml⁻¹ CcCoV-KY43 complex, 0.1 M sodium acetate pH 4.3, 21.5% PEG5K MME and 5% glycerol; 2.7 mg ml⁻¹ CcCoV-2B complex, 0.1 M sodium acetate, 20% PEG 6K and 0.2 M NaCl. Crystals were cryo-preserved by brief immersion in reservoir supplemented with 20–25% glycerol before collecting in SPINE standard nylon loops and flash-cryocooling in liquid nitrogen.

## X-ray data collection and structure determination

Diffraction data were recorded from single crystals on the Diamond beamline I24 at 20 keV using an Eiger2 9M detector (CcCoV-KY43 complex) or on the beamline I04 at 13 keV using an Eiger2 X 16M detector (CcCoV-2B complex). Both crystals produced severely anisotropic

diffraction; therefore, data were integrated using DIALS[64] before anisotropic scaling and merging using the STARANISO 'aniso merge' data processing pipeline[65]. The structure of human CEACAM6 in complex with CcCoV-KY43 was solved by molecular replacement using PHASER[66] with AlphaFold 3 (ref. 67) models of the ectodomain (residues 35–326) of CEACAM6 and the RBD (residues 500–630) of CcCoV-KY43, for which the atomic pLDDT values had been converted into pseudo-atomic displacement factors[68]. The model was manually improved using ISOLDE[69] with adaptive restraints based on the AlphaFold quality metrics[68] before refinement using phenix.refine[70], with the coordinates from ISOLDE used as a reference model to prevent deterioration of model geometry. Subsequent rounds of iterative model building were performed using COOT[71] and ISOLDE in consultation with the validation statistics provided by MolProbity[72], in each case using the coordinates from ISOLDE as a reference model in phenix.refine. The final model had one molecule of CEACAM6 and CcCoV-KY43 RBD per asymmetric unit, with 95.6% of residues in the favoured area of the Ramachandran plot and no outliers, and it included 10 N-acetylglucosamine residues (9 attached to CEACAM6 and 1 to CcCoV-KY43 RBD). The structure of human CEACAM6 in complex with CcCoV-2B was solved by molecular replacement using PHASER[66] with models of the CEACAM6 ectodomain and CcCoV-2B RBD (residues 496–625) that were generated with ColabFold[73] using the crystal structure of CEACAM6 in complex with the CcCoV-KY43 RBD as a template. The structure was refined as described above. The final model had one molecule of CEACAM6 and CcCoV-2B RBD per asymmetric unit, with 96.2% of residues in the favoured area of the Ramachandran plot and no outliers, and it included 12 N-acetylglucosamine residues (9 attached to CEACAM6 and 3 to CcCoV-2B RBD).

## Biolayer interferometry
To compare affinities of human CEACAM proteins for CcCoV-KY43 and CcCoV-2B, the recombinant viral RBDs were immobilized at 20 μg ml$^{-1}$ to Ni-NTA biosensors (Sartorius) pre-hydrated in 10 mM HEPES pH 7.5, 150 mM NaCl and 0.02% Tween-20. Biosensors were incubated with a a 2-fold serial dilution of CEACAM proteins from 600 nM to 37.5 nM (association) then buffer alone (dissociation). Between incubations, biosensors were regenerated by thrice incubating for 5 s with 10 mM glycine pH 1.7 then 5 s in buffer to neutralize, and biosensors were recharged using 10 mM NiCl$_2$ before fresh immobilization of the RBD. Two independent experiments were performed at 30 °C using a Octect Red (ForteBio). Results were analysed using data analysis software of the instrument (Sartorius).

## ITC
ITC experiments were performed using a MicroCal PEAQ-ITC automated calorimeter (Malvern Panalytical) at 25 °C. CcCoV RBDs were titrated into human CEACAM proteins using 13 × 3 μl or 19 × 2 μl injections. Results were analysed using the analysis software of the instrument and fitted using a one-site binding model. The experiments were performed twice (CEACAM5) or thrice (CEACAM6) independently, and data from each experiment are shown in Supplementary Table 4.

## *C. cor* and human population distribution data in Kenya
GPS points from all the study areas in Kenya where bats were surveyed were uploaded onto QGIS (2025, https://www.qgis.org). The points were later grouped based on bats species as either 'other species of bats' or 'heart-nosed bat'. A shape file of the known extent of *C. cor* was added to ascertain the accuracy of the sampling. Sampling points containing *C. cor* were later classified based on the cave type as either natural cave, tree cave or mine cave. In two instances, bats were found in human habitations.

## Phylogenetic reconstruction
Alphacoronavirus spike gene sequences of the 40 selected strains, in addition to those representing the local phylogeny of CcCoV-KY43,

were aligned using MAFFT (v.7.526)[49] and molecular clock calibration was performed in BEAST (v.1.10.5)[74]. Bayesian analysis was parameterized using the SRD06 codon position model (which implements a HKY$_{112}$ partition)[75,76], a relaxed clock defined by an underlying log-normal distribution to model variability in rates across branches[77], and a non-parametric skygrid prior with 50 grid points distributed over 65 years[78]. An uninformative continuous-time Markov chain rate was set as reference prior on the clock rate, whereas other priors were left at their default values. Moreover, to investigate the potential acquisition (or loss) of CEACAM6 usage along the full evolutionary history of alphacoronavirus, discrete CEACAM6 usage was modelled as a binary trait under an asymmetric continuous-time Markov chain process with Markov jump counting in BEAST[79]. CEACAM6 trait evolution was jointly inferred in BEAST (v.1.10.5) with time-calibrated phylogenies estimated under a SRD06 codon-partitioned substitution model, an uncorrelated log-normal relaxed molecular clock and a skygrid coalescent prior. Markov jumps were used to obtain posterior distributions for the number and direction of CEACAM6 transitions (gains and losses) and to reconstruct the most probable trait state at internal nodes. All Markov chain Monte Carlo analyses were run for 200 million iterations, with samples collected every 20,000 steps. Convergence and mixing of the chain were evaluated by ensuring that all posterior parameters returned effective sample size values greater than 200. Highest independent posterior subtree reconstructed[80] trees were constructed in TreeAnnotator by discarding the initial 10% of the chain.

## Additional computational analysis
Molecular graphics were generated using PyMOL (Schrödinger) and ChimeraX[81]. Structure interfaces were analysed using PDBePISA[82]. For analysis of relative changes in pseudotype entry, significance was quantified using the Python3 module statsmodels (https://www.statsmodels.org/) with technical replicates averaged and treatment effects quantified as log$_{10}$-transformed fold changes relative to the control (untreated) sample. Significance was determined using a one-sample *t*-test against a theoretical mean of 0, with standard errors calculated using a pooled variance estimate across all groups to ensure robust variance estimation given the small sample sizes. *P* values were adjusted for multiple comparisons using the Holm–Bonferroni correction. GraphPad Prism 9 was used to generate all the other figures and to perform all other statistical analyses. Non-significant difference is shown when *P* > 0.032.

## Biosecurity statement
Following the identification of CEACAM6 as a receptor in early 2024, we recognized the potential health implications of our findings and the need to notify and improve preparedness and biosecurity in Kenya. Contacting B. A. at the National Museum of Kenya, who first identified CcCoV-KY43 and is a co-author on this study, we let his team know of the potential risk, especially concerning sampling of this bat species. In consultation with biosecurity colleagues in the United Kingdom, we also made the decision to not attempt rescuing or propagating the virus in our laboratories. As such, all work detailed in this paper has been done without live virus work in UK laboratories.

## Reporting summary
Further information on research design is available in the Nature Portfolio Reporting Summary linked to this article.

## Data availability
All sequence data analysed were sourced from public databases, as described in the Methods. These, along with the raw, when necessary, analysed data presented in this article, have been deposited into Zenodo (https://doi.org/10.5281/zenodo.17951484)[58] and made freely accessible. Atomic coordinates, structure factors and protein

sequences have been deposited into the Protein Data Bank (PDB) with the following accession numbers: 9RCS for human CEACAM6 in complex with the CcCoV-KY43 RBD and 9RCU for human CEACAM6 in complex with the CcCoV-2B RBD. In addition to the structures determined in this study, the following previously experimentally determined 3D structures were included: CEACAM6 homodimer (PDB: 4Y8A); CEACAM6–CEACAM8 heterodimer (PDB: 4YIQ); BtCoV-MOW15-22 (PDB: 9C6O); HCoV-SARS2 (PDB: 6M0J); HCoV-SARS1 (PDB: 2AJF4); NeoCoV (PDB: 7WPO); BtCoV-HKU5 (PDB: 9D32); BtCoV-KY72 (PDB: 8K4U); BtCoV-PRD-0038 (PDB: 8U0T); HCoV-NL63 (PDB: 3KBH); HCoV-229E (PDB: 6ATK); CCoV strain HuPn2018 (PDB: 7U0L); PRCV (PDB: 4F5C); PDCoV (PDB: 7VPQ); and MHV spike trimer in complex with mouse CEACAM1 (PDB: 6VSJ). Human population data were obtained from the Kenya National Bureau of Statistics (2019, https://www.knbs.or.ke/download/2019-kenya-population-and-housing-census-volume-iii-distribution-of-population-byage-sex-and-administrative-units/) based on the 2019 national census. Viral sequences used for this study were downloaded (https://www.bv-brc.org/view/Virus/10239). Publicly available single-cell RNA-sequencing data from the Human Protein Atlas were downloaded (https://www.proteinatlas.org/humanproteome/single+cell/single+cell+type/data#datasets). Lung Cell Atlas (HLCA v1.0) dataset for normal human lungs samples was downloaded https://data.humancellatlas.org/hca-bio-networks/lung/atlases/lung-v1-0).

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

**Acknowledgements** We thank T. Peacock, N. Thakur and J. Newman at The Pirbright Institute for provision of the TMPRSS2 plasmids, other protease constructs and the ACE2 expression library; S. Uyoga at the KEMRI–Wellcome Trust Research Program for the blood donor samples used in the serological assays; all the coronavirus research teams that deposited sequence information to GenBank that were selected by our greedy-algorithm-based approach; staff at the Diamond Light Source for beamtime (proposal mx36838) and staff of beamlines I04 and I24 for assistance with crystal testing and data collection. For the purpose of open access, the authors have applied a Creative Commons Attribution (CC BY) licence to any Author Accepted Manuscript version arising from this submission. D.B., A.D.N., S.C.G. and G.G. were supported by a BBSRC grant (BB/W006162/1). D.B. is supported by a BBSRC Institute Strategic Program Grant (BBS/E/PI/230002B) to The Pirbright Institute. D.B. and G.G. acknowledge the Pirbright Institute Flow Cytometry Facility (BBS/E/PI/23NB0003). A.D.N. is supported by a grant funded by the UK Department for Environment, Food and Rural Affairs (Defra, SE2947). The human receptor screening platform was supported by a MRC Partnership Grant (MR/X019705/1) awarded to D.B. and G.J.W. J.N., D.L. and B.A. are funded by Wellcome Trust grants 226141/Z/22/Z and 226130/Z/22/Z. The funders had no role in study design, data collection and analysis, decision to publish, or preparation of the article.

**Author contributions** G.G., A.D.N., S.C.G. and D.B. conceptualized the study and designed the methodology. G.G., A.J.R., B.A.K., A.K.D., F.M.M.B. and B.A. performed the investigation. G.G., D.L. and A.J.R. conducted formal analyses. G.G., A.K.D. and F.M.M.B. performed validation. G.G., A.D.N., D.L. and M.O. performed visualization. G.G., S.C.G. and D.B. wrote the original draft of the manuscript. All authors reviewed and edited the manuscript. S.C.G. and D.B. supervised the study. G.J.W., J.N., A.D.N., S.C.G. and D.B. acquired funding.

**Competing interests** The authors declare no competing interests.

**Additional information**
**Correspondence and requests for materials** should be addressed to Stephen C. Graham or Dalan Bailey.

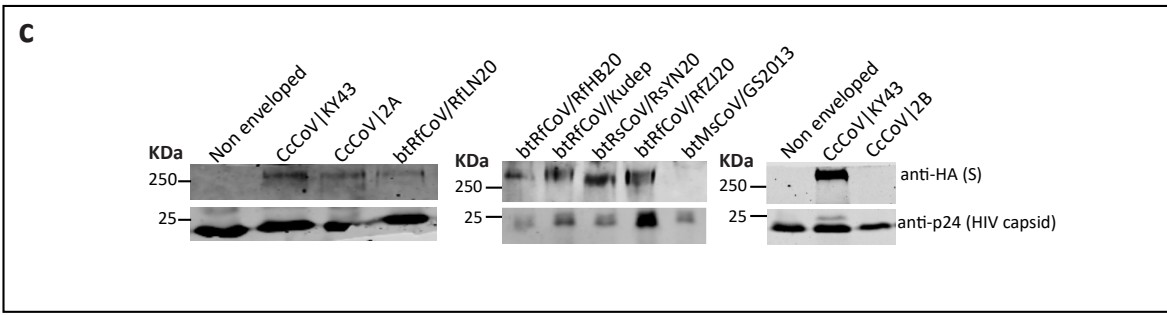

**Extended Data Fig. 1 | Lentiviral pseudotyping of the alphaCoV library.**
(**a**) Pseudoparticles were purified from the cellular supernatant using a 20% sucrose gradient, and centrifuged at 33,000 g for 2 h, at 4 °C. Pellets were resuspended in DMEM and lysed for immunoblot analysis. Spikes were detected using an antibody recognizing the attached HA tag, while efficiency of pseudoparticle purification was assessed using an antibody against the p24 capsid protein of HIV. (**b**) Pseudoparticles were also pre-treated with TDPK-trypsin for one hour at room temperature, before purification and immunoblot analysis, as described in (a). (**c**) Pseudoviruses of CcCoV-related viruses were purified from cellular supernatant using 20% sucrose, as previously described, and the incorporation of S on the particles assessed by immunoblot against an HA-tag fused to the S. To validate correct purification, expression of the lentiviral structural protein p24 was also assayed.

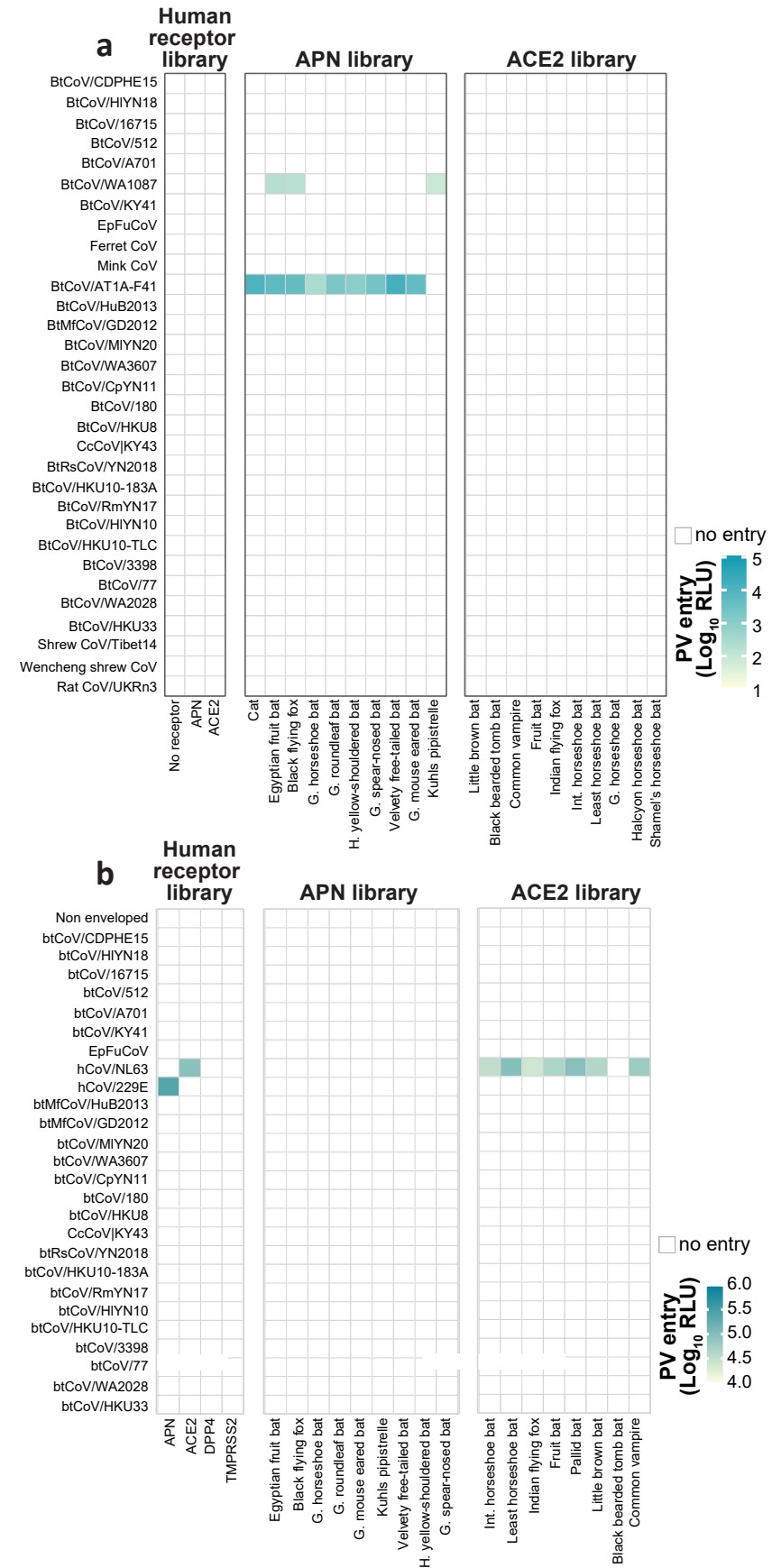

**Extended Data Fig. 2** | See next page for caption.

**Extended Data Fig. 2 | APN and ACE2 receptor screening in presence of human TMPRSS2, or preincubation with trypsin.** (**a**) HEK293T overexpressing libraries of APN or ACE2 from various mammals, as indicated, plus human TMPRSS2 were infected with the alphaCoV library of S pseudotypes. Expression of hTMPRSS2 did not impact the overall pattern of receptor usage. Initial screening was performed in technical triplicate, and positive results were validated in two additional independent experiments. (**b**) Treatment of pseudotyped alphaCoV S with trypsin does not affect usage of APN and ACE2 receptors. Pseudoviruses were incubated with 250 µg/mL of TDPK-trypsin for one hour at room temperature, before infection of HEK293T overexpressing selected APN or ACE2. Cleavage of S did not impact receptor usage, or host range (through comparison to results shown in Fig. 1b and Fig. 1c). Experiments were performed once in technical triplicate.

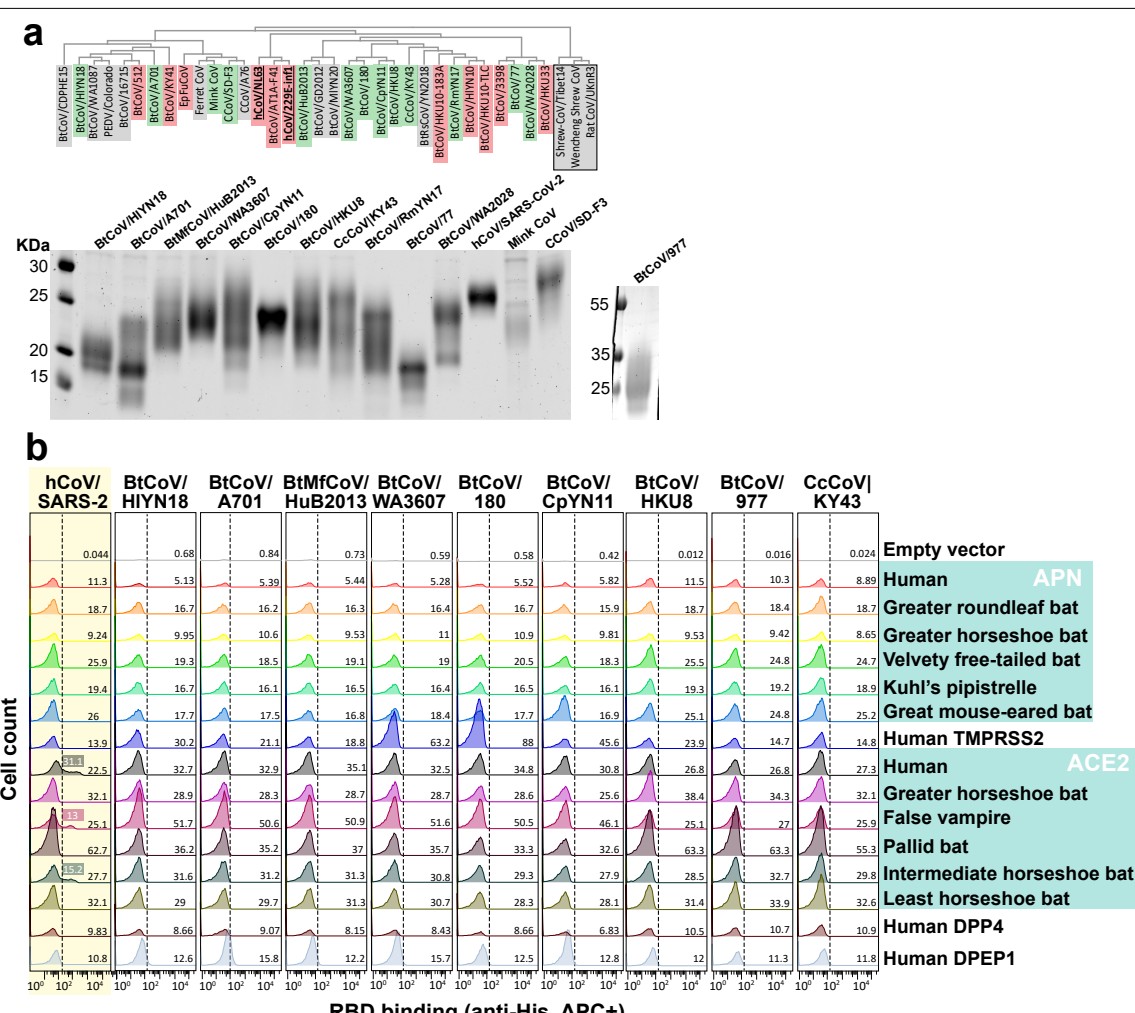

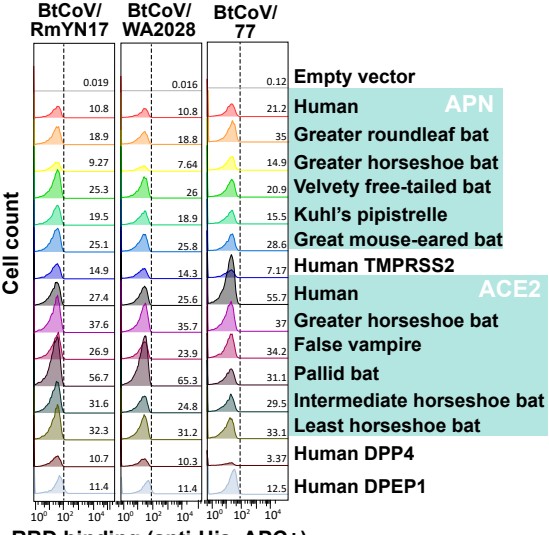

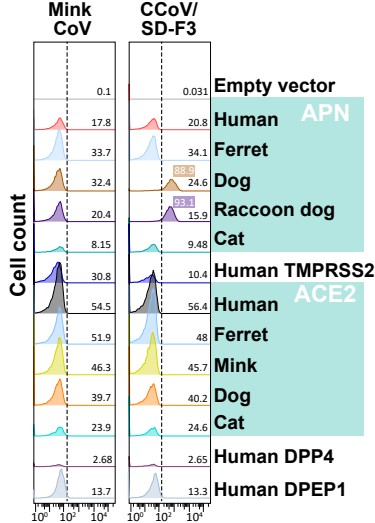

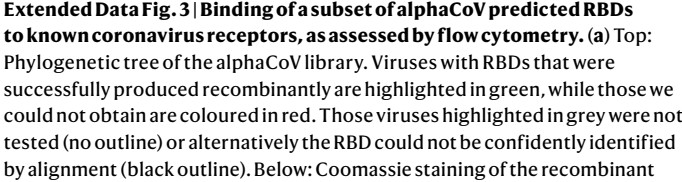

**Extended Data Fig. 3 | Binding of a subset of alphaCoV predicted RBDs to known coronavirus receptors, as assessed by flow cytometry. (a)** Top: Phylogenetic tree of the alphaCoV library. Viruses with RBDs that were successfully produced recombinantly are highlighted in green, while those we could not obtain are coloured in red. Those viruses highlighted in grey were not tested (no outline) or alternatively the RBD could not be confidently identified by alignment (black outline). Below: Coomassie staining of the recombinant RBDs used in the binding assay. **(b)** Binding of viral RBDs to relevant APN, ACE2, TMPRSS2, DPP4 and DPEP1 receptors, as determined by antibody staining and flow cytometry. The numbers by each sample indicate the percentage of PE+ positive cells among the singlets (receptor + population). The numbers in coloured boxes show the percentage of APC+ cells (viral RBD + population) in this PE+ population. At least 25,000 events were acquired per sample.

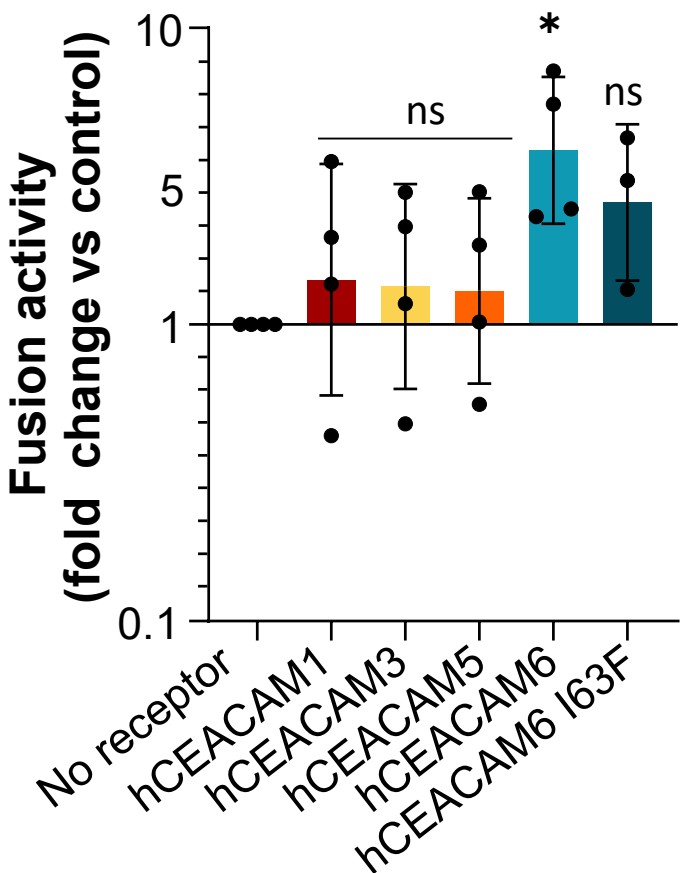

**Extended Data Fig. 4 | Human CEACAM6 supports cell-cell fusion in the presence of CcCoV-KY43 S.** A fusion assay, based on a split firefly luciferase and GFP, was used to validate usage of human CEACAM6 as a cognate receptor for CcCoV-KY43 S. The average fold change from at least three independent experiments, each performed in triplicate, is shown along with the standard deviation. Statistical significance was examined using a one-sample t test of $\log_{10}$ fold change against a theoretical mean of 0, with standard errors calculated using a pooled variance estimate across all groups to ensure robust variance estimation given the small sample sizes. $P$-values were adjusted for multiple comparisons using the Holm-Bonferroni correction (*$p$-value = 0.01, ns: non-significant difference).

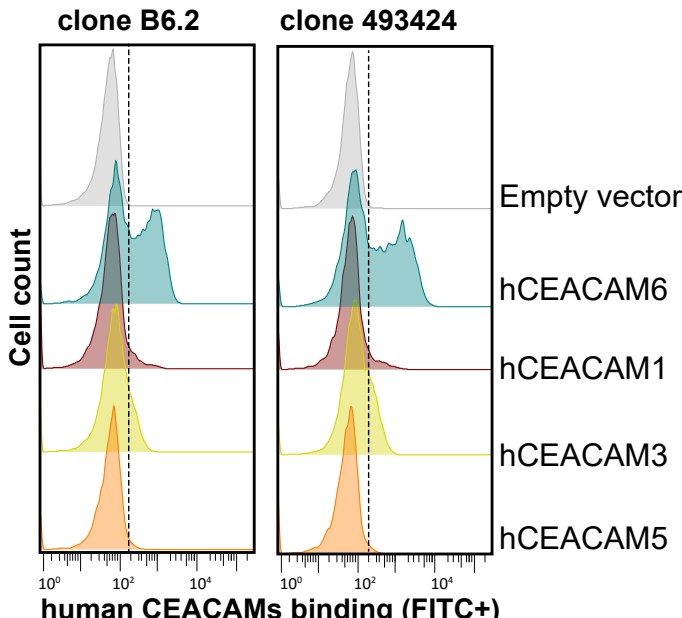

**Extended Data Fig. 5 | Monoclonal antibodies B6.2 and 493424 are specific to human CEACAM6.** HEK293T cells transfected with plasmids encoding the indicated human CEACAMs were stained using the anti-CEACAM6 commercial monoclonal antibodies, B6.2 and 493424. FITC-conjugated anti-mouse was used as secondary antibody to detect binding of the monoclonals. Surface expression was then quantified using a flow cytometer.

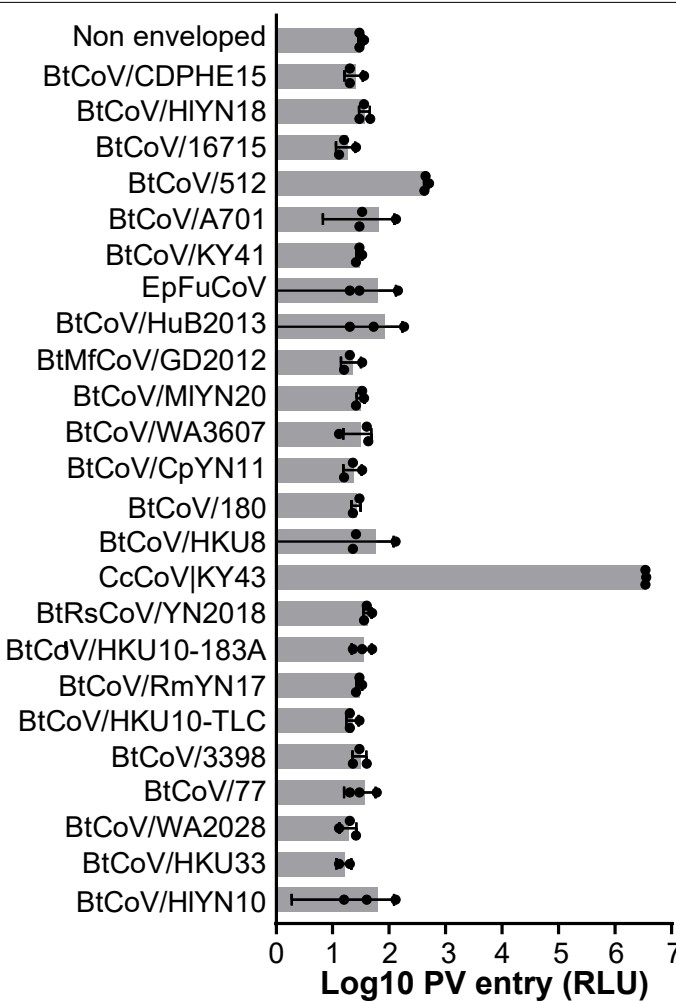

**Extended Data Fig. 6 | Human CEACAM6 usage within the alphaCoV pseudotype S library.** Pseudovirus (PV) entry assays showed that only CcCoV-KY43 requires CEACAM6 for entry. Experiment was performed in technical triplicate, with the mean and SD relative light units (RLU) plotted.

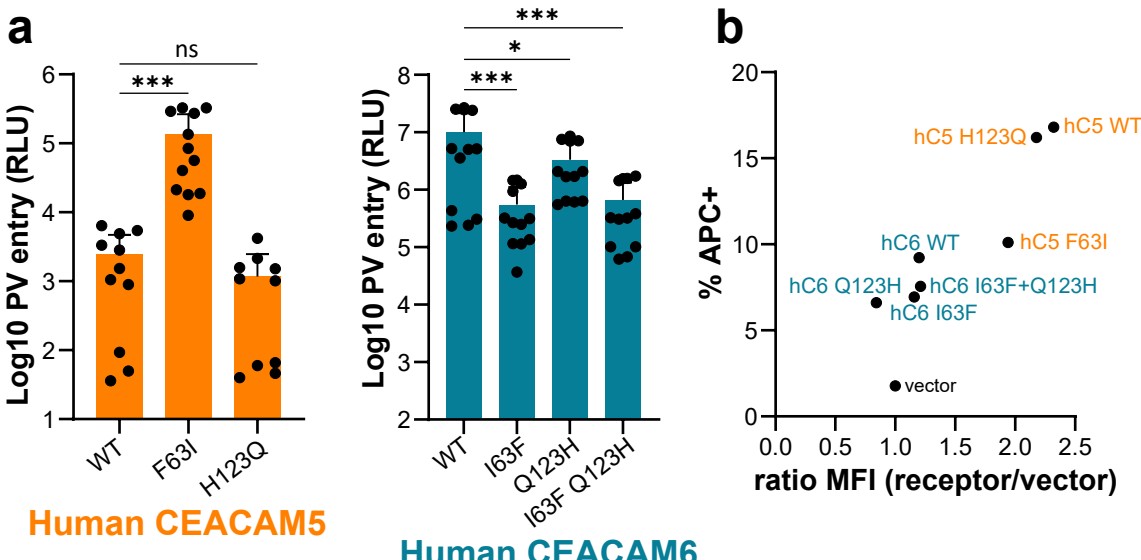

**Extended Data Fig. 7 | Additional amino-acid substitutions in human CEACAM5 and CEACAM6 confirm a major contribution of position 63 to receptor specificity.** (**a**) Entry assays using CcCoV-KY43 S pseudotypes with HEK293T cells transiently transfected with human CEACAM5 and CEACAM6, or mutated forms of these proteins. The substitutions I63F and Q123H confirms that position 63 is a major determinant of CEACAM usage for KY43. (**b**) Flow cytometry data shows comparable expression of human CEACAM5 and CEACAM6, and their derived mutants, in HEK293T cells.

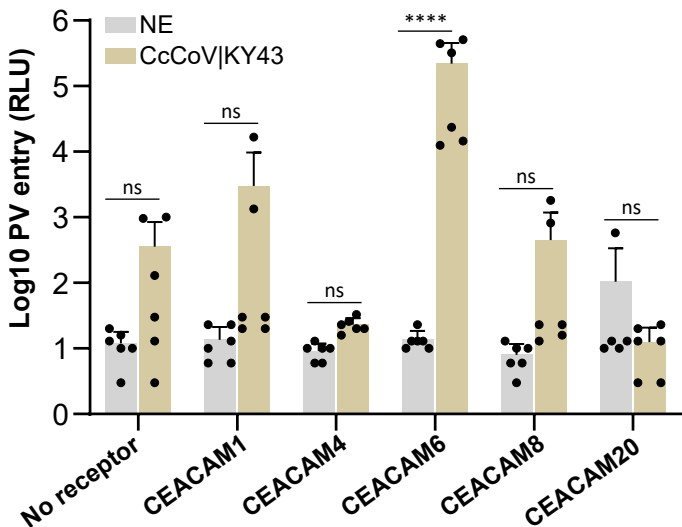

**Extended Data Fig. 8 | Entry of pseudotyped CcCoV|KY43 S using a wider library of human CEACAMs.** HEK293T cells were transiently transfected to individually overexpress human CEACAMs, as indicated. The experiment was performed with two independent replicates, both containing technical triplicates. The technical triplicates of two biological replicates, with SD, are shown as relative light units (RLU), compared to non-enveloped (NE) controls. ns: non-significant, ****$p$ = <0.001 statistical analysis using two-way ANOVA.

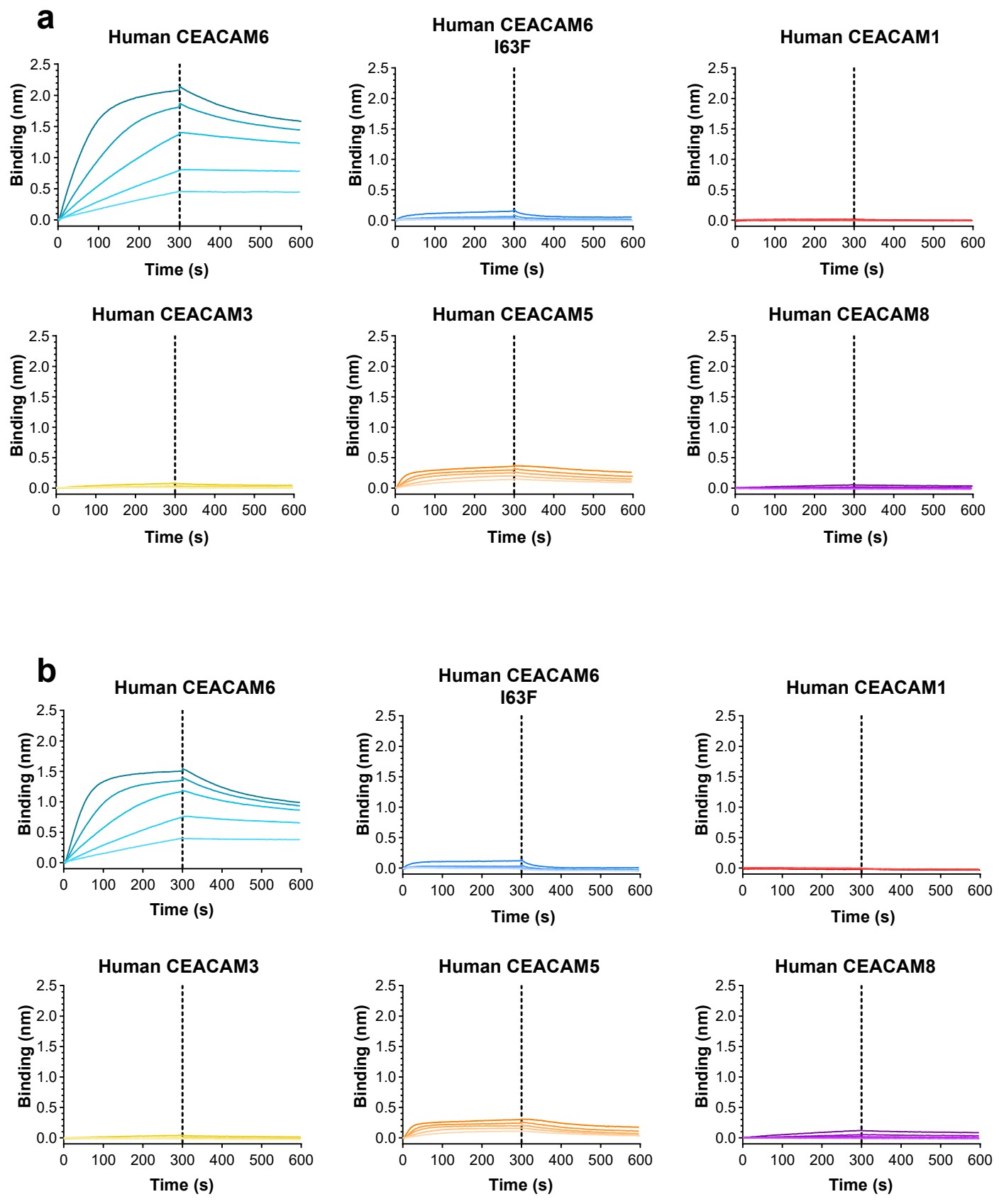

**Extended Data Fig. 9 | Comparison of RBD binding to different human CEACAMs using bio-layer interferometry (BLI).** Ni-NTA sensors were loaded with His-tagged CcCoV|KY43 RBD (**a**) or CcCoV-2B (**b**) and then incubated with different human CEACAMs in a two-fold concentration gradient from 600 nM (dark colours) to 37.5 nM (light colours). BLI responses following subtraction of a reference sensor incubated with buffer are shown. The end of the association phase is marked (dashed vertical line). Human CEACAM6 binds strongly to CcCoV RBDs, and this binding is lost upon introduction of an I63F substitution. CcCoV RBDs bind weakly to human CEACAM5 and displays negligible binding to CEACAM1, CEACAM3 and CEACAM8. Data are representative of two independent experiments.

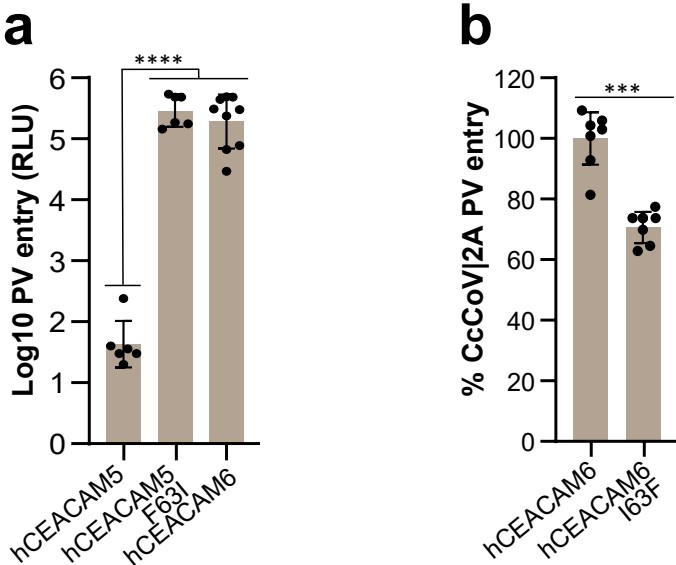

**Extended Data Fig. 10 | Usage of human CEACAM5, CEACAM6, and individual point mutants, by pseudotyped CcCoV-2A S.** (**a**) Pseudovirus entry of CcCoV-2A S in HEK293T cells expressing the indicated CEACAM protein. The F63I substitution in CEACAM5 increases CcCoV-2A S-mediated entry to the level seen with CEACAM6. Values are presented as relative light units (RLU). Assays were performed twice in technical triplicates with mean RLU and SD shown. (**b**) CcCoV-2A S entry into cells expressing CEACAM6 substitution I63F is reduced relative to wild type human CEACAM6 (hCEACAM6). Assays were performed twice in technical triplicates in triplicate with percentage PV entry and SD shown. Statistical analysis was performed using paired one-way ANOVA (****$p$-value < 0.0001) and paired t-test (***$p$-value = 0.0002), respectively.

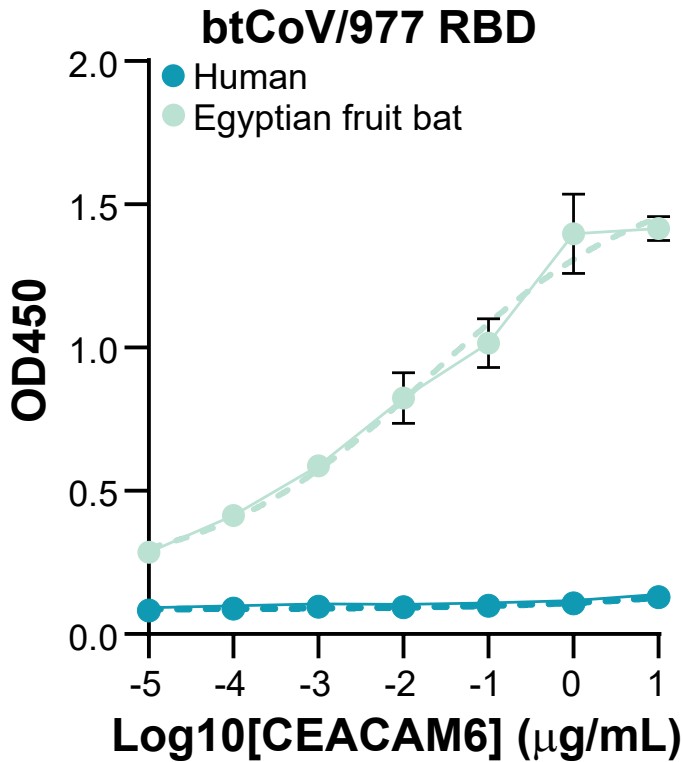

**Extended Data Fig. 11 | Binding of recombinant btCoV-977 to Egyptian fruit bat, but not human, CEACAM6.** ELISA using the recombinant RBD of btCoV-977 to assess binding to soluble Egyptian fruit bat CEACAM6-like, or human CEACAM6, proteins. The mean OD from a representative experiment, performed in technical quadruplicates, is shown together with the SD.

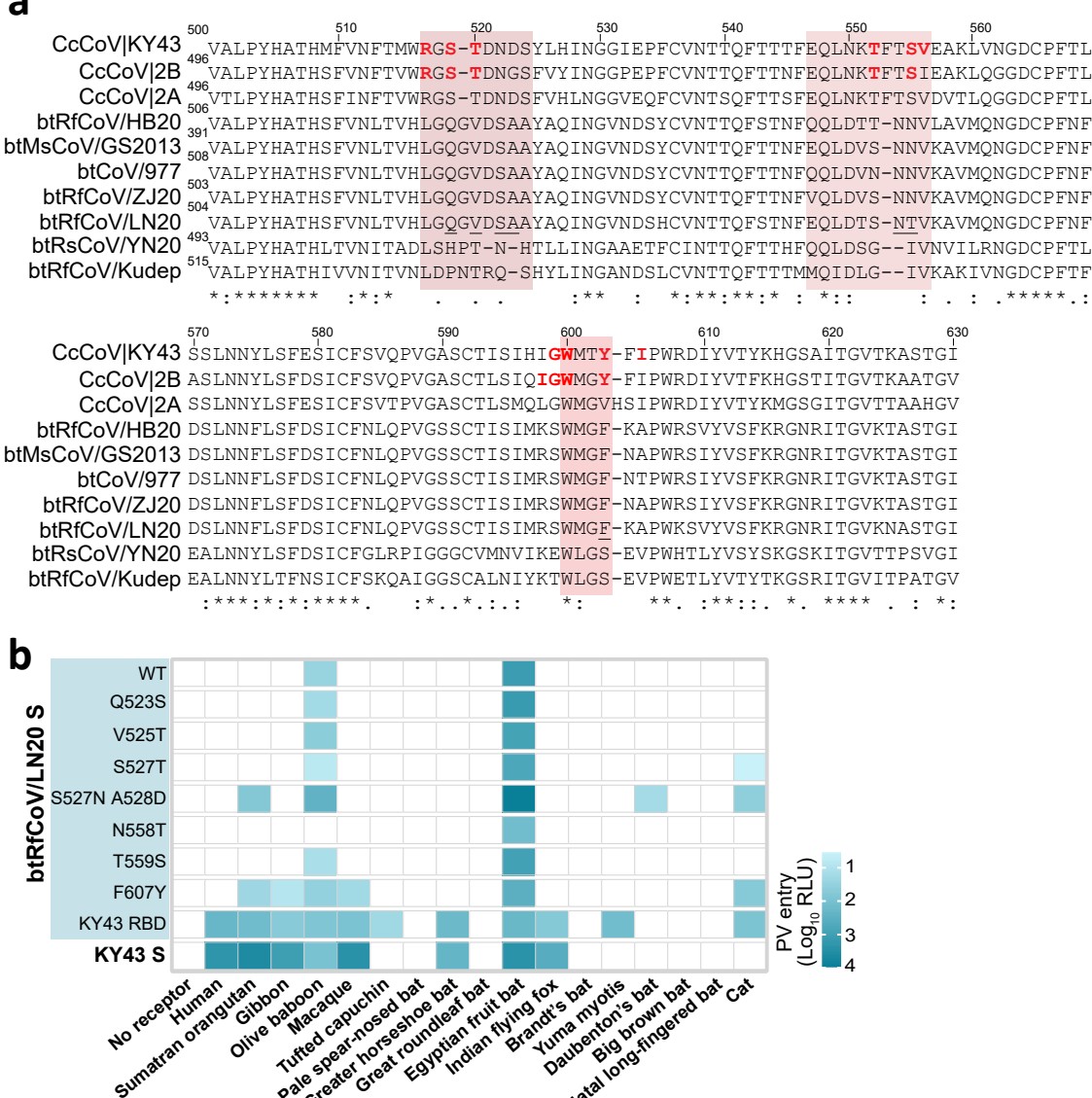

**Extended Data Fig. 12 | Sequence determinants of human CEACAM6 usage.**
(**a**) Alignment of RBD sequences from the CEACAM6-using alphaCoV RBDs
included in this study. Loops that interact with human CEACAM6 are highlighted
in red. Residues that interact with human CEACAM6 in the CcCoV-KY43 and
CcCoV-2B complex structure are shown in **bold red**. Residues of btCoV-LN20
that were mutated to probe the sequence determinants of human CEACAM6
usage (see below) are underlined in grey. (**b**) Host range of btCoV-LN20,

a CEACAM6 user, cannot be defined by point substitutions in its RBD. Entry
assays using pseudotyped btCoV-LN20 S show that single mutations in the
RBD, making it more like CcCoV-KY43 S, cannot recapitulate the broad host
range of CcCoV-KY43, including usage of human CEACAM6. The background-
corrected average of three independent experiments, each performed in
triplicate, is shown.

# Reporting Summary

## Statistics

For all statistical analyses, confirm that the following items are present in the figure legend, table legend, main text, or Methods section.

| n/a | Confirmed | |
|---|---|---|
| ☐ | ☒ | The exact sample size (*n*) for each experimental group/condition, given as a discrete number and unit of measurement |
| ☐ | ☒ | A statement on whether measurements were taken from distinct samples or whether the same sample was measured repeatedly |
| ☐ | ☒ | The statistical test(s) used AND whether they are one- or two-sided *Only common tests should be described solely by name; describe more complex techniques in the Methods section.* |
| ☐ | ☒ | A description of all covariates tested |
| ☐ | ☒ | A description of any assumptions or corrections, such as tests of normality and adjustment for multiple comparisons |
| ☐ | ☒ | A full description of the statistical parameters including central tendency (e.g. means) or other basic estimates (e.g. regression coefficient) AND variation (e.g. standard deviation) or associated estimates of uncertainty (e.g. confidence intervals) |
| ☐ | ☒ | For null hypothesis testing, the test statistic (e.g. *F*, *t*, *r*) with confidence intervals, effect sizes, degrees of freedom and *P* value noted *Give P values as exact values whenever suitable.* |
| ☐ | ☒ | For Bayesian analysis, information on the choice of priors and Markov chain Monte Carlo settings |
| ☒ | ☐ | For hierarchical and complex designs, identification of the appropriate level for tests and full reporting of outcomes |
| ☐ | ☒ | Estimates of effect sizes (e.g. Cohen's *d*, Pearson's *r*), indicating how they were calculated |

*Our web collection on statistics for biologists contains articles on many of the points above.*

## Software and code

Policy information about availability of computer code

| Data collection | Viral sequences used for this study were obtained from https://www.bv-brc.org/view/Virus/10239.<br><br>Diffraction data were recorded from single crystals on Diamond beamline I24 at 20 keV using an Eiger2 9M detector (CcCoV\|KY43 complex) or on beamline I04 at 13 keV using an Eiger2 X 16M detector (CcCoV\|2B complex).<br><br>Human population data was obtained from the Kenya National Bureau of Statistics (KNBS, 2019, https://www.knbs.or.ke/download/2019-kenya-population-and-housing-census-volume-iii-distribution-of-population-byage-sex-and-administrative-units/) based on the 2019 national census. |
|---|---|
| Data analysis | R 4.4.1 (ttps://www.r-project.org/) packages were downloaded from https://cran.r-project.org/web/packages/available_packages_by_name.html.<br><br>For protein structure visualization, PyMol version 2.4.0a0 (https://pymol.org/) or ChimeraX were used. Structural models were solved by molecular replacement using PHASER, and manually improved using ISOLDE and COOT software.<br>Structure interfaces were analysed using PDBePISA.<br><br>Sequence alignments were obtained using MAFFT 7.526. Molecular clock calibration was performed in BEAST 1.10.5. Highest Independent Posterior Subtree Reconstructed (HIPSTR)81 trees were constructed in TreeAnnotator.<br><br>For the design of primers used for site-directed mutagenesis, the Agilent online tool was used: https://www.agilent.com/store/primerDesignProgram.jsp |

For the design of shRNA used for knock-down of human CEACAM6, the web portal of the Genetic Perturbation Consortium (Broad Institute, https://portals.broadinstitute.org/gpp/public/seq/search) was used.

Publicly available single cell RNA-sequencing (scRNA-seq) data from the Human Protein Atlas were downloaded from https://www.proteinatlas.org/humanproteome/single+cell/single+cell+type/data#datasets.
Lung Cell Atlas (HLCA v1.0) dataset for normal human lungs samples from (https://data.humancellatlas.org/hca-bio-networks/lung/atlases/lung-v1-0). Processed cell-type annotations and expression matrices were loaded into Seurat (v.4.3.3.).

For data and statistical analysis, GraphPad Prism9 was used. For analysis of relative changes in pseudotype entry, significance was quantified using the Python3 module statsmodels (https://www.statsmodels.org/).

For data visualization, Adobe Illustrator was used.

For manuscripts utilizing custom algorithms or software that are central to the research but not yet described in published literature, software must be made available to editors and reviewers. We strongly encourage code deposition in a community repository (e.g. GitHub). See the Nature Portfolio guidelines for submitting code & software for further information.

# Data

Policy information about availability of data

All manuscripts must include a data availability statement. This statement should provide the following information, where applicable:
- Accession codes, unique identifiers, or web links for publicly available datasets
- A description of any restrictions on data availability
- For clinical datasets or third party data, please ensure that the statement adheres to our policy

Atomic coordinates, structure factors and protein sequences have been deposited to the Protein DataBank (PDB). Accession numbers are PDB: 9RCS and 9RCU for human CEACAM6 in complex with CcCoV|KY43 RBD and CcCoV|2B RBD, respectively. In addition to the structures determined in this study, the following previously experimentally determined 3D structures have been included in this article: CEACAM6 homodimer (PDB:4Y8A), CEACAM6-CEACAM8 heterodimer (PDB:4YIQ), btCoV/MOW15-22 (PDB: 9C6O), hCoV/SARS2, PDB: 6M0J; hCoV/SARS1, PDB: 2AJF4 NeoCoV, PDB: 7WPO; btCoV/HKU5, PDB: 9D32; btCoV/KY72, PDB: 8K4U; btCoV/PRD-0038, PDB: 8U0T, hCoV/NL63 (PDB: 3KBH), hCoV/229E (PDB: 6ATK), canine CoV (CCoV strain HuPn2018, PDB: 7U0L) and porcine coronavirus PRCV (PDB: 4F5C), PDCoV (PDB: 7VPQ), MHV S trimer in complex with murine CEACAM1 (PDB:6VSJ). All sequence data analysed were sourced from public databases, as described in Materials and Methods. These, along with the raw, when necessary, analysed data presented in this article, have been deposited in Zenodo and made freely accessible at the following link: 10.5281/zenodo.17951484.
Human population data was obtained from the Kenya National Bureau of Statistics (KNBS, 2019, https://www.knbs.or.ke/download/2019-kenya-population-and-housing-census-volume-iii-distribution-of-population-byage-sex-and-administrative-units/) based on the 2019 national census. Viral sequences used for this study were obtained from https://www.bv-brc.org/view/Virus/10239. Publicly available single cell RNA-sequencing (scRNA-seq) data from the Human Protein Atlas were downloaded from https://www.proteinatlas.org/humanproteome/single+cell/single+cell+type/data#datasets.
Lung Cell Atlas (HLCA v1.0) dataset for normal human lungs samples from (https://data.humancellatlas.org/hca-bio-networks/lung/atlases/lung-v1-0).
The raw human data shown in the manuscript are subject to controlled access because they are the subject of ongoing work and will be made available on request to the corresponding author and approval by the Data Governance Committee at the KEMRI-Wellcome Trust Research Programme. De-identified data has been published on the Harvard dataverse server  https://doi.org/10.7910/DVN/XSGOOF.

# Research involving human participants, their data, or biological material

Policy information about studies with human participants or human data. See also policy information about sex, gender (identity/presentation), and sexual orientation and race, ethnicity and racism.

| | |
|---|---|
| Reporting on sex and gender | yes |
| Reporting on race, ethnicity, or other socially relevant groupings | yes |
| Population characteristics | The study population is made up of blood donors. The Kenya National Blood Transfusion Services guidelines define eligible donors as individuals aged 16-65 years, weighing >=50kg, with hemoglobin of 12.5g/dl, a normal blood pressure (systolic 120-129 mmHg and diastolic BP of 80-89  mmHg), a pulse rate of 60-100 beats per minute and without any history of illness in the past 6 months. |
| Recruitment | The study population was recruited as anonymized residual samples from consecutive donor units submitted to the regional centres for transfusion compatibility-testing and infection screening. Since blood donors are restricted to those 16-65 years old, they are not representative of a population sample of all ages which may introduce a selection bias. In addition, the eligibility criteria for blood donation may select for more healthy members of the population which may lead to underestimation of KY43 antibody prevalence |
| Ethics oversight | The Scientific and Ethics Review Unit of the Kenya Medical Research Institute gave ethical approval |

Note that full information on the approval of the study protocol must also be provided in the manuscript.

# Field-specific reporting

Please select the one below that is the best fit for your research. If you are not sure, read the appropriate sections before making your selection.

☒ Life sciences ☐ Behavioural & social sciences ☐ Ecological, evolutionary & environmental sciences

For a reference copy of the document with all sections, see nature.com/documents/nr-reporting-summary-flat.pdf

# Life sciences study design

All studies must disclose on these points even when the disclosure is negative.

| | |
|---|---|
| Sample size | Among the total number of viral S protein available, a subset of 40 was used for this study. This number is based on the screening and funding capabilities associated to the study. The selected 40-sequence S panel (~1.5% of the full dataset) captured 53.4% of the total phylogenetic diversity, substantially exceeding size-matched random panels (13.7±3.2%; 10,000 permutations; ~3.9-fold enrichment; empirical p<0.0001). Once we identified the receptor of an alphacoronavirus S, we focused on validating its interaction. For analysis of human sera, 368 samples were analysed, based on the availability and the donor recruited. |
| Data exclusions | No data was excluded |
| Replication | Initial receptor screening was performed in technical triplicates, with two independent set of pseudoparticles (biological duplicates). Positive results were validated with at least 3 biological replicates, in technical triplicates. |
| Randomization | Human samples are convenient residual blood donor samples from the regional centres of transfusion, in the regions where CcCoV|KY43 was identified. Since blood donors are restricted to those 16-65 years old, they are not representative of a population sample of all ages which may introduce a selection bias. This is based on access to population. |
| Blinding | The population was recruited as anonymized residual samples. De-identified data has been published on the Harvard dataverse server https://doi.org/10.7910/DVN/XSGOOF. |

# Reporting for specific materials, systems and methods

We require information from authors about some types of materials, experimental systems and methods used in many studies. Here, indicate whether each material, system or method listed is relevant to your study. If you are not sure if a list item applies to your research, read the appropriate section before selecting a response.

## Materials & experimental systems

| n/a | Involved in the study |
|---|---|
| ☐ | ☒ Antibodies |
| ☐ | ☒ Eukaryotic cell lines |
| ☒ | ☐ Palaeontology and archaeology |
| ☒ | ☐ Animals and other organisms |
| ☒ | ☐ Clinical data |
| ☒ | ☐ Dual use research of concern |
| ☒ | ☐ Plants |

## Methods

| n/a | Involved in the study |
|---|---|
| ☒ | ☐ ChIP-seq |
| ☐ | ☒ Flow cytometry |
| ☒ | ☐ MRI-based neuroimaging |

# Antibodies

| | |
|---|---|
| Antibodies used | V5 Tag Monoclonal Antibody, PE conjugated (Invitrogen, clone TCM5)<br>FLAG Tag (DYKDDDDK) Antibody, APC conjugated (Miltenyi Biotec, clone REA216)<br>His tag antibody, APC conjugated (Miltenyi Biotec, clone GG11-8F3.5.1)<br>antibodies against human CEACAM6 (ThermoFisher, clones B6.2 and 439424)<br>HIV capsid p24 (Abcam, clone 5)<br>HA tag, unconjugated (Cell Signaling Technology, clone 6E2)<br>HA tag, PE conjugated (Miltenyi Biotec, clone GG8-1F3.3.1)<br>Flag tag, PE conjugated (Biolegend, clone L5)<br>Goat anti-mouse IgG(H+L), FITC conjugated (Invitrogen, polyclonal)<br>Goat anti-mouse IgG (H+L) DyLight 680 (Invitrogen, ployclonal) |
| Validation | V5 Tag Monoclonal Antibody, PE conjugated (Invitrogen, clone TCM5) - validation for flow cytometry:<br>https://www.thermofisher.com/antibody/product/V5-Tag-Antibody-clone-TCM5-Recombinant-Monoclonal/740058M<br>HEK-293E cells were transiently transfected with V5-H3-His construct (pink histogram) or left untransfected (blue histogram). Cells were fixed and permeabilized using the Foxp3 / Transcription Factor Staining Buffer Set (Product # 00-5523-00) and then stained intracellularly with 0.25 μg of V5 Tag Recombinant Mouse Monoclonal Antibody (TCM5), (Product # 740058M), followed by Goat anti-Mouse IgG (H+L), Superclonal™ Recombinant Secondary Antibody, Alexa Fluor™ Plus 647 (Product # A55060, 1:500). Viable cells were used for analysis, as determined by LIVE/DEAD™ Fixable Violet Dead Cell Stain Kit (Product # L34955). The flow cytometry data |

was acquired using Attune™ NxT Flow Cytometer (Product # A29004).

FLAG Tag (DYKDDDDK) Antibody, APC conjugated (Miltenyi Biotec, clone REA216) - validation for flow cytometry:
https://www.miltenyibiotec.com/GB-en/products/dykddddk-antibody-reafinity-rea216.html#conjugate=vio-b515:size=100-tests-in-200-ul
293HEK cells stably transfected with FLAG®-tagged human Argonaute 1 protein were stained intracellularly with Anti-DYKDDDDK antibodies and analyzed by flow cytometry using the MACSQuant® Analyzer.

His tag antibody, APC conjugated (clone GG11-8F3.5.1, Miltenyi Biotec) - validation for flow cytometry:
https://www.miltenyibiotec.com/GB-en/products/his-antibody-gg11-8f3-5-1.html#conjugate=fitc:size=100-tests-in-200-ul
Surface staining follow protocol (https://www.miltenyibiotec.com/GB-en/applications/all-protocols/cell-surface-flow-cytometry-staining-protocol-pbs-edta-bsa-1-50.html)

Antibodies against human CEACAM6 (ThermoFisher, clones B6.2 and 439424) - validation for western blot:
Lysates of HEK293T transfected with a plasmid encoding for human CEACAM6 after 24 hours. Expression was analysed by sodium dodecyl sulfate polyacrylamide gel electrophoresis (SDS-PAGE). Separated proteins were transferred onto a 0.45 μm nitrocellulose membrane (Cytiva), blocked in PBS supplemented with 0.05% TWEEN-20 and 5% (w/v) unskimmed milk powder, and incubated with mouse monoclonals overnight at 4C. The following day, goat anti-mouse DyLight 680 (Invitrogen, 1:10000) was used to probe primary antibodies, and signals detected with an Odyssey DLx imaging system (LI-COR Biosciences). Control lysate non-transfected, or transfected with non-relevant proteins, were run in parallel to assess specificity.
references citing the use of the antibody: DOI: 10.14348/molcells.2020.2230

HIV capsid p24 (clone 5, Abcam) - validation for Western blot:
references citing the use of the antibody: DOI: 10.1016/j.isci.2022.105016, DOI: 10.1021/acs.biochem.3c00109,  DOI: 10.1186/s12964-024-01795-4

HA tag, unconjugated (Cell Signaling Technology, clone 6E2) - validation for Western blot:
https://www.cellsignal.com/products/primary-antibodies/ha-tag-6e2-mouse-monoclonal-antibody/2367
references citing the use of the antibody: DOI: 10.1038/s41586-025-09727-z, DOI: 10.1038/s41467-025-66120-0, DOI: 10.1038/s41586-025-09768-4

HA tag, PE conjugated (Miltenyi Biotec, clone GG8-1F3.3.1) - validation for flow cytometry:
https://www.miltenyibiotec.com/GB-en/products/ha-antibody-gg8-1f3-3-1.html#conjugate=apc:size=100-tests-in-200-ul
Surface staining follow protocol (https://www.miltenyibiotec.com/GB-en/applications/all-protocols/cell-surface-flow-cytometry-staining-protocol-pbs-edta-bsa-1-50.html)

Flag tag, PE conjugated (Biolegend, clone L5) - validation for flow cytometry:
https://www.biolegend.com/en-gb/products/purified-anti-dykddddk-tag-antibody-4905
references citing the use of the antibody: DOI: 10.1074/jbc.M109.071696, DOI: 10.1016/j.molcel.2018.09.029,  DOI: 10.1016/j.celrep.2022.111582, DOI: 10.1016/j.cell.2022.11.033

Goat anti-mouse IgG(H+L), FITC conjugated (Invitrogen, polyclonal) - validation for Western blot:
https://www.thermofisher.com/antibody/product/Goat-anti-Mouse-IgG-H-L-Cross-Adsorbed-Secondary-Antibody-Polyclonal/31541
references citing the use of the antibody: DOI: 10.1096/fj.201900260R, DOI: 10.1007/978-1-60761-063-2_1

Goat anti-mouse IgG (H+L) DyLight 680 (Invitrogen, ployclonal) - validation for Western blot:
https://www.thermofisher.com/antibody/product/Goat-anti-Mouse-IgG-H-L-Secondary-Antibody-Polyclonal/35518
references citing the use of the antibody: DOI: 10.1038/s44318-025-00616-9, DOI: 10.1371/journal.pbio.3003320, DOI: 10.1371/journal.ppat.1013410

# Eukaryotic cell lines

Policy information about cell lines and Sex and Gender in Research

| Cell line source(s) | HEK239T - ATCC (CRL-3216), THP-1 - ATCC (TIB-202), Calu-3 - ATCC (HTB-55), Caco-2 - ATCC (HTB-37), HeLa - ATCC (CCL-2), Expi293 - ThermoFisher, HuH7 were kindly provided by dr. Jane McKeating (University of Oxford), LCL cells (patient B cells immortalised with EBV) were kindly provided by dr. Claire Shannon Lowe (University of Birmingham). |
|---|---|
| Authentication | Cell lines were not authenticated |
| Mycoplasma contamination | All cell lines tested negative for mycoplasma contamination |
| Commonly misidentified lines (See ICLAC register) | *Name any commonly misidentified cell lines used in the study and provide a rationale for their use.* |

## Plants

Seed stocks          n/a

Novel plant genotypes   n/a

Authentication       n/a

## Flow Cytometry

### Plots

Confirm that:

☒ The axis labels state the marker and fluorochrome used (e.g. CD4-FITC).

☒ The axis scales are clearly visible. Include numbers along axes only for bottom left plot of group (a 'group' is an analysis of identical markers).

☒ All plots are contour plots with outliers or pseudocolor plots.

☒ A numerical value for number of cells or percentage (with statistics) is provided.

### Methodology

Sample preparation    cells were resuspended in PBS, fixed using 2% paraformaldehyde for 20 min at 4°C and permeabilized using PBS supplemented with 0.5% Triton X100 for 5 min at 4°C. After washing, cells were incubated for 1 h with an anti-tag antibody. Cells were finally washed three times before analysis.

Instrument           MACSQuant Analyzer 10 cytometer

Software             FlowJo (BD Biosciences)

Cell population abundance   For PE staining:
Total cell count: around 18K cells/sample.

For APC staining:
Total cell count: around 55K cells/sample

For FITC staining:
Total cell count: around 25K cells/sample.

Gating strategy      Gating was performed on the negative controls (cell transfected with a vector not expressing the receptor and stained with the anti-tag antibody). Positive staining was defined above 10^3 of PE-A/APC-A/FITC-A signals.

☒ Tick this box to confirm that a figure exemplifying the gating strategy is provided in the Supplementary Information.

