## [Peer Review file · Nature]

Heart-nosed bat alphacoronaviruses use human CEACAM6 to enter cells

Corresponding Author: Dr Dalan Bailey

Version 0:

Reviewer comments:

Referee #1

(Remarks to the Author)

The manuscript by Gallo et al., titled “Heart-nosed bat alphacoronaviruses use CEACAM6 as a receptor to enter human cells”, investigates receptor usage across representative alphacoronaviruses from diverse phylogenetic clades. The authors screened human receptors and libraries of ACE2 and APN orthologues from multiple species. They found that most selected bat spikes did not utilize known coronavirus receptors, except for two APN-using viruses (AT1A-F41 and WA1087) that are phylogenetically related to previously identified APN-dependent strains. Notably, screening of 759 human receptor ectodomains identified CEACAM family proteins as binding partners for CcCoV-KY43. Functional studies demonstrated that CEACAM6 serves as a functional receptor for this strain and several related duvinacoviruses with broad geographic distribution. The authors further resolved a crystal structure of the CcCoV-KY43 RBD–CEACAM6 complex, revealing vertical binding to the N-terminal V-type Ig-like domain in an orientation reminiscent of CEACAM6 homodimerization. Interestingly, a single mutation (F63I) enabled CEACAM5 to mediate moderate receptor function. Analysis of human blood samples from high-risk population centers revealed no evidence of spillover events. Overall, the study identifies a novel alphacoronavirus receptor, supported by high-quality biochemical, genetic, and structural evidence. The findings are both novel and significant for pandemic preparedness. The related papers are properly cited. I have several suggestions that can probably further improve the quality of the current manuscript.

Major Comments:

1. The introduction does not mention the CEACAM family. Since CEACAM6 is the newly identified receptor, it would be helpful to introduce that CEACAM1a is a receptor for MHV (NTD-binding).
2. Can the authors demonstrate whether CEACAM6 supports cell–cell membrane fusion mediated by CcCoV-KY43 S?
3. In Fig. 5b, Egyptian fruit bat CEACAM6 appears more compatible with RBDs from this viral clade. I suggest the authors explore the molecular basis for species orthologue specificity and why some strains fail to recognize human CEACAM6. This information would be important for risk assessment.
4. In Fig. 2, Calu-3 and Caco-2 cells appear susceptible to CcCoV-KY43. Demonstrating strict CEACAM6 dependence via knockdown or knockout in these cell lines would strengthen the conclusion that this receptor is essential for entry.

Minor Comments:

1. Please show data points for all bar chart graphs.
2. Please consider including an alignment of the N-terminal domains (or binding interfaces) of CEACAM1, 3, 5, and 6 in Extended Data.
3. Line 68: Revise “two domains” to “two subunits.”
4. Line 109: Remove “spike” in “virus spike can entry,” as this is a property of the virus, not the protein.
5. Figure 2g: There is a marked difference between 2.5 and 5 µg/mL concentrations. Repeating the assay would help confirm the IC50.
6. Figure 2h: Include a control protein such as CEACAM1 or CEACAM3 to confirm specificity. Please also specify the cell type used in this assay.
7. Figure 4b: While overall reactivity to KY43 RBD is low, some samples show OD ~1 or higher. Clarify whether these are considered positive signs of potential spillover, or just false positives.

8. Figure 5: Show RBD sequence features and explain how they may influence species specificity in light of structural data.
9. Figure S2: Present multiple strains on the same blot with a reference control for comparison. Correct “33K g” in the legend to “33,000 × g.”
10. Figure S8: Explain why CEACAM3 and CEACAM5 show strong binding in this screen yet lack corresponding functional activity. Please also specify the positive controls used.
11. Figure S13a: The dashed box in the lower panel appears misplaced. Mention in the legend that the MOW15-22 footprint on the opposite side is not shown.
12. Figure S16: Specify the BLI concentration used; include at least three concentrations.

Referee #2

(Remarks to the Author)

Summary

Coronaviruses pose a constant zoonotic threat to both human and animal health. A detailed understanding of how these pathogens engage with host cell receptors is essential for pandemic preparedness. Since the onset of the COVID-19 pandemic, substantial effort has been directed toward elucidating the entry mechanisms of betacoronaviruses, while members of other genera remain comparatively understudied.

The manuscript by Gallo et al. reports the discovery of CEACAM6 as an entry receptor for a clade of bat alphacoronaviruses. The authors show that human CEACAM6 can be utilized by African members of this clade, whereas related viruses from Asia and Europe can only engage the bat homolog. This is the first study to identify CEACAM6 as a coronavirus receptor and includes structural analysis of the RBD–receptor interaction. The quality of the X-ray crystallography data is of sufficient quality to support the proposed mode of binding.

In addition, the authors propose that the majority of non-human alphacoronavirus spike proteins (37 out of 38 tested) are incompatible with known human CoV receptors (ACE2, APN, DPP4, and TMPRSS2) and cannot infect a range of human cell lines derived from different organs. These findings provide important insights into the barriers to cross-species transmission and the potential zoonotic risk posed by bat alphacoronaviruses.

I am particularly impressed by the breadth of the initial receptor screening and the integration of multiple disciplines, including genetic analysis, biochemistry, infection assays, structural biology, and serology, to explore the implications of CEACAM6 usage by bat alphacoronaviruses. This work represents a significant advance in our understanding of alphacoronavirus diversity, particularly within bats. The findings also underscore the need for follow-up studies to assess the potential for human transmission in regions such as Kenya, and to evaluate whether related bat alphacoronaviruses might similarly exploit human CEACAM6.

Considering the above, I believe this study would be an excellent fit for Nature once the following points have been addressed.

Primary point to address

The discovery and validation of CEACAM6 as a novel coronavirus receptor is convincing and I have little, if any, concerns with this aspect of the manuscript. However, I have reservations regarding the first part of the study which claims that the majority of non-human alphacoronavirus spike proteins are incompatible with known human CoV receptors.

While the authors present evidence that the majority of tested alphacoronavirus spikes do not enter cells expressing ACE2, APN, DPP4, and TMPRSS2, I wondered whether alternative explanations for the observed lack of infection could be considered. Specifically, I am curious about the potential role of spike processing and conformational dynamics in receptor engagement.

There is precedent in the literature for betacoronaviruses and deltacoronaviruses that spike trimers may not bind to a known receptor, or receptor-mimicking antibodies, while isolated RBDs can[1,2]. In the study by Du et al, it was demonstrated that PDCoV S ectodomains could only bind to APN after being heat treated. The same paper also demonstrated that full-length, cell surface-expressed PDCoV spikes could not bind antibodies targeting the RBD, suggesting that some additional biological cue might be required for receptor binding and subsequent fusion. Considering this, can the authors rule out the possibility that some of the spikes examined in their study may adopt conformations that prevent receptor binding in the context of the pseudoviruses, but not in the context of an exposed RBD?

Related to this, some of the authors anti-HA Western blots used to assess pseudotyping efficiency show double bands, which could suggest differences in spike processing, potentially similar to the molecular weight differences seen in trypsin-treated PDCoV, or FCoV-23 after loss of domain 0[3,4]. Since pseudoviruses were produced in human cells, is it possible that key proteolytic processing events required for receptor binding or entry might not occur (or might occur aberrantly) in this system? Along those lines, can the authors exclude the possibility that additional host factors, such as proteases or co-receptors, such as observed for HKU1[5], might be necessary for enabling receptor usage in their screening assays?

Overall, I suggest the authors consider whether testing all, or a subset, of RBDs for receptor binding could help clarify these possibilities and strengthen their conclusions regarding receptor incompatibility.

Secondary points to address

- Please explain in more detail in the main text how the greedy algorithm works that was used to decide on which S sequences to test while retaining overall alphacoronavirus sequence diversity. It would be helpful to know which decisions were made to support the claim that the known alphacoronavirus diversity was studied here and to align with the final paragraph of the discussion.
- It appears the algorithm was trained using complete S sequences. Would it have made a difference if it were to be trained using RBD sequences only (arguably a better indicator for receptor usage and hence zoonotic potential)?
- While the evidence for use of CEACAM6 as an entry receptor by CcCoV KY43 is convincing, it would be good to see that the observed entry of Calu3 and Caco2 cell lines by CcCoV KY43 depends on this membrane protein.
- I appreciate that the authors do not make overly strong claims about their serology data (Fig. 4), yet it appears to me that zoonotic introduction has taken place in a limited number of individuals. Why is this not discussed in more detail? Are there more likely alternative explanations for this observation? It could indicate that the virus is zoonotically transmitted in the field but is incapable of sustained transmission among humans and it would be good to follow up on this or at least comment on it. Additionally, it would have been preferred to use the CcCoV KY43 full spike ectodomain instead of the RBD as this allows for a better comparison to the controls (229E, NL63 and SARS2) for which full length is used. The signal would presumably be greater in that case and provide a better understanding of the extent of zoonotic transmission that may be occurring in Kenya. This should, at the very least, be mentioned as a limitation of the study.
- Please provide data on CEACAM6 Q123H mutation and CEACAM5 H123Q, perhaps in combination with the discussed I63F mutation. Alternatively, it would be interesting to see what changes in the viral RBD would allow functional binding to CEACAM5. Given the similarity between CEACAM6 and 5, it is plausible that minor changes would already allow for this or even that the higher affinity CcCoV 2A variant can also use CEACAM5 as a receptor.
- Given the usage of CEACAM1a as a receptor by MHV, could you please include any thoughts in the discussion as to why CEACAM members might make for excellent CoV receptors?

Minor comments

- Could you explain why some entry is observed for pseudovirus with NL63 spike in presence of human APN (Fig 1)?
- Line 40: change alphacoronavirus to alphacoronaviruses.
- Line 61: please change to O-acetylated sialic acids. Also, HKU1 should be included along with OC43 as O-acetylated sialic acid as it is equally important as TMPRSS2 for fusion (see Figure 1H in Fernandez et al)[6].
- Line 60-61: You are missing references to support this statement[7–10]. Please check throughout the manuscript to ensure the relevant literature is cited.
- Line 68: please change domains to subunits.
- Line 95-96: please include references to existing literature that disproves APN as an entry receptor for PEDV (i.e. critical for fusion of viral and host membranes).
- Line 106-108: ACE2 and APN of non-human origin studied but not DPP4 or TMPRSS2, please specify statement accordingly.

References:

1. Du, W. et al. Neutralizing antibodies reveal cryptic vulnerabilities and interdomain crosstalk in the porcine deltacoronavirus spike protein. *Nat Commun* 15, 5330 (2024).
2. Wang, C. et al. Antigenic structure of the human coronavirus OC43 spike reveals exposed and occluded neutralizing epitopes. *Nat Commun* 13, (2022).
3. Xiong, X. et al. Glycan Shield and Fusion Activation of a Deltacoronavirus Spike Glycoprotein Fine-Tuned for Enteric Infections. *J Virol* 92, (2018).
4. Tortorici, M. A. et al. Loss of FCoV-23 spike domain 0 enhances fusogenicity and entry kinetics. *Nature* (2025) doi:10.1038/s41586-025-09155-z.
5. Pronker, M. F. et al. Sialoglycan binding triggers spike opening in a human coronavirus. *Nature* (2023) doi:10.1038/s41586-023-06599-z.
6. Fernández, I. et al. Structural basis of TMPRSS2 zymogen activation and recognition by the HKU1 seasonal coronavirus. *Cell* 187, 4246-4260.e16 (2024).
7. Saunders, N. et al. TMPRSS2 is a functional receptor for human coronavirus HKU1. *Nature* 624, (2023).
8. Raj, V. S. et al. Dipeptidyl peptidase 4 is a functional receptor for the emerging human coronavirus-EMC. *Nature* 495, (2013).
9. Huang, X. et al. Human Coronavirus HKU1 Spike Protein Uses O -Acetylated Sialic Acid as an Attachment Receptor Determinant and Employs Hemagglutinin-Esterase Protein as a Receptor-Destroying Enzyme . *J Virol* 89, (2015).
10. Vlasak, R., Luytjes, W., Spaan, W. & Palese, P. Human and bovine coronaviruses recognize sialic acid-containing receptors similar to those of influenza C viruses. *Proc Natl Acad Sci U S A* 85, (1988).

(Remarks to the Author)

I co-reviewed this manuscript with one of the reviewers who provided the listed reports.

Referee #4

(Remarks to the Author)

I was asked to review the phylodynamic analysis in this manuscript, which I take to also include the details needed to eventually run the analysis and interpret its results (e.g., data curation). As it pertains to the phylodynamics, the authors performed Bayesian phylodynamics on spike sequences using a protein evolution model and concluded that usage of CEACAM6 is a relatively recent acquisition. My comments will be primarily about this part of the work.

1. Neither the alignment, BEAST XML, nor the resulting log/tree files were provided. At a minimum, the alignment and XML should be provided for reproducibility.
2. A maximum likelihood phylogeny of all the initial sequences from “Construction of gene libraries” should be provided, highlighting the 40 spike protein coding sequences ultimately selected.
3. Subsequently, the authors could construct a nucleotide-based maximum likelihood phylogeny of the resulting 40 selected strains coupled with the sequences belonging to the local phylogeny of CcCoV|KY43 (these local sequences could also be highlighted on the larger phylogeny in point 2). When doing so, they should infer the support values at internal nodes. This could inform our understanding of how frequently the usage of CEACAM6 arose.
4. Nucleotide sequences are available for at least the samples in the local alignment, if not all the samples in this dataset. The Bayesian analysis should therefore use a nucleotide or codon substitution model, rather than an amino acid substitution model. An amino acid model cannot account for synonymous substitutions, thereby potentially resulting in an inaccurate inference of when CEACAM6 usage was acquired.
5. What is the posterior support for key internal nodes in the tree in Fig. 5A, particularly for the most recent common ancestor (MRCA) of the local alignment? If there is low posterior support at the MRCA of this node, the tMRCA of this particular node might not be precisely what the authors are hoping to infer. Rather, they can parameterize their analysis by defining the key taxa as a taxon set in BEAUti and specifically infer the tMRCA of that taxon set. When doing this analysis, they should also examine how frequently this taxon set is monophyletic (importantly, this taxon set should not be specified as monophyletic a priori in BEAST).
6. Fig. 5A suggests YN2020 and Kuddep are part of the local phylogeny, but the focal MRCA is not ancestral to these two. Should these two samples therefore not be considered part of the local alignment? Additionally, Kuddep appears to use CEACAM6 as well, which raises the possibility that this trait was acquired or lost multiple times. This more complex evolutionary history should be briefly discussed. As it relates to point 5, the authors should consider whether Kuddep should be part of the taxon set and interpret the subsequent results in that context.

Minor comments:

- Lines 214–221. Please also report nucleotide identities.
- Supplementary Figures S4, S5, etc.: It could be helpful to list pathogens that were not tested, even if this results in rows of “X”s, for completeness.
- Supplementary Figure S6: It could be useful to showcase an analogous analysis for hCoV-229E:inf-1, for completeness.
- Supplementary Figure S18: should CcCor|2A and CcCor|2B be CcCoV|2A and CcCoV|2B?
- Coronaviruses, including sarbecoviruses, are known to exhibit the time dependent rate phenomenon (TDRP), which can further bias time to the MRCA (tMRCA) estimates toward more recent dates. The ability to use CEACAM6 could therefore have been acquired earlier than the authors suggest. Although the examination of TDRP is outside the scope of the manuscript, it should at least be mentioned in the discussion or limitations.

Referee #5

(Remarks to the Author)

In this paper Gallo et al used a computational approach (greedy algorithm) to select a limited but phylogenetically wide range of alphacoronavirus spike proteins for study. Including controls, 40 spike proteins were examined - 27 of which were from bats. A lentiviral spike protein pseudo typed entry assay was used to assess entry and receptor usage. Tested were various cells lines and cells expressing 25 APNs and 34 ACE2s (the two known alphacoronavirus receptors) from diverse species. In addition, a protein array screening approach was used and a new alphacoronavirus receptor was identified.

The main conclusions can be summarized as follows: i) a limited number of the alphacoronavirus spike proteins tested use APN or ACE2 as receptors and ii) the bat alphacoronavirus CcCoV|KY43 can use human CEACAM6. Given these results, the authors speculate that there must be other alphacoronavirus receptors that have not yet been identified.

Given the potential for spillover to humans the identification of a bat coronavirus that can use human CEACAM6 as receptor is an important finding from a pandemic preparedness and human health perspective. It also broadens our knowledge of coronavirus receptor usage, specifically for alphacoronaviruses, a genus less well studied than that of betacoronaviruses. As such, the work will be of broad interest in the fields of virology and public health.

However, the following need to be considered/addressed:

1) The main concern is the conclusion that very few of the viruses whose spike proteins were tested use APN or ACE2 as receptors. This is stated in the abstract and at a number of places in the paper. It is also possible that these viruses are highly specific for the APN or ACE2 of the species they infect and that those receptors were not tested. It must be made clear what percentage were tested against the APN or ACE2 from the species they infect (see line 86). The authors acknowledge that some may be "hyper-specialised" and in a paper in BioRxiv (Gallo et al 2024), containing some of the data in this paper, they describe the CcCoV's as specialists. Is it not possible that the majority of the bat viruses tested are specialists? This must be explained/discussed.

2) The entry assay is also a potential source of error as it is possible that many of the spike proteins do not give functional pseudo types. It seems that positive controls - feline CoV and two different PEDV S proteins - did not show entry (see lines 94-97). How do the authors rationalize this? The authors checked spike incorporation into the purified pseudo viruses, but this does not establish that they are in the prefusion conformation or at levels sufficient for membrane fusion. These concerns must all be addressed.

3) A lack of spike cleavage is another factor that may be leading to false negative entry results with the various APN and ACE2 cell lines. How well conserved is the S2' cleavage site among the spike proteins tested? The authors have included TMPRSS2 in the entry assay, but they show that it made no difference in the few cases where entry was observed. How do the authors interpret that observation? Verifying that the purified pseudoviruses can be cleaved by TMPRSS2 (as described by Dufloo et al, ref 9 in this paper) would provide greater confidence in the entry results and strengthen the claim that most of the viruses tested do not use APN or ACE2. This would still not rule out the possibility that they are all specialists. Trypsin is often used in these entry assays, and it too might be tested.

4) The authors have determined two crystal structures that provide a detailed description of the residues involved in binding CEACAM6. The results should be better discussed in relation to the other results presented. Among the 40 spikes examined, do any of the others contain similar loops/residues (eg W600)? The authors have also expanded the phylogenetic tree around CcCoV/KY63 - are the important residues conserved? If not - does this provide further insight into what defines the critical interactions? An expanded CEACAM6 screen was also performed - are the critical CEACAM6 residues conserved?

5) As shown by the PDB validation reports both refined structures have a relatively high R_{free} and a relatively large number of RSRZ outliers - values that can indicate problems with structures (CcCoV-2B is better). In this case, these values are almost certainly the result of the fact that both crystals diffract anisotropically. The authors have taken appropriate steps to minimize the negative impacts and the electron density at the RBD-CEACAM6 interface supports the reported models and the conclusions made regarding the interaction. Figure 4h would be improved by showing a comparison of the two interfaces, not just a reference to Figure 3b. I think that the use of stereo pairs in Figure S11 should be reconsidered. It is important to show the density and a detailed comparison of the two complexes, but many readers do not have the required viewer.

6) The methods section mentions BLI but only ITC data is described in the main.

Version 1:

Reviewer comments:

Referee #1

(Remarks to the Author)

The authors have undertaken substantial additional work since the previous submission, and the revised manuscript represents a clear and significant improvement. The new data and explanations address my major concerns and convincingly strengthen the central conclusions:

(1) CEACAM6 functions as a bona fide receptor for CcCoV-related strains with global distribution, which is both evolutionarily and physiologically plausible; and (2) most alphacoronaviruses do not rely on ACE2 or APN for entry.

This study has important implications for coronavirus receptor biology, cross-species transmission, and the identification of potential vulnerabilities relevant to preparedness for future epidemics. The discussion integrating these findings into coronavirus receptor evolution is thoughtful and compelling. Overall, this work represents a substantial advance, and the manuscript is close to being suitable for publication, pending minor revisions.

Minor comments

1.Line 76: Consider adding "significantly" or "markedly" before "the richness and heterogeneity" to improve accuracy.

- 2.Line 98: Please double-check why feline CoV could not be pseudotyped, as it works well in our hands and in published studies.
- 3.Line 126: I recommend removing the word “spike” from the title.
- 4.Figure 5: Ensure consistent nomenclature between “YN20” and “YN2020.”
- 5.Figure S10: Please specify the Y-axis values (between 1 and 10). Given the modest fold change in luciferase activity, it would be helpful to also show representative GFP images.
- 6.Figure S12: All bars, including the siRNA groups, appear close to 100%. Please double-check the data and presentation.

Referee #2

(Remarks to the Author)

The authors have gone to great lengths to address our concerns, as well as those of the other reviewers. The new experimental data and textual revisions have substantially strengthened the conclusions, and we have no further comments. We believe this manuscript represents a welcome addition to the coronavirus literature.

Referee #3

(Remarks to the Author)

I co-reviewed this manuscript with one of the reviewers who provided the listed reports.

Referee #4

(Remarks to the Author)

The authors have adequately addressed my comments, and I would recommend the manuscript for publication. My only additional suggestion is to include the discrete-trait reconstruction of CEACAM6 usage provided in the rebuttal as a supplementary figure, as it makes clear the point regarding multiple emergence described in the text.

Referee #5

(Remarks to the Author)

I have read the revised manuscript and the rebuttal and feel that the authors have done a very good job at addressing reviewer comments. My concerns have been addressed, and I would also note that the addition of the RBD binding data (response to reviewer 2) is a significant addition. This data certainly helps to support the suggestion that many of the tested viruses do not use APN or ACE2 as receptor. I would recommend that the paper now be published.

2025-07-18615 Gallo et al., Response to reviewers

Dear Editors and Reviewers,

Firstly, we would like to thank you for your positive comments on the quality of our work and for recognizing the novelty of our study and its importance in improving pandemic preparedness and public health. We are also grateful for the suggestions made and questions raised, which we address point-by-point below. We also attach a tracked changes version of the amended manuscript. Of note, we have now created a Zenodo repository that hosts all the relevant files, along with Supplementary Materials and additional data rearranged from the initial submission. Until publication a temporary link to that directory is available for you here:

https://zenodo.org/records/17951484?preview=1&token=eyJhbGciOiJIUzUxMiJ9.eyJpZCI6ImVhNjc5MmEwLWM1NjgtNDk5Yi1iZi1LWYxMzc1N2E1ODRjMSIsImRhdGEiOnt9LCJyYW5kb20iOiJkZW_RhNzgwY2U4NjYxYWQzNDM5MTQ2YzlmMjRkMWRhMCJ9.IR8Vkb14Gp0NojUJs79g4QeurWj2bsQbXO61-1sqtkWU1iux6zQJtbqUHRFxb10er5lqvVxl3IWTnozUGUJKg

Yours sincerely,

Dr. Dalan Bailey

On behalf of all co-authors 18/12/25

Referee #1 (Remarks to Author):

The manuscript by Gallo et al., titled “Heart-nosed bat alphacoronaviruses use CEACAM6 as a receptor to enter human cells”, investigates receptor usage across representative alphacoronaviruses from diverse phylogenetic clades. The authors screened human receptors and libraries of ACE2 and APN orthologues from multiple species. They found that most selected bat spikes did not utilize known coronavirus receptors, except for two APN-using viruses (AT1A-F41 and WA1087) that are phylogenetically related to previously identified APN-dependent strains. Notably, screening of 759 human receptor ectodomains identified CEACAM family proteins as binding partners for CcCoV-KY43. Functional studies demonstrated that CEACAM6 serves as a functional receptor for this strain and several related duvinacoviruses with broad geographic distribution. The authors further resolved a crystal structure of the CcCoV-KY43 RBD–CEACAM6 complex, revealing vertical binding to the N-terminal V-type Ig-like domain in an orientation reminiscent of CEACAM6 homodimerization. Interestingly, a single mutation (F63I) enabled CEACAM5 to mediate moderate receptor function. Analysis of human blood samples from high-risk population centers revealed no evidence of spillover events. Overall, the study identifies a novel alphacoronavirus receptor, supported by high-quality biochemical, genetic, and structural evidence. The findings are both novel and significant for pandemic preparedness. The related papers are properly cited. I have several suggestions that can probably further improve the quality of the current manuscript.

We would like to thank the reviewer for their positive feedback and for recognising the significance of our work. We have addressed their concerns and suggestions point by point below:

Major Comments:

1. The introduction does not mention the CEACAM family. Since CEACAM6 is the newly identified receptor, it would be helpful to introduce that CEACAM1a is a receptor for MHV (NTD-binding).

We agree with the reviewer and have now more conclusively summarised the known receptor usage of a wider range of coronaviruses in the main text, to include MHV and murine CEACAM1 as well as the novel identification of DPEP1 as the PHEV receptor. As follows:

“Recently, porcine and human DPEP1 were both shown to be efficient receptors for porcine hemagglutinating encephalomyelitis virus (PHEV) although there is no evidence for zoonotic infection¹. Entry of the betacoronavirus murine hepatitis virus (MHV) is supported by murine CEACAM1², although in this case the human orthologue is not efficiently used³.”

2. Can the authors demonstrate whether CEACAM6 supports cell–cell membrane fusion mediated by CcCoV-KY43 S?

Using an adapted split reporter assay we have now confirmed that human CEACAM6, but not CEACAMs 1, 3 or 5, can support CcCoV|KY43 S mediated cell-cell fusion. This supplementary data has been added to the main text as follows:

“A split luciferase-based cell-cell fusion reporter assay subsequently confirmed this receptor usage profile in a separate cellular context (Fig.S10).”

Legend: **Figure S10. Human CEACAM6 supports cell-cell fusion in the presence of CcCoV|KY43 S.**

A fusion assay, based on a split firefly luciferase and GFP, was used to validate usage of human CEACAM6 as a cognate receptor by CcCoV|KY43 S. The average fold change from three independent experiments, each performed in triplicate, is shown along with the standard deviation. Statistical significance was examined using a one-sample t test against a theoretical mean of 0, with standard errors calculated using a pooled variance estimate across all groups to ensure robust variance estimation given the small sample sizes. P-values were adjusted for multiple comparisons using the Holm-Bonferroni correction (p-value <0.033, ns: non-significant difference).*

3. In Fig. 5b, Egyptian fruit bat CEACAM6 appears more compatible with RBDs from this viral clade. I suggest the authors explore the molecular basis for species orthologue specificity and why some strains fail to recognize human CEACAM6. This information would be important for risk assessment.

Understanding the molecular determinants of host range is at the core of our investigation, with the overall aim being, as you rightly say, to improve our ability to risk assess viruses for their zoonotic potential. To address your question, we focused our investigations on both the known human CEACAM6-using virus, CcCoV|KY43, and the more restricted ‘non-human’ but ‘Egyptian fruit bat-using’ CEACAM6 user, BtRfCoV/LN20 (see Figure 5) – which was isolated in greater horseshoe bats in 2020. We chose BtRfCoV/LN20 because we had already demonstrated this spike could use a non-human primate CEACAM6 in our assays; this, we considered made it the most likely candidate for subsequent adaptation to human CEACAM6. Initially, we established that the RBD domain of spike is the main determinant of restriction by making a BtRfCoV/LN20 chimera, where the cognate RBD was replaced with that of CcCoV|KY43, and confirmed that this chimera could use human CEACAM6 (‘KY43 RBD’ below). Supported by our knowledge of the crystal structure of the interaction, we next focused on individual amino acid changes within the spike-CEACAM6 interface. Although no single amino acid change was able to confer human CEACAM6 tropism to BtRfCoV/LN20, we did show that F607Y and combinatorial changes at position 557 and 558 could expand tropism to other mammalian

CEACAM6 proteins (see below). Whilst this is not the ‘human CEACAM6-using smoking gun’ we were after, it does provide a greater understanding of “species orthologue specificity”. The following changes have been made to the main text:

“No consistent amino acid substitution pattern at the RBD interface distinguishes CEACAM6-like proteins that permit CcCoV entry from those that do not. However, viruses that utilise human CEACAM6 conserve multiple RBD sequence features that are absent from other RBDs, for example the length and amino acid compositions of loops 1 and 2 (Fig.29a). Point mutations were introduced into the RBD of BtCoV/LN20, which uses various CEACAM6-like proteins but not the human orthologue, to better define the genetic determinants of human receptor usage and zoonotic spillover (Fig.S29b). Although we could expand the host range phenotype of BtCoV/LN20, individual changes did not confer human CEACAM6 tropism, suggesting multiple changes are needed and the risk of rapid adaptation of these viruses to zoonotic spillover is lower.”

Legend: Figure S27b. Investigating the genetic determinants of btCoV/LN20 receptor usage, a CEACAM6-specialist. Entry assays using pseudotyped btCoV/LN20 S show that single mutations within the RBD, to make it more like CcCoV|KY43 S, do not individually introduce human CEACAM6 receptor usage. However, a complete RBD chimera (‘KY43 RBD’) confirms the RBD is in itself the main determinant of receptor usage. The background-corrected average of three independent experiments, each performed in triplicate, is shown.

4. In Fig. 2, Calu-3 and Caco-2 cells appear susceptible to CcCoV-KY43. Demonstrating strict CEACAM6 dependence via knockdown or knockout in these cell lines would strengthen the conclusion that this receptor is essential for entry.

We have now used a panel of siRNAs and a scrambled control to human CEACAM6, to validate KY43 entry in both cell lines, Calu-3 and Caco-2 (those that were permissive in our earlier cell-line screening, Fig. 2i). siRNA knockdown of human CEACAM6 was confirmed by immunoblot. In a separate control experiment we confirmed that entry of the APN-using relative, 229E, was unaffected by this siRNA treatment. We also used shRNA and saw the same effect. The following changes to the main text have been made to include these results:

“To further confirm the role of human CEACAM6 in CcCoV|KY43 entry we used CEACAM6-specific siRNAs to knockdown endogenous protein expression in the susceptible cell lines Caco2 and Calu3, observing reduced virus entry for CcCoV|KY43 (Fig.2i, left) but not hCoV/229E-inf1 (Fig.S12). The same results were obtained in shRNA-treated Calu3 and Caco2, selected to stably downregulate the expression of human CEACAM6 (Fig.2i, right).”

Legend: Figure 2i. Knock-down of human CEACAM6 in Caco2s and Calu3s reduces CcCoV|KY43.

Human CEACAM6-specific siRNAs, or negative scrambled controls, were electroporated into Caco2 and Calu3 cells and then infected with KY43 (left panel). To confirm CEACAM6 reduction, cell lysates were analysed by immunoblot. For CcCoV|KY43, the average reduction in entry from three independent experiments, each performed with four replicates is shown, normalized to the scrambled siRNA control. Cells were used for entry assay with pseudotyped CcCoV|KY43 S. Significance of \log_{10} fold change was determined using a one-sample *t*-test against a \log_{10} fold change of 0 (no change), with *P*-values were adjusted for multiple comparisons (* for $p < 0.05$). Using lentivirus expressing shRNAs, we produced stable knock-down of Caco2 and Calu3 cell lines to reduce human CEACAM6 expression, and validated it by immunoblot (right panel). One biological replicate performed with technical triplicates is shown.

Legend: Figure S12. Transient knock-down of human CEACAM6 using siRNA does not affect hCoV/229E entry. One day after electroporation, Caco2 and Calu3 cells (the same used in experiments described in Figure 2i) with different siRNA against human CEACAM6 were tested for their permissivity to hCoV/229E-inf1, showing no phenotypical effect. One experiment consisting of technical triplicates is shown.

Minor Comments:**1. Please show data points for all bar chart graphs.**

As requested, we have now updated all the bar charts to show the individual data points.

2. Please consider including an alignment of the N-terminal domains (or binding interfaces) of CEACAM1, 3, 5, and 6 in Extended Data.

An alignment of the N-terminal domains of human CEACAM proteins has been included as follows, highlighting the spike interaction interface, with reference made in the text as follows:

“Since the human CEACAM N-terminal Ig domains are relatively conserved (Fig.S22, RBD binding region in teal), we also examined CEACAM1, 4, 8 and 20 in CcCoV|KY43 entry assays (Fig.S23, Table S7).”

	35	61	83	
CEACAM6	KLTIESTPFNVAEGKEVLLLAHNLPQ	NRIGYSWYKGERVDGNSLIVGYV	IGTQQATPGPA	
CEACAM1	QLTTESMPFNVAEGKEVLLLVHNLPQ	QLFGYSWYKGERVDGNRQIVGYA	IGTQQATPGPA	
CEACAM3	KLTIESMPLSVAEGKEVLLLVHNLPQ	HLFGYSWYKGERVDGNSLIVGYV	IGTQQATPGAA	
CEACAM4	QFTIEALPSSAAEGKDVLLLACNISE	TIQAYYWHKGTKAEGSPLIAGYIT	TDIQANIPGAA	
CEACAM5	KLTIESTPFNVAEGKEVLLLVHNLPQ	HLFGYSWYKGERVDGNRQIIGYV	IGTQQATPGPA	
CEACAM8	QLTIEAVPSNAAEGKEVLLLVHNLPQ	DPRGYNWYKGETVDANRRIIGYV	ISNQQITPGPA	
CEACAM20	QLTLNANPLDATQSEDVVLVVF	GTPTPQIH	-----	
		121	133	145
CEACAM6	YSGRETIYPNASLLIQNVTQNDTGFY	TLQVIKSDLVNEE	ATGQFHVYPELP	
CEACAM1	NSGRETIYPNASLLIQNVTQNDTGFY	TLQVIKSDLVNEE	ATGQFHVYPELP	
CEACAM3	YSGRETIYTNASLLIQNVTQNDIGFY	TLQVIKSDLVNEE	ATGQFHVYQENA	
CEACAM4	YSGRETVYPNGSLLFQNI	TLEDAGSYTLRTINASYDSDQ	ATGQLHVHQN	NV
CEACAM5	YSGREI IYPNASLLIQNI	IQNVTGFIYTLHVIKSDLVNEE	ATGQFRVYPELP	
CEACAM8	YSNRETIYPNASLLMRNVTRNDTGS	YTLQVIKLNLMSEE	VTGQFVHPETP	
CEACAM20	--GRSR-----			ELA

Legend: Figure S22. Sequence comparison of the amino-terminal V-type Ig-like domain of human CEACAMs.

The alignment shows high conservation of CcCoV|KY43 binding site; however, human CEACAM6 is the only protein with an isoleucine at position 63, which is a key determinant of receptor usage. Residues involved in the interaction with CcCoV|KY43 RBD are underlined. The percentage amino acid identity of each domain with CEACAM6 is as follows: CEACAM1, 90.1%; CEACAM3, 88.3%; CEACAM4, 48.6%; CEACAM5, 89.2%; CEACAM8, 72.1%; CEACAM20, 26.3%.

3. Line 68: Revise “two domains” to “two subunits.”

Thank you, this has been corrected in the text.

4. Line 109: Remove “spike” in “virus spike can entry,” as this is a property of the virus, not the protein.

Thank you for highlighting this discrepancy. We have checked and corrected all instances where it was suggested that entry is a functional characteristic of the Spike protein, not the virus.

5. Figure 2g: There is a marked difference between 2.5 and 5 µg/mL concentrations. Repeating the assay would help confirm the IC50.

This experiment was performed on two independent occasions, as shown in Figure 2g. In both experiments we observed the marked difference in entry between the 2.5 and 5 µg/mL concentrations. We ascribe this to multivalency in binding to KY43 Spike of the Fc-tagged CEACAM6 used in this assay, and to the cooperative nature of glycoprotein-mediated pseudoparticle entry. The Fc tag mediates dimerisation of CEACAM6 and this will strongly enhance the avidity of binding to the trimeric

KY43 spike protein, increasing the Hill coefficient and thus the steepness of the IC50 curve⁴. Furthermore, virus glycoprotein-mediated membrane fusion is often cooperative, driven by ~2-7 glycoprotein trimers⁵. There will be a dramatic drop in fusion efficiency once the concentration of an inhibitor (e.g. soluble receptor) reaches a threshold where the number of uninhibited trimers is below the number required to mediate membrane fusion, which would manifest again as a steep IC50 curve with Hill coefficient > 1. We note that a similarly steep IC50 curve was observed for entry of SARS-CoV-2 Spike pseudotypes when inhibited by dimeric or trimeric ACE2⁶.

6. Figure 2h: Include a control protein such as CEACAM1 or CEACAM3 to confirm specificity. Please also specify the cell type used in this assay.

We agree with the need for further validation of the KY43 entry inhibition experiments, in which we used two anti-human CEACAM6 antibodies to inhibit virus entry (Fig.2h). To examine potential binding to related CEACAMs, which would confound our conclusions, we used flow cytometry to demonstrate that these antibodies are specific and do not bind CEACAMs 1, 3 or 5 (see below). This figure is now referenced in the relevant part of the main text as follows:

“Additionally, we demonstrated dose-dependent neutralization of CcCoV|KY43 pseudotyped S using soluble, recombinant CEACAM6 (Fig.2g), as well as blockage of entry using monoclonal antibodies against human CEACAM6 (Fig.2h, Fig.S11)”

Legend: Figure S11. Monoclonal antibodies B6.2 and 493424 are specific to human CEACAM6. HEK293T cells transfected with plasmids encoding the indicated human CEACAMs were stained using the anti-CEACAM6 commercial monoclonal antibodies, B6.2 and 493424. FITC-conjugated anti-mouse was used as secondary antibody to detect binding of the monoclonals. Surface expression was then quantified using a flow cytometer.

7. Figure 4b: While overall reactivity to KY43 RBD is low, some samples show OD ~1 or higher. Clarify whether these are considered positive signs of potential spillover, or just false positives.

We agree that higher CcCoV|KY43 ELISA signals (OD>1) could, in principle, be consistent with limited zoonotic transmission without sustained human-to-human spread. In the amended main text, we now discuss this possibility alongside alternative explanations, including cross-reactive antibody responses elicited by exposure to antigenically related coronaviruses. In addition, and to provide wider context for the observed low-frequency KY43 ELISA reactivity, we have included ELISAs (performed with all 368 sera) from additional, distantly-related, bat-derived alphaCoVs, HIYN18 and A701 (also in our algorithmic library). These antigens were selected to provide additional comparators and to allow assessment of the specificity of sporadic high OD signals. This is important in the absence of well-defined negative-control sera in this emergence setting. Our discovery of similarly infrequent high-OD responses against HIYN18 and A701 supports one interpretation that rare KY43 signals are likely cross-reactive or non-specific reactivity, rather than widespread zoonotic spillover. However, we also demonstrate, using statistical correlation analyses, that the higher CcCoV|KY43 ELISA signals do not associate strongly with the included coronaviruses, possibly providing evidence for spillover. We have modified the main text as follows to discuss this additional experimentation and analyses:

“A total of 368 blood donor samples collected in 2020 and 2021, comprising mainly males (95%) under 35 years of age (72%) living in the Tana River and Taita-Taveta counties where CcCoV|KY43 was identified (Table S8), were examined for CcCoV|KY43, 229E, NL63 and SARS-CoV-2 S-specific responses by ELISA (Fig. 4b, inset, Taita-Taveta county highlighted in pale red). We also included two RBDs from more distantly related alphaCoVs (HIYN18 and A701, both isolated in bats in China) as comparators. Across the full dataset, Spearman’s rank correlation analysis revealed weak monotonic associations between CcCoV|KY43 and 229E, SARS-CoV-2, HIYN18 and A701 ($\rho = 0.14\text{--}0.30$), whereas no association was observed with NL63 ($\rho = 0.06$) (Fig. 4b). Although ELISA reactivity against the human alphaCoVs 229E and NL63 was commonly observed, elevated CcCoV|KY43 ELISA signals were detected only sporadically, with OD values exceeding 1 in nine samples. Using this OD threshold as a descriptive marker of high reactivity rather than evidence of seropositivity, similarly infrequent high-OD signals were observed for HIYN18 ($n = 5$) and A701 ($n = 2$). Restricting the analysis to samples comprising the top 10% of CcCoV|KY43 ELISA signals, Pearson’s correlation analysis of matched sera revealed moderate correlations with NL63 and SARS-CoV-2, but only weak correlation with 229E, HIYN18 and A701 (Fig. S25). Together, these data suggest that widespread CcCoV|KY43 spillover is unlikely in these populations. Alternatively, sporadic KY43 reactivity may reflect cross-reactive antibody responses elicited by exposure to antigenically related coronaviruses.”

the analysis to samples comprising the top 10% of CcCoV|KY43 ELISA signals, Pearson’s correlation analysis of matched sera revealed moderate correlations with NL63 and SARS-CoV-2, but only weak correlation with 229E, HIYN18 and A701 (Fig. S25). Together, these data suggest that widespread CcCoV|KY43 spillover is unlikely in these populations. Alternatively, sporadic KY43 reactivity may reflect cross-reactive antibody responses elicited by exposure to antigenically related coronaviruses.”

Legend: **Figure 4b.** Human sera from volunteers ($n=368$) in Tana River and Taita-Taveta counties (highlighted in pale red in the inset map), were analyzed for their reactivity to different human coronavirus glycoproteins. Spearman’s rank correlations ρ compared to the CcCoV|KY43 RBD are provided.

Legend: **Figure S25.** **Correlation of high CcCoV|KY43 ELISA reactivity with responses to other coronavirus antigens.** Top: ELISA optical density (OD) values for all samples within the upper 10% of the KY43 distribution. Below: Pearson’s correlation analysis was performed on this enriched subset of ‘high’ KY43 samples to assess co-variation. Each point represents an individual serum sample tested against matched antigens. Correlation coefficients (r) are shown for each comparison. Of note, the top 10% threshold was used as a descriptive enrichment criterion

and not to infer seropositivity or compare absolute signal magnitudes across assays.

8. Figure 5: Show RBD sequence features and explain how they may influence species specificity in light of structural data.

We have now included an additional, expanded sequence alignment (see below) as Figure S29a, to avoid redundancy with Figure 4c. Of note, we are happy to include it instead as an additional panel in Figure 5, at the Reviewer and Editor’s discretion. We note that the RBDs of human CEACAM6-using viruses have high sequence conservation in loops 1 and 2, whereas the other RBDs have divergent sequences. For example, only human CEACAM6-users conserve R517 (CcCoV|KY43 numbering) and S519, both of which make hydrogen bonds to CEACAM6, and that substitution of S555 with a bulkier amino acid like asparagine or threonine could cause steric hindrance that would disrupt the interaction. We also note that the lengths of loops 1 and 2 differ between the human CEACAM6 users and other viruses, with an insertions and deletions in loops 1 and 2 (respectively) that would alter conformations of these loops and thus their interactions with CEACAM6. We hope the functional aspects of this question (on species specificity) have been resolved in response to Q3 above. Briefly, selected single amino acid mutations were introduced into the RBD of the CEACAM6-using virus btRfCoV/LN20, to probe the structural determinants of primate CEACAM6 restriction. Pseudotype entry assays showed that while several mutations expanded CEACAM6 usage, none conferred human tropism.

The main text has been changed as follows to reference the availability of this alignment:

“However, viruses that utilise human CEACAM6 conserve multiple RBD sequence features that are absent from other RBDs, for example the length and amino acid compositions of loops 1 and 2 (Fig.S29a).”

Legend: Figure S29a. Alignment of RBD sequences from CEACAM6 using RBDs included in this study. Loops that interact with human CEACAM6 are highlighted in red. Residues that interact with human CEACAM6 in the CcCoV|KY43 and CcCoV|2B complex structure are shown in **bold red**. Residues of btCoV/LN20 that are mutated to probe the sequence determinants of human CEACAM6 usage (see below) are underlined.

9. Figure S2: Present multiple strains on the same blot with a reference control for comparison. Correct “33K g” in the legend to “33,000 × g.”

As way of confirmation, these blots were run at the same time with multiple strains on the same blot and with the use of internal controls. For clarity’s sake, however, we then cropped and reordered them to match the phylogenetic order of the spikes used elsewhere in the manuscript. We would like to keep this, as it helps comparison between figures, but we do understand the need to share the raw data. To this end, we have added and labelled all the original immunoblots, as requested and uploaded these to the Zenodo data repository. As requested the legend has also been corrected to 33,000 x g.

10. Figure S8: Explain why CEACAM3 and CEACAM5 show strong binding in this screen yet lack corresponding functional activity. Please also specify the positive controls used.

The assay used to identify the CEACAM receptors is called "Avidity-based EXtracellular Interaction Screening" (AVEXIS), which was developed to identify low affinity extracellular receptor-ligand interactions. The original paper describing the assay is available here: Bushell et al. Genome Research 2008⁷ and has been developed and improved over the years. The specific implementation of the assay used in this manuscript is described in more detail here: Shilts et al. Nature 2022⁸. It is known that many extracellular interactions can be of extremely low affinity, and the assay was developed to be able to detect these highly transient interactions. For example, the CD4-MHC Class II interaction has an equilibrium dissociation constant of around 100 micromolar⁹ and the IZUMO1-JUNO interaction involved in mammalian sperm-egg recognition has a K_D of around 10 micromolar¹⁰. Of more relevance, it is known that the affinity of virus attachment factors for their host receptors can also be very weak¹¹ making this assay an appropriate approach to identify putative host receptors for virus attachment factors. This is achieved by increasing binding avidity using the clustering of proteins around tetrameric streptavidin. This approach ensures that interactions with a very weak monomeric binding affinity can still be detected with excellent signal:noise ratios because their binding avidity is increased, as can be seen in the screening plates (Fig.S9). In our manuscript, we measured the interaction affinity between the KY43 spike protein and both CEACAM6 and CEACAM5 using isothermal calorimetry, demonstrating that the affinity for CEACAM6 (271 nM) is higher than CEACAM5 (which was so weak as to not provide a confident measurement). In the biological context of the functional virus experiments described here, the affinity for CEACAM5 and CEACAM3 receptors is likely too low for them to function as functional receptors, but is sufficient for them to be detected by our optimised AVEXIS.

11. Figure S13a: The dashed box in the lower panel appears misplaced. Mention in the legend that the MOW15-22 footprint on the opposite side is not shown.

We have now made the dashed boxes clearer, as requested, and modified the legend. Of note, the relevant figure is now numbered Figure S18.

12. Figure S16: Specify the BLI concentration used; include at least three concentrations.

To address the point, we repeated the BLI experiment, with both CcCoV|KY43 and CcCoV|2B RBDs, and five different concentrations of human CEACAMs, confirming our previous findings. The concentrations used are specified in the legend of this updated supplemental figure (Fig.S24), which is referenced as follows in the main text:

"None of these proteins were able to support entry of CcCoV|KY43 pseudotypes and CEACAM1, 4 and 8 did not show appreciable binding in bio-layer interferometry assays, including human CEACAM6 I63F (Fig.S24)."

Legend: Figure S24. Comparison of RBD binding to different human CEACAMs using bio-layer interferometry (BLI).

Ni-NTA sensors were loaded with His-tagged CcCoV|KY43 RBD (a) or CcCoV|2B (b) and then incubated with different human CEACAMs in a two-fold concentration gradient from 600 nM (dark colours) to 37.5 nM (light colours). BLI responses following subtraction of a reference sensor incubated with buffer are shown. The end of the association phase is marked (dashed vertical line). Human CEACAM6 binds strongly to CcCoV RBDs, and this binding is lost upon introduction of an I63F substitution. CcCoV RBDs bind weakly to human CEACAM5 and displays negligible binding the CEACAM1, CEACAM3 and CEACAM8. Data are representative of two independent experiments.

Referee #2 (Remarks to the Author):

Coronaviruses pose a constant zoonotic threat to both human and animal health. A detailed understanding of how these pathogens engage with host cell receptors is essential for pandemic preparedness. Since the onset of the COVID-19 pandemic, substantial effort has been directed toward elucidating the entry mechanisms of betacoronaviruses, while members of other genera remain comparatively understudied.

The manuscript by Gallo et al. reports the discovery of CEACAM6 as an entry receptor for a clade of bat alphacoronaviruses. The authors show that human CEACAM6 can be utilized by African members of this clade, whereas related viruses from Asia and Europe can only engage the bat homolog. This is the first study to identify CEACAM6 as a coronavirus receptor and includes structural analysis of the RBD–receptor interaction. The quality of the X-ray crystallography data is of sufficient quality to support the proposed mode of binding.

In addition, the authors propose that the majority of non-human alphacoronavirus spike proteins (37 out of 38 tested) are incompatible with known human CoV receptors (ACE2, APN, DPP4, and TMPRSS2) and cannot infect a range of human cell lines derived from different organs. These findings provide important insights into the barriers to cross-species transmission and the potential zoonotic risk posed by bat alphacoronaviruses.

I am particularly impressed by the breadth of the initial receptor screening and the integration of multiple disciplines, including genetic analysis, biochemistry, infection assays, structural biology, and serology, to explore the implications of CEACAM6 usage by bat alphacoronaviruses. This work represents a significant advance in our understanding of alphacoronavirus diversity, particularly within bats. The findings also underscore the need for follow-up studies to assess the potential for human transmission in regions such as Kenya, and to evaluate whether related bat alphacoronaviruses might similarly exploit human CEACAM6.

Considering the above, I believe this study would be an excellent fit for Nature once the following points have been addressed.

We would like to thank the reviewer for their kind words and their overall strong support for our study. We have addressed their concerns and suggestions point by point below:

Primary point to address

The discovery and validation of CEACAM6 as a novel coronavirus receptor is convincing and I have little, if any, concerns with this aspect of the manuscript. However, I have reservations regarding the first part of the study which claims that the majority of non-human alphacoronavirus spike proteins are incompatible with known human CoV receptors. While the authors present evidence that the majority of tested alphacoronavirus spikes do not enter cells expressing ACE2, APN, DPP4, and TMPRSS2, I wondered whether alternative explanations for the observed lack of infection could be considered. Specifically, I am curious about the potential role of spike processing and conformational dynamics in receptor engagement. There is precedent in the literature for betacoronaviruses and deltacoronaviruses that spike trimers may not bind to a known receptor, or receptor-mimicking antibodies, while isolated RBDs can [1,2]. In the study by Du et al, it was demonstrated that PDCoV S ectodomains could only bind to APN after being heat treated. The same paper also demonstrated that full-length, cell surface-expressed PDCoV spikes could not bind antibodies targeting the RBD, suggesting that some additional biological cue might be required for receptor binding and subsequent fusion. Considering this, can the authors rule out the possibility that some of the spikes examined in their study may adopt conformations that prevent receptor binding in the context of the pseudoviruses, but not in the context of an exposed RBD? ... Overall, I suggest the authors consider whether testing all, or a subset, of RBDs for receptor binding could help clarify these possibilities and strengthen their conclusions regarding receptor incompatibility.

We thank the reviewer for these excellent suggestions. As the reviewer points out, it is becoming clear that many coronavirus RBDs are held in a closed ('3 heads down') conformation, likely to avoid antibody mediated neutralisation 'in the wild'. As a result, it is right to question whether, in the context of a pseudotyped spike, their RBDs could bind, but don't, because additional biological cues are missing in this context. To address this, we cloned, expressed and purified an extensive panel of putative alphaCoV RBDs and used these in RBD-receptor binding assays. Briefly, flow cytometry assays were used to assess RBD binding (via fluorescent antibody-labelling of both the recombinant RBD and the receptor) to a range of cell-surface expressed human, bat and carnivore APN and ACE2. As controls, we included other known human coronavirus receptors, namely TMPRSS2, DPP4 and the recently discovered DPEP1 – a receptor for the betacoronavirus, embecovirus PHEV. We also included two virus positive controls: SARS-CoV-2, which binds ACE2 and canine CoV SD-F3, which binds different canid APNs. Of note, it was not always possible to match the RBDs of the alphaCoV Spikes to the exact host and receptor from, which the virus was isolated. This is often due to a lack of information of the actual bat species where isolation took place or, alternatively, that the receptor sequence is not known (for more detail on this point see the reply below to Reviewer 5). However, we included the closest relatives that we possess and have already tested in the pseudotype entry assay. In all instances, our results from the pseudotype entry assays (Figure 1) were recapitulated in these RBD binding screens, indicating good correlation between binding and entry for the alphacoronaviruses. These experiments strengthen our negative results by confirming that the lack of pseudotype entry is not a result of RBD masking in the context of the full-length Spike pseudotypes and we thank the reviewer for suggesting this experiment. We have referenced these additional experiments in the main text as follows:

"To examine whether closed RBD conformations were leading to false negatives in our screens, we purified a selection of RBDs (Fig.S8a) and assessed receptor binding at the cell surface by flow cytometry. The observed correlation between entry and receptor binding suggested that the absence of pseudotype entry reflects a genuine inability to bind APN and ACE2 (Fig.S8b)."

Figure (see overleaf)

Legend: Figure S8. Binding of a subset of alphaCoV predicted RBDs to known receptors, assessed by flow cytometry. (a) Top: Phylogenetic tree of the alphaCoV library. Viruses with RBDs that were successfully produced recombinantly are highlighted in green, while those we could not obtain are coloured in red. Those viruses highlighted in grey were not tested (no outline) or alternatively the RBD could not be confidently identified by alignment (black outline). Below: Coomassie staining of the recombinant RBDs used in the binding assay. (b) Binding of viral RBDs to relevant APN, ACE2, TMPRSS2, DPP4 and DPEP1 receptors. The numbers by each sample indicate the percentage of PE+ positive cells among the singlets (receptor + population). The numbers in coloured boxes show the percentage of APC+ cells (viral RBD + population) in the PE+ population. At least 25,000 events were acquired per sample.

Related to this, some of the authors anti-HA Western blots used to assess pseudotyping efficiency show double bands, which could suggest differences in spike processing, potentially similar to the molecular weight differences seen in trypsin-treated PDCoV, or FCoV-23 after loss of domain 0 [3,4]. Since pseudoviruses were produced in human cells, is it possible that key proteolytic processing events required for receptor binding or entry might not occur (or might occur aberrantly) in this system? Along those lines, can the authors exclude the possibility that additional host factors, such as proteases or co-receptors, such as observed for HKU1 [5], might be necessary for enabling receptor usage in their screening assays?

Agreed, it is indeed possible that not all our alphaCoV spikes might be fully processed to maturity in transfected cells and, therefore, may lack some final activation steps that would otherwise confer an ability to infect human cells. In our initial submission we included data for co-expression of the cofactor TMPRSS2 – a well-known activator of spike. However, upon reflection we realise the supplemental data (original Figure S7) was not clear. To address we have re-plotted the data to show the individual, rather than collated results (see below). These data show that TMPRSS2 co-expression does not alter the APN or ACE2 receptor usage profile, apart from for btCoV/AT1A-F41. In our first submission we demonstrated that this relative of 229E uses a narrow range of APN receptors, which interestingly is expanded by TMPRSS2, presumably through the lowering of a thermodynamic threshold for particle entry. We have also now performed additional experiments with trypsin, using our library of pseudotyped S. Before transduction, we treated the pseudotypes with 250mg/mL of DPCK-trypsin for 1h at room temperature. We inactivated the trypsin with 10% FBS before adding the pseudoviruses onto the cells overexpressing the receptors. Similar to the TMPRSS2 data we did not observe any difference in receptor usage with the pre-treated pseudovirions. We have changed the main text as follows to address these points:

“To address this, we repeated experiments in the presence or absence of human serine protease TMPRSS2 (Fig.S7), and trypsin (Fig.S2c) but co-expression or pre-treatment, respectively, did not affect host range...”

(Figures overleaf)

Legend: Figure S2c. Treatment of pseudotyped alphaCoV S with trypsin does not affect usage of APN and ACE2 receptors. Pseudoviruses were incubated with 250 $\mu\text{g}/\text{mL}$ of TDPK-trypsin for one hour at room temperature, before infection of HEK293T overexpressing selected APN or ACE2. Cleavage of S to promote entry did not impact receptor usage, or host range (through comparison to results shown in Fig.1b and Fig.1c). Experiments were performed in technical triplicate.

Secondary points to address

- Please explain in more detail in the main text how the greedy algorithm works that was used to decide on which S sequences to test while retaining overall alphacoronavirus sequence diversity. It would be helpful to know which decisions were made to support the claim that the known alphacoronavirus diversity was studied here and to align with the final paragraph of the discussion.

We have now expanded the Materials and Methods section to provide a more detailed explanation of our greedy selection procedure. Our aim was to choose k sequences that provide broad coverage of known alphaCoV spike diversity, under a fixed assay capacity and/or laboratory workload (here $k=40$, representing $\sim 1.5\%$ of the 2714 post-QC sequence dataset). Formally, we maximised the minimum

pairwise distance (MMD) among the k selected sequences. Starting from the initial maximum-likelihood phylogeny inferred from full-length spike protein (branch lengths in substitutions per site), we computed patristic distances (sum of branch lengths) between all pairs of tips. The greedy procedure proceeds then as follows: (i) initialise with the two tips with the largest patristic distance; (ii) iterate by adding the tip whose nearest distance to the current set was maximal; (iii) stop when k tips have been selected. Intuitively, each added sequence is the one most distant from its nearest already-chosen neighbour, which enforces even coverage of sequence space. This construction is nested (the $k+1$ set contains the k set), which is desirable for practical panel growth and ensure stability if resources change. To quantify how well the selected 40 sequences capture known alphaCoV diversity, we have now performed three validations (compared to 10,000 size-matched random panels), indicating that our selection captures >50% of phylogenetic diversity and represents all major clades of the full tree while maintaining uniform phylogenetic coverage:

1. Phylogenetic diversity (PD) coverage: our set of 40 selected sequences capture 53.4% of the total Faith's PD of the full tree, while representing only 1.5% of sequences. By comparison, 10,000 random panels of 40 selected sequences yielded a mean PD coverage of only $13.7 \pm 3.2\%$;
2. Uniformity in phylogenetic coverage: to assess whether any phylogenetic regions are underrepresented, we calculated the distance from each unselected sequence to the nearest selected one. 90.0% of all unselected sequences are within 0.338 substitution/site of at least one selected representative, with a maximum of 6.663 substitution/site distance for a single outlier. Within-set dispersion is also higher than random (minimum pairwise distance of 0.07 substitutions/site, compared to a mean of 0.001 ± 0.0008 for random selections; mean pairwise distance 2.97, compared to 1.07 for random selections);
3. Clade-level representation: using distance-based hierarchical clustering, we identified 6 major phylogenetic clades (minimum clade size $\sim 1\%$ of the dataset) with all represented in our 40-sequence panel. Representation rates across clades range from 0.1% to 22%, with a coefficient of variation of 1.67, reflecting the greater internal diversity of larger clades.

To incorporate this explanation into the main text we have made the following changes:

Materials and methods:

"Briefly, let $\delta(i, j)$ denote the patristic distance between tips i and j on the reconstructed maximum-likelihood tree with branch length estimated in substitutions/sites, and let k be a prior number of tips to be selected. The greedy algorithm (i) identifies the farthest pair $\arg \max_{i, j} \delta(i, j)$ and initialise the final selection set S with these two tips; (ii) for each subsequent selection step, adds the tip $x \notin S$ that maximise its nearest-neighbour distance to the current set $x = \arg \max_{x \notin S} \min_{y \in S} \delta(i, j)$; and (iii) repeats until $|S| = k$. Although heuristic, this approach ensures that an optimal subset of tips (i.e. evolutionary units) is returned under the assumption of maximising both minimum phylogenetic distance and phylogenetic diversity (PD), as theoretically proposed and discussed by Bordewich et al. We report Faith's PD¹² of the induced minimal subtree and benchmark against a 10,000 random panels of matching size."

Results:

"The selected 40-sequence S panel ($\sim 1.5\%$ of the full dataset) captured 53.4% of the total phylogenetic diversity, substantially exceeding size-matched random panels ($13.7 \pm 3.2\%$; 10,000 permutations; ~ 3.9 -fold enrichment; empirical $p < 0.0001$)."

• It appears the algorithm was trained using complete S sequences. Would it have made a difference if it were to be trained using RBD sequences only (arguably a better indicator for receptor usage and hence zoonotic potential)?

To clarify, the selection procedure is deterministic rather than ‘trained’: given a distance matrix, the greedy algorithm returns a set; no parameters are fitted. We chose the full-length spike protein for the primary analysis to: (i) leverage greater phylogenetic signal; (ii) mitigate artifacts from recombination and convergent evolution concentrated in RBD; and (iii) preserve sequence features outside RBD (e.g., S1 NTD, S1/S2 cleavage region, S2 fusion machinery) that can affect entry and antigenicity. Also of note is that any selection of ‘RBD sequences’ for analysis would be sequence alignment-led and rely on assumptions about the general structural organisation of all alphaCoV S proteins. We were, and remain, cautious to do this because for some spikes, e.g. shrew and rat alphaCoV, it is very difficult to predict an RBD. The only realistic option would be to use AlphaFold to predict structures, but we are not confident this will work well for phylogenetic outliers. Since use of the entire Spike sequence avoids any subjective choices regarding domain/subunit boundaries, we consider this to be the most robust and reproducible protocol for sequence selection.

• While the evidence for use of CEACAM6 as an entry receptor by CcCoV KY43 is convincing, it would be good to see that the observed entry of Calu3 and Caco2 cell lines by CcCoV KY43 depends on this membrane protein.

This is a very good point, which was also requested by Reviewer 1. To address this, we used a panel of siRNAs and a scrambled control, or alternatively shRNAs, to human CEACAM6, to validate its role in KY43 entry in both cell lines, Calu-3 and Caco-3 (those that were permissive in our earlier cell-line screening, Fig. 2a) – see Figure 2i below. siRNA- and shRNA-mediated knockdown of human CEACAM6 was confirmed by Western blot and we also showed, in a separate negative control, that entry of the APN-using relative, 229E, was unaffected (Fig S12 below). The following changes to the main text have now been made to include these results.

“To further confirmed the role of human CEACAM6 in CcCoV|KY43 entry we used CEACAM6-specific siRNAs to knockdown endogenous protein expression in the susceptible cell lines Caco2 and Calu3, observing reduced virus entry for CcCoV|KY43 (Fig.2i, left) but not hCoV/229E-inf1 (Fig.S12). The same results were obtained in shRNA-treated Calu3 and Caco2, selected to stably downregulate the expression of human CEACAM6 (Fig.2i, right).”

Legend: Figure 2i. Knock-down of human CEACAM6 in Caco2s and Calu3s reduces CcCoV|KY43.

Human CEACAM6-specific siRNAs, or negative scrambled controls, were electroporated into Caco2 and Calu3 cells and then infected with KY43 (left panel). To confirm CEACAM6 reduction, cell lysates were analysed by immunoblot. For CcCoV|KY43, the average reduction in entry from three independent experiments, each performed with four replicates is shown, normalized to the scrambled siRNA control. Cells were used for entry assay with pseudotyped CcCoV|KY43 S. Significance of \log_{10} fold change was determined using a one-sample *t*-test against a \log_{10} fold change of 0 (no change), with *P*-values were adjusted for multiple comparisons (* for $p < 0.05$). Using lentivirus expressing shRNAs, we produced stable knock-down of Caco2 and Calu3 cell lines to reduce human CEACAM6 expression, and validated it by immunoblot (right panel). One biological replicate performed with technical triplicates is shown.

Legend: Figure S12. Transient knock-down of human CEACAM6 using siRNA does not affect hCoV/229E entry. One day after electroporation, Caco2 and Calu3 cells (the same used in experiments described in Figure 2i) with different siRNA against human CEACAM6 were tested for their permissivity to hCoV/229E-inf1, showing no phenotypical effect. One experiment consisting of technical triplicates is shown.

• I appreciate that the authors do not make overly strong claims about their serology data (Fig. 4), yet it appears to me that zoonotic introduction has taken place in a limited number of individuals. Why is this not discussed in more detail? Are there more likely alternative explanations for this observation? It could indicate that the virus is zoonotically transmitted in the field but is incapable of sustained transmission among humans and it would be good to follow up on this or at least comment on it. Additionally, it would have been preferred to use the CcCoV KY43 full spike ectodomain instead of the RBD as this allows for a better comparison to the controls (229E, NL63 and SARS2) for which full length is used. The signal would presumably be greater in that case and provide a better understanding of the extent of zoonotic transmission that may be occurring in Kenya. This should, at the very least, be mentioned as a limitation of the study.

We agree that high CcCoV|KY43 ELISA signals could, in principle, be consistent with limited zoonotic transmission without sustained human-to-human spread. In the amended main text, we now discuss this possibility alongside alternative explanations, including cross-reactive antibody responses elicited by exposure to antigenically related coronaviruses. To provide wider context for the observed low-frequency KY43 ELISA reactivity, we have now included ELISAs (performed with all 368 sera) from additional, distantly-related, bat-derived alphaCoVs, HIYN18 and A701 (also in our algorithmic library). These antigens were selected to provide additional comparators and to allow assessment of the specificity of sporadic high OD signals. This is important in the absence of well-defined negative-control sera in this emergence setting. Our discovery of similarly infrequent high-OD responses

against HIYN18 and A701 supports one interpretation that rare KY43 signals are likely cross-reactive or non-specific reactivity, rather than widespread zoonotic spillover. However, we also demonstrate, using statistical correlation analyses, that the higher CcCoV|KY43 ELISA signals do not associate strongly with the included coronaviruses, providing evidence for actual spillover. We have modified the main text as follows to discuss this additional experimentation and analyses:

“A total of 368 blood donor samples collected in 2020 and 2021, comprising mainly males (95%) under 35 years of age (72%) living in the Tana River and Taita-Taveta counties where CcCoV|KY43 was identified (Table S8), were examined for CcCoV|KY43, 229E, NL63 and SARS-CoV-2 S-specific responses by ELISA (Fig. 4b, inset, Taita-Taveta county highlighted in pale red). We also included two RBDs from more distantly related alphaCoVs (HIYN18 and A701, both isolated in bats in China) as comparators. Across the full dataset, Spearman’s rank correlation analysis revealed weak monotonic associations between CcCoV|KY43 and 229E, SARS-CoV-2, HIYN18 and A701 ($\rho = 0.14–0.30$), whereas no association was observed with NL63 ($\rho = 0.06$) (Fig. 4b). Although ELISA reactivity against the human alphaCoVs 229E and NL63 was commonly observed, elevated CcCoV|KY43 ELISA signals were detected only sporadically, with OD values exceeding 1 in nine samples. Using this OD threshold as a descriptive marker of high reactivity rather than evidence of seropositivity, similarly infrequent high-OD signals were observed for HIYN18 ($n = 5$) and A701 ($n = 2$). Restricting the analysis to samples comprising the top 10% of CcCoV|KY43 ELISA signals, Pearson’s correlation analysis of matched sera revealed moderate correlations with NL63 and SARS-CoV-2, but only weak correlation with 229E, HIYN18 and A701 (Fig. S25). Together, these data suggest that widespread CcCoV|KY43 spillover is unlikely in these populations. Alternatively, sporadic KY43 reactivity may reflect cross-reactive antibody responses elicited by exposure to antigenically related coronaviruses.”

With regards to the use of full-length CcCoV|KY43 spike ectodomain – we agree this would have provided a more direct comparison with the stabilised full-length spikes used for 229E, NL63, and SARS-CoV-2. However, technical constraints limited our ability to produce sufficient quantities of full-length spike, and the efficacy of proline stabilisation for this novel alphaCoV spike remains uncertain. For this reason, we elected to use the RBD, which is typically the most immunogenic region of coronavirus spike proteins and is therefore well suited for detecting potential exposure-associated antibody responses. The addition of RBD antigens HIYN18, and A701 also now ensures increased consistency among the comparators. We have also acknowledged the use of RBD rather than full-length spike as a limitation of the study in the materials and methods section. Our statistical approaches are also caveated with the statement that our serological analyses are intended to identify patterns of reactivity rather than to infer definitive seropositivity or the extent of zoonotic transmission. The materials and methods has accordingly been modified as follows:

“Since full-length spike ectodomains could not be produced in sufficient quantities for CcCoV|KY43, ELISAs employed stabilised full-length spikes for 229E, NL63, and SARS-CoV-2 and RBDs for CcCoV|KY43, HIYN18, and A701; this represents a limitation of the study and precludes direct quantitative comparison of ELISA signal magnitudes across antigens.”

...

“Spearman’s rank correlations were applied to assess associations across the full dataset, whereas Pearson’s correlations were applied to the top 10% of KY43 ELISA signals to explore co-variation among high responders. Since true negative control sera are unavailable in this emergent-virus setting, OD thresholds were used solely as descriptive markers of high reactivity (within each antigen-specific ELISA) and not to infer seropositivity or compare absolute signal magnitudes across assays.”

Legend (right): **Figure 4b.** Human sera from volunteers (n=368) in Tana River and Taita-Taveta counties (highlighted in pale red in the inset map), were analyzed for their reactivity to different human coronavirus glycoproteins. Individual Spearman's rank correlations (ρ) for each data set (compared to CcCoV|KY43) are provided.

Legend (left): **Figure S25. Correlation of high CcCoV|KY43 ELISA reactivity with responses to other coronavirus antigens.** Top: ELISA optical density (OD) values for all samples within the upper 10% of the KY43 distribution. Below: Pearson's correlation analysis was

performed on this enriched subset of 'high' KY43 samples to assess co-variation. Each point represents an individual serum sample tested against matched antigens. Correlation coefficients (r) are shown for each comparison. Of note, the top 10% threshold was used as a descriptive enrichment criterion and not to infer seropositivity or compare absolute signal magnitudes across assays.

• Please provide data on CEACAM6 Q123H mutation and CEACAM5 H123Q, perhaps in combination with the discussed I63F mutation. Alternatively, it would be interesting to see what changes in the viral RBD would allow functional binding to CEACAM5. Given the similarity between CEACAM6 and 5, it is plausible that minor changes would already allow for this or even that the higher affinity CcCoV 2A variant can also use CEACAM5 as a receptor.

We have now examined additional mutations within the CEACAM receptors, in particular H123Q in CEACAM5 and both Q123H and the combinatorial mutant I63F and Q123H in CEACAM6. To summarise, the changes at 123 appear less important than those at 63, which appears to be the main determinant of viral restriction. We have included this data in a new supplemental figure (Figure S20 – see below) and modified the text accordingly as follows:

“Alignment of the surface contact interface identified two important amino acid substitutions, I63F and Q123H, in CEACAM5 relative to CEACAM6 (Fig.3d). A F63I, but not H123Q, substitution in CEACAM5 partially overcame the CEACAM5 entry restriction of CcCoV|KY43 pseudotypes, with the corresponding substitutions in CEACAM6, I63F, Q123H or both, reducing pseudotype entry (Fig.3d, Fig.S20a); the mutated receptors being expressed comparably to wild type (Fig.S20b).”

Also, in response to the reviewer's questions about mutating the RBD to use human CEACAM5, which they suggested as an alternative approach we would like to draw their attention to our reply to Reviewer 1, wherein we examined in finer detail the mechanistic nature of human vs non-human CEACAM6 usage (Fig.S29). We hope that the reviewer considers this of equal interest and a good surrogate data set.

Legend: **Figure S20. Additional amino-acid substitutions in human CEACAM5 and CEACAM6 confirm the major contribution of position 63 in receptor specificity.** (a) Entry assay using pseudotyped CcCoV|KY43 S on HEK293T transiently transfected with human CEACAM5 and CEACAM6, or their mutated forms. The double substitution I63F and Q123H shows

confirms that position 63 is the determinant of CEACAM usage for KY43. (b) Flow cytometry data shows the comparable expression of human CEACAM5 and CEACAM6 with their derived mutants in HEK293T.

• Given the usage of CEACAM1a as a receptor by MHV, could you please include any thoughts in the discussion as to why CEACAM members might make for excellent CoV receptors?

We thank the reviewer for this thought-provoking question. We have added text to the Discussion as follows to explain why we believe CEACAM proteins make excellent CoV receptors:

Many viruses utilise the N-terminal V-set Ig-domain of cellular adhesion molecules for cell attachment¹³, including MHV which binds CEACAM1¹⁴. The high abundance of CEACAM family proteins on apical surfaces of mucosal epithelia make them highly suitable for pathogen attachment. Indeed, *Candida albicans*¹⁵ and several Gram-negative bacteria¹⁶ utilise CEACAM proteins for adhesion. Given their Ig-domain architecture and abundance on barrier membranes, we hypothesise that additional CoVs will employ CEACAM family proteins for cell entry.

Minor comments

- Could you explain why some entry is observed for pseudovirus with NL63 spike in presence of human APN (Fig 1)?

To perform our receptor entry assays, we use HEK293T because: they transfect well, are human, they correctly glycosylate proteins, and lastly, robustly express proteins. However, they are known to express a low level of endogenous human ACE2 which likely explains the NL63 entry experiment. These signals are quite close to background levels, which potentially explains why it is not seen in the DPP4 or TMPRSS2 experiment. The clear distinction between this background signal (around 5×10^4 RLU), and the luciferase signal in the presence of permissive ACE2s (10^7 RLU) gives us confidence we can distinguish between the two signals.

- Line 40: change alphacoronavirus to alphacoronaviruses.

We have corrected this in the text, thank you.

- Line 61: please change to O-acetylated sialic acids. Also, HKU1 should be included along with OC43 as O-acetylated sialic acid as it is equally important as TMPRSS2 for fusion (see Figure 1H in Fernandez et al) [6].

This has now been corrected in the introduction.

- Line 60-61: You are missing references to support this statement [7–10]. Please check throughout the manuscript to ensure the relevant literature is cited.

We have now added relevant references throughout the manuscript, for example in the introduction referencing the papers that first demonstrated receptor usage, e.g.

“Human CoV NL63 is the only known alphaCoV to use ACE2 as a receptor¹⁷. Aside from ACE2 (SARS-CoV-1/2), human betacoronaviruses also utilise the receptors DPP4 (MERS-CoV)¹⁸, O-acetylated sialic acids (HKU1¹⁹ and OC43²⁰) and TMPRSS2 (HKU1)²¹. Recently, porcine and human DPEP1 were both shown to be efficient receptors for porcine hemagglutinating encephalomyelitis virus (PHEV) although there is no evidence for zoonotic infection¹. Entry of the betacoronavirus murine hepatitis virus (MHV) is supported by murine CEACAM1², although in this case the human orthologue is not efficiently used³”

- Line 68: please change domains to subunits.

This has been corrected in the text, thank you.

- Line 95-96: please include references to existing literature that disproves APN as an entry receptor for PEDV (i.e. critical for fusion of viral and host membranes).

We have now added reference to a study published this year where the role of porcine APN is discussed in detail with regards to entry of different porcine coronaviruses (PEDV, TGEV, SADS)²². This work builds on previous papers that provide evidence of the facultative role of APN in PEDV entry, both *in vitro*^{23, 24} and *in vivo*²⁵. Since we observed successful expression of porcine APN by flow cytometry and its efficient use in entry assays by PRCV, as well as incorporation of PEDV/Colorado S into pseudoviruses (Fig S2), our conclusion is that our system – transient expression of APN in non-permissive HEK293T - is not compatible to the study of PEDV receptor usage. We have modified the main text as follows:

“Despite a feline CoV being included in the library, it did not pseudotype, while two different PEDV S failed to use porcine APN in our experimental settings, which can be explained by the dynamics of PEDV S internalization in non-susceptible cells²²”

- Line 106-108: ACE2 and APN of non-human origin studied but not DPP4 or TMPRSS2, please specify statement accordingly.

Apologies, you are correct. We have modified the statement to reflect that our conclusions are specific to the alphaCoV receptors ACE2 and APN, not coronavirus receptors used by the wider family, as follows:

“These findings suggest that use of the known receptors ACE2 and APN may be a relatively rare phenomenon within the alphaCoV genus.”

Referee #3 (Remarks to the Author):

I co-reviewed this manuscript with one of the reviewers who provided the listed reports.

We thank the reviewer for their input and hope their suggestions elsewhere have been addressed.

Referee #4 (Remarks to the Author):

I was asked to review the phylodynamic analysis in this manuscript, which I take to also include the details needed to eventually run the analysis and interpret its results (e.g., data curation). As it pertains to the phylodynamics, the authors performed Bayesian phylodynamics on spike sequences using a protein evolution model and concluded that usage of CEACAM6 is a relatively recent acquisition. My comments will be primarily about this part

of the work.

Firstly, thanks to the reviewer for their specific input and comments on the phylodynamics, which we have now addressed with additional experimental analysis, as follows:

1. Neither the alignment, BEAST XML, nor the resulting log/tree files were provided. At a minimum, the alignment and XML should be provided for reproducibility.

We apologise to the reviewer for not having made available the files generated during our phylogenetic analyses. We have now created a Zenodo repository that hosts all the relevant files, along with Supplementary Materials and additional data rearranged from the initial submission. Until publication a temporary link to that directory is available here:

<https://zenodo.org/records/17951484?preview=1&token=eyJhbGciOiJIUzUxMiJ9.eyJpZCI6ImVhNjc5MmEwLWw1NjgtNDk5Yi1iZjg1LWYxMzc1N2E1ODRjMSIsImRhdGEiOnt9LCJyYW5kb20iOiJkZWRhNzgwY2U4NjYxYWQzNDM5MTQ2YzlmMjRkMWRhMCMJ9.IR8Vkb14Gp0NojUJs79g4QeurWj2bsQbXO61-1sqtkWU1iux6zQJtbqUHRFxb10er5lqvVxl3IWTnozUGUJKg>

2. A maximum likelihood phylogeny of all the initial sequences from “Construction of gene libraries” should be provided, highlighting the 40 spike protein coding sequences ultimately selected.

Hopefully, the reviewer would concur with us that a phylogeny with >2000 tips is difficult to render in a way that would be clear for a user to read from a printed article (especially when only 40 tips need to be highlighted out of the full tree). This was the main reason why we did not include the initial maximum-likelihood phylogeny generated using the full set of alphaCoV spike sequences from NCBI in our figures. However, we agree with the reviewer that this information would benefit readers for any subsequent analysis they might be interest in doing, therefore we have now included a new Supplementary Figure (Fig.S32, *below*) showing the initial maximum-likelihood tree. We have also included the corresponding .nexus file in the Zenodo repository. This is referenced in the main text, as follows:

“Maximum-likelihood phylogenetic reconstruction was performed in IQTREE 2.3.4²⁶ with 1000 ultrafast bootstrap replicates (UFBoot) and 1000 SH-like approximate likelihood ratio tests (aLRTs)²⁷ (Fig.S32).”

Legend: Figure S32. Global diversity of alphacoronavirus spike genes and coverage of the experimental panel. Maximum-likelihood phylogeny inferred with IQ-TREE 2.3.4 from 2,714 full-length spike coding sequences, retrieved from the Virus Pathogen Database and Analysis Resource (ViPR) platform, under a codon-partitioned substitution model. The 40 greedily selected spike sequences used for functional assays are indicated by orange tip labels. Branch lengths are in units of expected substitutions per site on the x axis, as estimated by IQ-TREE.

3. Subsequently, the authors could construct a nucleotide-based maximum likelihood phylogeny of the resulting 40 selected strains coupled with the sequences belonging to the local phylogeny of CcCoV|KY43 (these local sequences could also be highlighted on the larger phylogeny in point 2). When doing so, they should infer the support values at internal nodes. This could inform our understanding of how frequently the usage of CEACAM6 arose.

We have now reconstructed a further maximum-likelihood tree using the nucleotide-based alignment of the 40 GMMD selected sequences using codon model in IQTREE. We reported the bootstrap support of internal nodes when these are >75%. This new figure (shown below) is now included in the Zenodo repository, along with the .nexus file of the tree.

4. Nucleotide sequences are available for at least the samples in the local alignment, if not all the samples in this dataset. The Bayesian analysis should therefore use a nucleotide or codon substitution model, rather than an amino acid substitution model. An amino acid model cannot account for synonymous substitutions, thereby potentially resulting in an inaccurate inference of when CEACAM6 usage was acquired.

We agree with the reviewer that a nucleotide/codon framework might be preferable for time-scaled inference, and we have now run BEAST analyses using codon-aware models. To compare different codon models implemented in BEAST, we have run both the SRD60 and the Yang96 models, obtaining similar results. These are further consistent with the initial results obtained using the WAG amino acid substitution model, making the latter as robust as the codon-based ones. We provide here below a summary of the main evolutionary estimates obtained across these different models:

Model	tMCRA root (95% HPD)	tMRCA taxon (95% HPD)	mean UCLD (95% HPD)
WAG (aa)	1304 (1094 to 1488)	1809 (1747 to 1883)	5.51×10^{-3} (3.99×10^{-3} to 7.12×10^{-3})
SRD06 (nt)	1386 (1197 to 1611)	1833 (1794 to 1884)	4.84×10^{-3} (3.68×10^{-3} to 6.28×10^{-3})
Yang96 (nt)	1204 (869 to 1548)	1809 (1747 to 1884)	5.47×10^{-3} (3.71×10^{-3} to 7.45×10^{-3})

We have now updated Fig.5A by including the phylogeny reconstructed using the SRD60 model and reported statistics using the posterior results generated from that analysis. To reflect these changes in our approach we have changed the main text as follows:

"To explore the evolutionary history of CEACAM6 receptor usage, we assembled a library of S proteins from related viruses, reconstructing the "local phylogeny" of CcCoVs on the alphaCoV tree (Fig.S18, Table S9). Overall, CcCoVs were phylogenetically placed in a sister clade to alphaCoVs isolated from *Rhinolophus ferrumequinum* bats in South China (Fig.5a). The CcCoV clade was returned as monophyletic in all sampled posterior trees ($n=9000$; 95% binomial CI 0.999 – 1.000), with its most recent common ancestor estimated to have evolved circa 1833 (95% HPD 1794 to 1884)."

Legend: **Figure 5a.** Evolutionary reconstruction of the S gene for selected alphaCoV demonstrates relatively recent acquisition of CEACAM6 usage. Inset map: Location of isolation of indicated CcCoV-related viruses.

5. What is the posterior support for key internal nodes in the tree in Fig. 5A, particularly for the most recent common ancestor (MRCA) of the local alignment? If there is low posterior support at the MRCA of this node, the tMRCA of this particular node might not be precisely what the authors are hoping to infer. Rather, they can parameterize their analysis by defining the key taxa as a taxon set in BEAUti and specifically infer the tMRCA of that taxon set. When doing this analysis, they should also examine how frequently this taxon set is monophyletic (importantly, this taxon set should not be specified as monophyletic a priori in BEAST).

In the updated phylogeny shown in Fig.5A, we now report posterior clade probabilities for the internal nodes and, to provide a clearer visualisation, we display only those internal nodes with an estimated posterior probability of >0.75 (highlighted in blue). To target 'local phylogeny' of CEACAM6-adapted strains, and as suggested by the reviewer, we ran an additional analysis in which these key taxa were specified as a taxon set in BEAUti and the tMRCA of that taxon set was estimated in BEAST without imposing monophyly. This yielded an estimated tMRCA for the 'local phylogeny' of 1833 (95% HPD

1794 – 1884). We additionally quantified how often this taxon set is monophyletic across the full posterior by calculating the proportion of posterior trees in which the ‘local phylogeny’ forms a clade: the focal set is monophyletic in all sampled posterior trees after burn-in (N=9000, 95% binomial CI 0.999 – 1.000) – i.e. >99.99% posterior probability of monophyly. This analysis confirms that the lineage we refer to as the ‘local phylogeny’ is very strongly supported as a clade across the posterior, and that the reported tMRCA indeed corresponds to this predefined set of CEACAM6-adapted viruses rather than to a single arbitrarily chosen internal node. We have amended the main text as follows to acknowledge these changes:

*"To explore the evolutionary history of CEACAM6 receptor usage, we assembled a library of S proteins from related viruses, reconstructing the ‘local phylogeny’ of CcCoVs on the alphaCoV tree (Fig.S18, Table S9). Overall, CcCoVs were phylogenetically placed in a sister clade to alphaCoVs isolated from *Rhinolophus ferrumequinum* bats in South China (Fig.5a). The CcCoV clade was returned as monophyletic in all sampled posterior trees (n=9000; 95% binomial CI 0.999 – 1.000), with its most recent common ancestor estimated to have evolved circa 1833 (95% HPD 1794 to 1884)."*

6. Fig. 5A suggests YN2020 and Kuddep are part of the local phylogeny, but the focal MRCA is not ancestral to these two. Should these two samples therefore not be considered part of the local alignment? Additionally, Kuddep appears to use CEACAM6 as well, which raises the possibility that this trait was acquired or lost multiple times. This more complex evolutionary history should be briefly discussed. As it relates to point 5, the authors should consider whether Kuddep should be part of the taxon set and interpret the subsequent results in that context.

We have now clarified the definition of the ‘local phylogeny’ and reassessed the placement of YN2020 and Kuddep. Because the ‘focal MRCA’ is not ancestral to these two sequences, we excluded them from the primary taxon set used for the tMRCA analysis (as reported above). However, we also performed sensitivity analyses including them and obtained similar qualitative conclusions. To investigate the potential acquisition (or loss) of the CEACAM6 usage along the full evolutionary history of alphaCoV, we further explicitly modelled CEACAM6 usage as a discrete trait in a Markov-jump framework to assess whether the data support single versus multiple gains (or losses). The posterior distribution of CEACAM6 transitions supports ≥ 2 independent gains (median number of gains = 2; 95% HPD 1 - 3), whereas losses are rare and are not mapped on the resulting HIPSTR tree (median 0, 95% HPD 0 - 1). Consistent with this, the posterior probability that the ‘focal MRCA’ used CEACAM6 is 0.97. In the reconstructed history, CEACAM6 usage in Kuddep arises on the branch leading to the MRCA of a sister clade to the ‘local phylogeny’, while the MRCA of these two sister clades itself is reconstructed as not using CEACAM6. This pattern is most parsimoniously explained by two independent acquisitions of CEACAM6 usage, one along the lineage giving rise to the ‘local phylogeny’ and one along the lineage leading to the YN2020 and Kuddep strains. To acknowledge these additional results, we have modified the main text as follows:

“To investigate the potential acquisition (or loss) of the CEACAM6 usage along the full evolutionary history of alphaCoV, we further explicitly modeled CEACAM6 usage as a discrete trait in a Markov-jump framework to assess whether the data support single versus multiple gains (or losses). The posterior distribution of CEACAM6 transitions supports ≥ 2 independent gains (median number of gains = 2; 95% HPD 1 - 3), whereas losses are rare and are not mapped on the resulting HIPSTR tree (median 0, 95% HPD 0 - 1). Consistent with this, the posterior probability that the ‘focal MRCA’ used CEACAM6 is 0.97. In the reconstructed history, CEACAM6 usage in BtRsCoV/Kudep arises on the branch leading to the MRCA of a sister clade to the ‘local phylogeny’, while the MRCA of these two sister clades itself is reconstructed as not using CEACAM6. This pattern is most parsimoniously explained by two independent acquisitions of CEACAM6 usage, one along the lineage giving rise to the ‘local phylogeny’ and one along the lineage leading to the BtRsCoV/YN2020 and BtRfCoV/Kudep strains.”

For reviewer’s information: **Legend: Discrete-trait reconstruction of CEACAM6 usage on the alphacoronavirus spike phylogeny.** Maximum clade credibility time-scaled tree from the BEAST Markov jump analysis, in which CEACAM6 usage was modelled as a binary discrete trait. Branch colours indicate the inferred state (CEACAM6-utilising vs non-utilising), and circles on branches mark inferred transition events (gains and losses) counted under the Markov jump framework.

Minor comments:

- Lines 214–221. Please also report nucleotide identities.

Nucleotide identities have been added, as follows:

“Interestingly, both viruses are relatively divergent; while CcCoV|2B is more related to KY43, it shares only 78.82% and 83% nucleotide and amino acid identity, respectively, within S (79% and 85% within the RBD). The more distantly related isolate, 2A, shares only 70% and 77% nucleotide and amino acid identity, respectively, across S (72% and 76% for the RBD) with CcCoV|KY43 (Fig.4c).”

- Supplementary Figures S4, S5, etc.: It could be helpful to list pathogens that were not tested, even if this results in rows of “X”s, for completeness.

Thank you for the suggestion. In the supplemental figures we have now included results from S protein and receptors that were and were not tested (see below: amended Figure S5 and S6).

Legend: Figure S5a. Entry screening of the APN library using alphaCoV pseudotyped S library.

HEK293T transiently expressing APN from varied mammals were infected with pseudoviruses bearing the indicated alphaCoV S on their surface. Raw pseudovirus (PV) entry data, where background has been subtracted, is shown as log₁₀ relative light units (RLU). Experiments were performed in technical triplicate, and positive results validated with two additional independent experiments.

Legend: Figure S6a. Entry screening of the ACE2 library using alphaCoV pseudotyped S library.

HEK293T transiently expressing ACE2 from different mammals were infected with pseudoviruses bearing the indicated alphaCoV S on their surface. For hCoV/NL63, a wider panel of ACE2 receptors was screened (as shown). Raw pseudovirus (PV) entry data, where background has been subtracted, is shown as log₁₀ relative light units (RLU). Experiments were performed in technical triplicate, and positive results validated with two additional independent experiments.

Supplementary Figure S6: It could be useful to showcase an analogous analysis for hCoV-229E:inf-1, for completeness.

We have now included the data for hCoV-229E/inf-1, using human APN, side-by-side with hCoV-229E/ITA. We have referenced this in the main text as follows:

“The algorithm selected 229E S was not functional, and we replaced it with the reference sequence for the virus (Fig.S4a).”

Legend: Figure S4. hCoV/229E-inf1 and PRCV-ISU1 pseudotyped S can use human and porcine APN, respectively.

(a) Entry assay demonstrating that hCoV/229E (strain: ITA, left panel) pseudotyped S does not enter HEK293T cells over-expressing human APN. 229E-ITA S was subsequently replaced with the S of hCoV/229E (strain: inf-1,) which successfully entered cells overexpressing human APN (right panel). A representative experiment performed in technical triplicates is shown, along with SD.

(b) Porcine respiratory coronavirus, strain ISU1 (PRCV-ISU1) S pseudotypes can specifically enter HEK293T transiently transfected with a plasmid encoding porcine APN, but not human APN. This is despite the porcine APN receptor not supporting entry for two PEDV strains

selected by the greedy algorithm. Raw data from two entry assay performed in technical triplicate is shown, along with SD.

- Supplementary Figure S18: should CcCor|2A and CcCor|2B be CcCoV|2A and CcCoV|2B?

Thank you for spotting this, we have now corrected the legend.

- Coronaviruses, including sarbecoviruses, are known to exhibit the time dependent rate phenomenon (TDRP), which can further bias time to the MRCA (tMRCA) estimates toward more recent dates. The ability to use CEACAM6 could therefore have been acquired earlier than the authors suggest. Although the examination of TDRP is outside the scope of the manuscript, it should at least be mentioned in the discussion or limitations.

We acknowledge and agree with the reviewer that, because of TDRP, time-of-divergence estimates based only on short-term molecular clock rates can be biased toward more recent dates. In the context of our study, this means the inferred tMRCA for alphaCoV – and by extension, the timing of when it acquired the ability to use the CEACAM6 receptor – could be older than our estimates suggest. We acknowledge the reviewer’s point that if alphaCoV follow a slower long-term clock, the adaptation to use CEACAM6 might have occurred well before the nominal tMRCA calculated under a simple clock model. Indeed, TDRP could have caused our dating analysis to underestimate the true evolutionary age of this trait. While a full investigation of TDRP (e.g. using epoch or heterogenous clock models) is beyond the scope of the current manuscript, we have added text to address this limitation. We note that viral molecular clocks can decay over time and cite studies of coronaviruses showing far deeper evolutionary roots when accounting for rate variation. We emphasize that our timeline is therefore a conservative (recent-biased) estimate, and the ability to utilize CEACAM6 may have been acquired earlier than we initially inferred. We have now added few considerations about the limitations of our non-TDRP approach in the text, as follows:

“Time-dependent rate effects in viral molecular evolution suggest that our date estimates for the CEACAM6-adapted lineage are likely to be conservative (i.e. biased toward more recent times). Across diverse viruses, substitution rate estimates are systematically higher over short timescales and decay as the measurement interval increases, consistent with a general time-dependent rate phenomenon²⁸. Mechanistic work further implicates purifying selection and multiple-hit saturation as key drivers of this bias, causing long-term evolutionary change to accumulate more slowly than expected under a simple, constant-rate clock²⁸. For coronaviruses specifically, models that explicitly accommodate variable selection pressure and substitution saturation push the inferred origin of the coronavirus radiation from $\sim 10^4$ years to an ancient timescale of millions of years, in much closer agreement with the diversification of their hosts²⁹. While a full investigation of the time-dependent rate phenomenon (e.g. using epoch or heterogenous clock models³⁰) is beyond the scope of this study, we acknowledge that the tMRCA estimates reported here should be interpreted as lower bounds and that the acquisition of CEACAM6 usage may predate our point estimates.”

Referee #5 (Remarks to the Author):

In this paper Gallo et al used a computational approach (greedy algorithm) to select a limited but phylogenetically wide range of alphacoronavirus spike proteins for study. Including controls, 40 spike proteins were examined - 27 of which were from bats. A lentiviral spike protein pseudo typed entry assay was used to assess entry and receptor usage. Tested were various cells lines and cells expressing 25 APNs and 34 ACE2s (the two known alphacoronavirus receptors) from diverse species. In addition, a protein array screening approach was used and a new alphacoronavirus receptor was identified.

The main conclusions can be summarized as follows: i) a limited number of the alphacoronavirus spike proteins tested use APN or ACE2 as receptors and ii) the bat alphacoronavirus CcCoV/KY43 can use human CEACAM6. Given these results, the authors speculate that there must be other alphacoronavirus receptors that have not yet been identified.

Given the potential for spillover to humans the identification of a bat coronavirus that can use human CEACAM6 as receptor is an important finding from a pandemic preparedness and human health perspective. It also broadens our knowledge of coronavirus receptor usage, specifically for alphacoronaviruses, a genus less well studied than that of betacoronaviruses. As such, the work will be of broad interest in the fields of virology and public health.

Firstly, we would like to thank the reviewer for their support and enthusiasm for our research and for their insightful comments and experimental suggestions. We have revised the document in line with their recommendations and performed additional experiments, as described below in our point-by-point response:

However, the following need to be considered/addressed:

1) The main concern is the conclusion that very few of the viruses whose spike proteins were tested use APN or ACE2 as receptors. This is stated in the abstract and at a number of places in the paper. It is also possible that these viruses are highly specific for the APN or ACE2 of the species they infect and that those receptors were not tested. It must be made clear what percentage were tested against the APN or ACE2 from the species they infect (see line 86). The authors acknowledge that some may be "hyper-specialised" and in a paper in BioRxiv (Gallo et al 2024), containing some of the data in this paper, they describe the CcCoV's as specialists. Is it not possible that the majority of the bat viruses tested are specialists? This must be explained/discussed.

We agree that hyper-specialisation, by which we mean a virus can only the receptor from its natural host and nothing else, might be a confounding factor in concluding ACE2/APN usage profiles. However, we were conscious of this from the very start and carefully curated our receptor library to mitigate these concerns, albeit within the limitations of the available data. To summarise this directly, we have now compiled matrices (Fig.S4b and Fig.S5b for APN and ACE2, respectively – see below) wherein for each selected spike we include: the organism where the virus was identified (if specified or known) and whether the corresponding APN or ACE2 sequence is known. If the sequence was known but we did not synthesise it, we also state the percentage identity (at the amino acid level) of the closest related receptor in our screen (and the species identity).

To summarise this work, from the 36 spikes that pseudotyped, the original host was specified for 28/36 viruses (77.8%). Among these 28, the receptor sequences from the matched host were available for only 14/28 APNs (50%; left column) and 22/28 ACE2s (78.6% right column). We successfully matched

the viral Spike with its host receptor in our screen for 10/14 APNs (71.4%; left column, green boxes) and 9/22 ACE2s (40.9%; right column, green boxes). For the 4/14 APNs where this was not achieved, we specify the percentage identity of the most closely related receptor (left column, blue boxes) and the same for the 13/22 ACE2s (right column, blue boxes). When dropping the receptor similarity threshold from 100 to >85% amino acid identity, 93% (APN) and 82% (ACE2) of viruses were screened against a matched or close relative, where one is known. We have modified the discussion to reflect these limitations in our study as follows:

“One limitation of our study is that only 28/36 pseudotyped viruses have a single species assigned as a host, and for these 28 species the corresponding receptor sequences are known in only 50% (APN) and 79% (ACE2) of cases. We directly matched 10/28 and 9/28 of viruses to their cognate host’s APNs and ACE2s, respectively, increasing to 13/28 and 18/28 when the identity threshold for receptors is dropped to >85%, yet only 5 pseudotypes used APN and only 1 used ACE2 (Figs S4, S5). While we cannot formally discount the hypothesis that some bat alphaCoVs are hyper-specialised to only one cognate bat APN or ACE2 that was not included in our libraries, our results strongly suggested the presence of other receptors.”

Legend: **Figure 5b:** For each virus in our library, we report whether the sequence of the APN protein from the respective animal host species is known (numbered boxes) or unknown (grey boxes; NB: either host species was not listed or the APN sequence for a known host species is not available). ‘100’ reflects when a matched APN was used in screening, values <100 reflect the % identity of the closest relative APN used in our screen.

Legend: **Figure 6b:** For each virus in our library, we report whether the sequence of the ACE2 protein from the respective animal host species is known (numbered boxes) or unknown (grey boxes; NB: either host species was not listed or the ACE2 sequence for a known host species is not available). ‘100’ reflects when a matched ACE2 was used in screening, values <100 reflect the % identity of the closest relative ACE2 used in our screen.

2) The entry assay is also a potential source of error as it is possible that many of the spike proteins do not give functional pseudo types. It seems that positive controls - feline CoV and two different PEDV S proteins - did not show entry (see lines 94-97). How do the authors rationalize this? The authors checked spike incorporation into the purified pseudo viruses, but this does not establish that they are in the prefusion conformation or at levels sufficient for membrane fusion. These concerns must all be addressed.

Thank you for your suggestions on how to improve the manuscript. Establishing the analysis pipeline in this way, grounded in an algorithmic approach and treating all viruses equally (to some degree) was done purposefully to avoid biasing our research towards viruses with an already established tropism. That being said, the two initial PEDVs and the feline CoV chosen by the algorithm were among the limited number that did not express well on pseudotypes (Figure S1 and S2). As described in the material and methods and Figure S28 we established a selection pipeline to pick potential replacements. This led, as described in the main text, to us introducing another PEDV which did pseudotype (Figure S1/S2); however, this did not show demonstrable use of porcine APN. As discussed above in response to Reviewer 2 there is some disagreement as to the role of APN in PEDV entry. Previous papers provide evidence of a facultative role for APN in PEDV entry, both *in vitro*^{23, 24} and *in vivo*²⁵. Since we observed successful expression of porcine APN by flow cytometry and its efficient use in entry assays by a surrogate porcine coronavirus PRCV, as well as incorporation of PEDV/Colorado S into pseudoviruses (Fig S2), our conclusion is that our system – transient expression of APN in non-permissive HEK293T - is not compatible to the study of PEDV receptor usage and we have modified the main text accordingly, as follows:

“Despite a feline CoV being included in the library, it did not pseudotype, while two different PEDV S failed to use porcine APN in our experimental settings, which can be explained by the dynamics of PEDV S internalization in non-susceptible cells²²”

Regarding the pre-fusion conformation of the spike, our immunoblots (Fig S2) show the majority of spikes proteins running at a high molecular weight (Mw), typical for highly glycosylated viral glycoproteins. This contrasts to SARS-CoV-2, where the furin cleavage site typically yields additional lower Mw bands for S1 and S2 in immunoblots. Briefly, the S1-S2 cleavage and subsequent S2' activation by TMPRSS2 allows triggering of the fusion peptide and enables transition of the protein to its more post-fusion state. Given this knowledge and the high Mw of most spikes in our purified pseudoparticle immunoblots, we are confident most of the alphaCoV spikes in our screen were in their pre-fusion state. To acknowledge the need for more detailed examination of the molecular weights of the S proteins we have now provided the complete immunoblots images for the data in the Zenodo data repository (see below).

The last point on whether there is sufficient spike to initiate membrane fusion is a very interesting one. HIV pseudoparticles likely non-specifically trans-incorporate viral glycoproteins into their surface. Comparing the results from the purified pseudoparticle immunoblots (Fig.S2) with the screen results (Fig.1) shows that in instance where we see expressed protein and entry (namely NL63, CCoV SD-F3 and A76, 229E, btCoV-AT1A-F41 and btCoV-WA1087) we do not see a strong correlation between

the two. For example, expression of NL63 and AT1A-F41 is relatively low in comparison to WA1087 and the two CCoV, but all show robust entry. Given this observation, combined with the non-specific incorporation of glycoproteins by lentiviruses, it is difficult to establish how we could set a minimum threshold needed for membrane fusion. We feel our current criteria, i.e. seeing detectable protein in an immunoblot of purified pseudoparticles is a reasonable, albeit qualitative, cut off criteria. To acknowledge this point of discussion in more detail we have made the following changes to the materials and methods:

“Of note, for those spikes where we saw entry (Fig.1; NL63, CCoV SD-F3 and A76, 229E, btCoV-AT1A-F41 and btCoV-WA1087) protein expression in the purified pseudoparticle immunoblots (Fig.S2) did not quantitatively correlate with entry signals, indicating it is difficult to establish a minimum threshold of spike incorporation needed for membrane fusion.”

3) A lack of spike cleavage is another factor that may be leading to false negative entry results with the various APN and ACE2 cell lines. How well conserved is the S2' cleavage site among the spike proteins tested? The authors have included TMPRSS2 in the entry assay, but they show that it made no difference in the few cases where entry was observed. How do the authors interpret that observation? Verifying that the purified pseudoviruses can be cleaved by TMPRSS2 (as described by Dufloo et al, ref 9 in this paper) would provide greater confidence in the entry results and strengthen the claim that most of the viruses tested do not use APN or ACE2. This would still not rule out the possibility that they are all specialists. Trypsin is often used in these entry assays, and it too might be tested.

To address a role for trypsin in increasing infectivity of our pseudotyped S proteins we performed additional entry assays on a subset of our spikes and APN and ACE2 receptors, and also included human TMPRSS2 and DPP4 as additional controls. Before transduction, we treated the pseudotypes with 250mg/mL of DPCK-trypsin for 1h at room temperature. We subsequently inactivated the trypsin with 10% FBS before adding the pseudoviruses onto cells overexpressing the designated receptors (receptor usage assays – Fig.S7) or lysed them for Western blot (Fig.S2b – uncropped images of full membrane are uploaded to Zenodo). In this case we did not observe any difference in receptor usage with the pre-treated pseudovirions. Of note, we could not find mention of TMPRSS2 cleavage assays in the Dufloo paper, but we have re-drawn the TMPRSS2 co-expression data to make it clearer (new Fig.S7). We have addressed these additional experiments in the main text as follows:

“To address this, we repeated experiments in the presence or absence of human serine protease TMPRSS2 (Fig.S7), and trypsin (Fig.S2c) but co-expression or co-treatment, respectively, did not affect host range, although there was some evidence of digestion of spike proteins by trypsin (Fig.S2b).”

Legend: Figure S2b Pseudoparticles were pre-treated with TDPK-trypsin for one hour at room temperature, before purification and immunoblot analysis, as described in (a).

Legend: Figure S7. APN and ACE2 receptor screening in the presence of human TMPRSS2. HEK293T overexpressing libraries of APN or ACE2 from various mammals, as indicated, plus human TMPRSS2 were infected with the alphaCoV library of *S* pseudotypes. Expression of hTMPRSS2 did not impact the overall pattern of receptor usage. Initial screening was performed in technical triplicate, and positive results were validated in two additional independent experiments.

4) The authors have determined two crystal structures that provide a detailed description of the residues involved in binding CEACAM6. The results should be better discussed in relation to the other results presented. Among the 40 spikes examined, do any of the others contain similar loops/residues (eg W600)? The authors have also expanded the phylogenetic tree around CcCoV/KY63 - are the important residues conserved? If not - does this provide further insight into what defines the critical interactions? An expanded CEACAM6 screen was also performed - are the critical CEACAM6 residues conserved? I think the RBDs are too variable to rationalize that residues or loops alone can confer CEACAM6 usage.

Firstly, to have an idea of the similarity of KY43 to the CTD of the other alphaCoVs, we aligned the predicted receptor binding domains and calculated the percentage identity, showing low similarity in their sequences (Fig.S19b – see below). Secondly, we aligned the predicted RBD in the CTD of S1 to identify epitopes suggesting potential CEACAM6 usage (Fig.S19a - see below). We identified that the three loops that engage with contact with the receptor are the least conserved regions of the predicted RBDs. We have included this sequence alignment as an additional supplemental figure (Fig.S19) and have described this observation in the main results text, as follows:

“Elucidation of the structure of the binding interface enabled us to dissect the genetic determinants of CEACAM6 receptor specificity. Loops 1–3 of the RBDs from alphaCoV species included in our library differ dramatically from CcCoV|KY43 in both length (Fig.S19a) and amino acid composition (Fig.S19b), consistent with only CcCoV|KY43 using CEACAM6 for entry (Fig.S13).”

Legend: **Figure S19. Comparison of CcCoV|KY43 S amino acid sequence with the alphaCoV library.** (a) the RBD of CcCoV|KY43 aligned with the RBD of the S of the other alphaCoV in the library. Loops involved in the interaction with human CEACAM6 are highlighted, showing low conservation. (b) Percentage identity of the alphaCoV library with CcCoV|KY43 S and RBD is shown. btCoV/977, which also uses CEACAM6 as receptor, shares only 61% amino acid identity with CcCoV|KY43.

As discussed above in response to referee 1 comment 3, we have also performed additional mutagenesis showing that single amino acid substitutions in the RBD of the CEACAM6 using virus btRfCoV/LN20 did not confer tropism to the human receptor, suggesting that a more complex contribution of different residues is responsible for the interaction. It is therefore highly unlikely that the more divergent viruses included in our library could engage the CEACAM6 via an interaction analogous to the human CEACAM6+CcCoV|KY43 RBD structure.

With regards sequence determinants for CEACAM6 homologue usage across different mammals, we note that the RBD-binding amino-terminal IgV domains of human CEACAMs 1, 3 and 5 are more closely related to human CEACAM6 than are the equivalent domains from CEACAM6 homologues of new world monkeys (hCEACAM3, 88.3%; hCEACAM1, 86.9%; hCEACAM5, 84.7%; Olive baboon CEACAM6, 79%). This is shown in a sequence alignment of CEACAM6-like proteins (panel below, now part of Fig.S27). We were unable to identify any amino acids, or combinations of amino acids, that are consistently conserved in mammalian CEACAM6-like proteins that support CcCoV|KY43 and CcCoV|2A/B entry but altered in those that do not. We have added a sentence to the results, as follows:

“No consistent amino acid substitution pattern at the RBD interface distinguishes CEACAM6-like proteins that permit CcCoV entry from those that do not.”

Taken together, we concur with the referee that receptor usage cannot be inferred based on the sequence alone for the highly divergent RBD and CEACAM6 sequences presented in this study.

Human	35	KLT IESTP FNVAEGKEVLLLAHNLPQNRIGYSWYKGERVDGNLSLIVGYVI GTQQTTPGPAYSGRETIYPNASLLIQNVTONDTGFYTLQVIKSDLVNEEATGQFHVYP
Sumatran orangutan	35	QLT IESTP FNVAEGKEVLLLAHNLSQNRIGYIWKGERVDANRLIVAYKIETQQTTPGPAYSGRETIYPNASLLIQNVTONDTGFYTLQVIKSDLVNEEATGQFHVYP
Gibbon	35	QLT IESTP FNVAEGKEVLLLAHNLPQNRIGYSWYKGERVDGNRLIVAYAIETAQTTPGPAYSGRETIYPNASLLIQNVTONDTGSYTLQVIKSDLVNEEATGQFRVYP
Olive baboon	35	QLTVESRPFNVAEGKEVLLLAHNLPQNTLGFNWKGERVDAKRLIVAYVI GTNQTTPGPAHSGRETIYPNASLLIQNVTONDTGSYTLQAIKGNLVTEEATGRFWVYP
Macaque	35	QLT IESRPFNVAEGKEVLLLAHNLPQNTLGFNWKGERVDAKRLIVAYVIETQQTTPGPAHSGRETIYPNASLLIQNVTONDTGSYTLQVIKSDLVNEEATGRFWVYP
Tufted capuchin	35	QLT IESMPNAAEGKEVLLLAHNLPQNIAGFNWYKQSDVGNRRI IGYVIATQLTTPGPAYSGRETIYPNASLLIQNVTLNDTGFYTLQVIKADLVNEEAAGQFRVYP
Pale spear-nosed bat	35	--RLTVASTNAAEGKDVLLLVNLPESELLTYNWFGRKSAANSRRIGAFVTTQTQKFTPGPAHSGRETIYPNGSLLFPQVTLNDTGYTTIAI I KKDGLTEEATGQLRVYP
Greater horseshoe bat	35	QLALEFMPNAAEWEDVLELAHNLPEDLAGYAWFKGGRVDSRQIASYKIDTGVNPPGPAYSGRETIYPNGSLLFPQVTLNEDTGYTLQAIKTNFHNNEEVTGQLRVYP
Great roundleaf bat	35	-----MPPSAAEGKDVLELAHNLPEDLAGYAWFKGGRVDSRQIASYRIDTGVNPPGPTYSGRETIYPNGSLLVQVTLNEDTGYTLQAIKTHFHNNEEVTGQLRVYP
Egyptian fruit bat	95	QLTLEILPNSAAEGKDVLLLVHNLPELAGYSWYKASVDSNHLIESYVIDSQTNVPGPAYSGRETIYPNGSLLIQNVTLNEDTGYTLQAIKKNLLNEQVTLGQLRVYA
Indian flying fox	35	QITLESPLNSAAEGKDVLLLVHNLVNMGTGYAWYKASVDSNRLIASYVIDSQESI PGPAHSGRETIYPNGSLLIQVTLNEDTGYTLQAIKKNFLNEEVTGQLRVYP
Brandt's bat	35	QLT IESVPPNAAEGKDVLLRVHNLPGNLGGYIWKGERVDNSHKIVSYVIDTQKITLGPAYSGRETIYPNGSLLFQNLTKLDTGYTLQAIKDFQSKQVTLGQLRVYL
Yuma myotis	35	QLT IESVPPNAAEGKDVLLRVHNLPGNLFGYAWYKGEIVDTSIQI I SYVIETQATLGPESSGREKIYPNGSLLFQNLTKLDTGYTLQATGKDLQNKQVTLGQLRVYP
Daubenton's bat	35	QLT IESVPPNAAEGKDVLLRVHNLPGDLVGYAWYKGETVDSDLRIVSYVIDTQKTTYGPAYSDREKIYPNGSLLFQNVTLKDTGYTLHATEKKNFQSKQVTLGQLRVYL
Big brown bat	35	QLT IESVPPDAAGKDVLLRVHNLPGNLVGYAWYKGETVDSNHLISYIIDTQKITFGPAYSDREKIYPNGSLLFPQVTLNEDTGYTLVATDKDFKDKQVTLGQLRVHP
Natal long-fingered bat	35	QPRVESVPPNAAEGKDVLLRVHSLPGLIGYAWYKGESEDSNRKIASYVIDTQIVSHGPAYSGRETIYPNGSLLFPQVTLNEDTGYTLVLAIKKFKSEEGTQGFHVYA
Cat	35	QVTVESVPPNAAEGKDVLLRVHNLPGNLIGFGWFKGTTIDPREIIVSYAADSQEI TLGFAHSGRETIYPNGSLLFQNLISLEDTGYTLQAIKRNHVHVERVIGQLRVYP

Legend: **Figure S27.** On the left, sequences of the IgV domain of CEACAM6 included in our library is shown. Domains responsible for the binding to CcCoV|KY43 are highlighted in teal, while residues in contact with the CcCoV RBDs are underlined.

5) As shown by the PDB validation reports both refined structures have a relatively high Rfree and a relatively large number of RSRZ outliers - values that can indicate problems with structures (CcCoV-2B is better). In this case, these values are almost certainly the result of the fact that both crystals diffract anisotropically. The authors have taken appropriate steps to minimize the negative impacts and the electron density at the RBD-CEACAM6 interface supports the reported models and the conclusions made regarding the interaction. Figure 4h would be improved by showing a comparison of the two interfaces, not just a reference to Figure 3b. I think that the use of stereo pairs in Figure S11 should be reconsidered. It is important to show the density and a detailed comparison of the two complexes, but many readers do not have the required viewer.

As suggested, we have updated Figure 4h to aid direct comparison of the interaction interfaces by showing a superposition of the RBD loops 1-3 from CcCoV|KY43 and 2B (see opposite).

Legend: Figure 4h. Zoom-in of key residues at the CEACAM6-binding interfaces of CcCoV|2B (dark brown) and CcCoV|KY43 (light brown), highlighting the similarity of the interactions.

With regards the stereo diagrams, we have included them in the supplement because it was our understanding that Nature family journals require this for all new structures (see point 11 of <https://www.nature.com/nsmb/submission-guidelines/aip-and-formatting>). We are happy to simplify this figure by including a single image instead of the stereo pair at the Editor's discretion.

6) The methods section mentions BLI but only ITC data is described in the main.

Bio-layer interferometry (BLI) data is included in Figure S21, and is now expanded to include measurements of both CcCoV|KY43 and CcCoV|2B RBD in response to referee 1 comment 12. These measurements confirm the pseudotype, ELISA and ITC data presented in the main text, including the loss up binding upon CEACAM6 I63F mutation. We thus believe it is appropriate to retain these methods in the main text.

References:

1. Dufloo, J. *et al.* Dipeptidase 1 is a functional receptor for a porcine coronavirus. *Nat Microbiol* **10**, 2981-2996 (2025).
2. Williams, R.K., Jiang, G.S. & Holmes, K.V. Receptor for mouse hepatitis virus is a member of the carcinoembryonic antigen family of glycoproteins. *Proc Natl Acad Sci U S A* **88**, 5533-5536 (1991).
3. Peng, G. *et al.* Crystal structure of mouse coronavirus receptor-binding domain complexed with its murine receptor. *Proc Natl Acad Sci U S A* **108**, 10696-10701 (2011).
4. Erlendsson, S. & Teilum, K. Binding Revisited-Avidity in Cellular Function and Signaling. *Front Mol Biosci* **7**, 615565 (2020).
5. Brandenburg, O.F., Magnus, C., Rusert, P., Regoes, R.R. & Trkola, A. Different infectivity of HIV-1 strains is linked to number of envelope trimers required for entry. *PLoS Pathog* **11**, e1004595 (2015).
6. Xiao, T. *et al.* A trimeric human angiotensin-converting enzyme 2 as an anti-SARS-CoV-2 agent. *Nat Struct Mol Biol* **28**, 202-209 (2021).
7. Bushell, K.M., Söllner, C., Schuster-Boeckler, B., Bateman, A. & Wright, G.J. Large-scale screening for novel low-affinity extracellular protein interactions. *Genome Res* **18**, 622-630 (2008).
8. Shilts, J. *et al.* A physical wiring diagram for the human immune system. *Nature* **608**, 397-404 (2022).
9. Davis, S.J. *et al.* The nature of molecular recognition by T cells. *Nat Immunol* **4**, 217-224 (2003).
10. Bianchi, E., Doe, B., Goulding, D. & Wright, G.J. Juno is the egg Izumo receptor and is essential for mammalian fertilization. *Nature* **508**, 483-487 (2014).

11. Maginnis, M.S. Virus-Receptor Interactions: The Key to Cellular Invasion. *J Mol Biol* **430**, 2590-2611 (2018).
12. Daniel, P.F. Conservation evaluation and phylogenetic diversity. *Biological Conservation* **61**, 1-10 (1992).
13. Dermody, T.S., Kirchner, E., Guglielmi, K.M. & Stehle, T. Immunoglobulin superfamily virus receptors and the evolution of adaptive immunity. *PLoS Pathog* **5**, e1000481 (2009).
14. Dveksler, G.S. *et al.* Cloning of the mouse hepatitis virus (MHV) receptor: expression in human and hamster cell lines confers susceptibility to MHV. *J Virol* **65**, 6881-6891 (1991).
15. Klaile, E. *et al.* Binding of *Candida albicans* to Human CEACAM1 and CEACAM6 Modulates the Inflammatory Response of Intestinal Epithelial Cells. *mBio* **8** (2017).
16. Tchoupa, A.K., Schuhmacher, T. & Hauck, C.R. Signaling by epithelial members of the CEACAM family - mucosal docking sites for pathogenic bacteria. *Cell Commun Signal* **12**, 27 (2014).
17. Hofmann, H. *et al.* Human coronavirus NL63 employs the severe acute respiratory syndrome coronavirus receptor for cellular entry. *Proc Natl Acad Sci U S A* **102**, 7988-7993 (2005).
18. Raj, V.S. *et al.* Dipeptidyl peptidase 4 is a functional receptor for the emerging human coronavirus-EMC. *Nature* **495**, 251-254 (2013).
19. Huang, X. *et al.* Human Coronavirus HKU1 Spike Protein Uses O-Acetylated Sialic Acid as an Attachment Receptor Determinant and Employs Hemagglutinin-Esterase Protein as a Receptor-Destroying Enzyme. *J Virol* **89**, 7202-7213 (2015).
20. Vlasak, R., Luytjes, W., Spaan, W. & Palese, P. Human and bovine coronaviruses recognize sialic acid-containing receptors similar to those of influenza C viruses. *Proc Natl Acad Sci U S A* **85**, 4526-4529 (1988).
21. Saunders, N. *et al.* TMPRSS2 is a functional receptor for human coronavirus HKU1. *Nature* **624**, 207-214 (2023).
22. An, D. *et al.* Resolving the APN controversy in PEDV infection: Comparative kinetic characterization through single-virus tracking. *PLoS Pathog* **21**, e1013317 (2025).
23. Li, W. *et al.* Aminopeptidase N is not required for porcine epidemic diarrhea virus cell entry. *Virus Res* **235**, 6-13 (2017).
24. Shirato, K. *et al.* Porcine aminopeptidase N is not a cellular receptor of porcine epidemic diarrhea virus, but promotes its infectivity via aminopeptidase activity. *J Gen Virol* **97**, 2528-2539 (2016).
25. Luo, L. *et al.* Aminopeptidase N-null neonatal piglets are protected from transmissible gastroenteritis virus but not porcine epidemic diarrhea virus. *Sci Rep* **9**, 13186 (2019).
26. Minh, B.Q. *et al.* IQ-TREE 2: New Models and Efficient Methods for Phylogenetic Inference in the Genomic Era. *Mol Biol Evol* **37**, 1530-1534 (2020).
27. Hoang, D.T., Chernomor, O., von Haeseler, A., Minh, B.Q. & Vinh, L.S. UFBoot2: Improving the Ultrafast Bootstrap Approximation. *Mol Biol Evol* **35**, 518-522 (2018).
28. Aiewsakun, P. & Katzourakis, A. Time-Dependent Rate Phenomenon in Viruses. *J Virol* **90**, 7184-7195 (2016).
29. Wertheim, J.O., Chu, D.K., Peiris, J.S., Kosakovsky Pond, S.L. & Poon, L.L. A case for the ancient origin of coronaviruses. *J Virol* **87**, 7039-7045 (2013).
30. Mifsud, J.C.O., Suchard, M.A., Holmes, E.C. & Lemey, P. Recent advances in the inference of deep viral evolutionary history. *J Virol* **99**, e0029225 (2025).

2025-07-18615 Gallo et al., Response to reviewers

Dear Editors and Reviewers,

Again, we would like to thank you for the time taken to read our revised article, including the additional data and analysis we provided. In addition, thank you for your positive comments on the quality of our work and for recognizing the novelty of our study and its importance in improving pandemic preparedness and public health.

Yours sincerely,

Dr. Dalan Bailey

On behalf of all co-authors 02/02/2026

Referee #1 (Remarks to the Author):

The authors have undertaken substantial additional work since the previous submission, and the revised manuscript represents a clear and significant improvement. The new data and explanations address my major concerns and convincingly strengthen the central conclusions:

(1) CEACAM6 functions as a bona fide receptor for CcCoV-related strains with global distribution, which is both evolutionarily and physiologically plausible; and (2) most alphacoronaviruses do not rely on ACE2 or APN for entry.

This study has important implications for coronavirus receptor biology, cross-species transmission, and the identification of potential vulnerabilities relevant to preparedness for future epidemics. The discussion integrating these findings into coronavirus receptor evolution is thoughtful and compelling. Overall, this work represents a substantial advance, and the manuscript is close to being suitable for publication, pending minor revisions.

We are glad the additional experiments have properly answered the reviewer's queries and questions.

Minor comments

1.Line 76: Consider adding “significantly” or “markedly” before “the richness and heterogeneity” to improve accuracy.

The term “significantly” has been added to the text.

2.Line 98: Please double-check why feline CoV could not be pseudotyped, as it works well in our hands and in published studies.

We have repeated these experiments multiple times, and repeatedly we do not observe good incorporation of the feline CoV Spike. We followed our rationale to replace “misbehaving” Spikes (Supplementary Fig.21), but we could not find any pseudotyping S reported in the literature that was at least 95% identical to the one selected by the algorithm.

3.Line 126: I recommend removing the word “spike” from the title.

The title has been amended.

4.Figure 5: Ensure consistent nomenclature between “YN20” and “YN2020.”

This has been corrected, thank you.

5. Figure S10: Please specify the Y-axis values (between 1 and 10). Given the modest fold change in luciferase activity, it would be helpful to also show representative GFP images.

The figure has been modified to include Y-axis values.

6. Figure S12: All bars, including the siRNA groups, appear close to 100%. Please double-check the data and presentation.

Thank you for raising the point. Indeed, we double-checked the analysis and there was a mistake in the normalization. We corrected the figure now.

Supplementary Fig.7. Transient knock-down of human CEACAM6 using siRNA does not affect hCoV/229E entry. One day after electroporation with different siRNA against human CEACAM6, Caco2 and Calu3 cells (the same used in the experiments described in Figure 2i) were tested for their permissivity to hCoV/229E-inf1, showing no phenotypic effect on entry. Technical triplicates from one experiment are shown.

Referee #2 (Remarks to the Author):

The authors have gone to great lengths to address our concerns, as well as those of the other reviewers. The new experimental data and textual revisions have substantially strengthened the conclusions, and we have no further comments. We believe this manuscript represents a welcome addition to the coronavirus literature.

Thank you.

Referee #3 (Remarks to the Author):

I co-reviewed this manuscript with one of the reviewers who provided the listed reports.

Thank you.

Referee #4 (Remarks to the Author):

The authors have adequately addressed my comments, and I would recommend the manuscript for publication. My only additional suggestion is to include the discrete-trait reconstruction of CEACAM6 usage provided in the rebuttal as a supplementary figure, as it makes clear the point regarding multiple emergence described in the text.

Thank you for your comments. The discrete-trait reconstruction tree has been added as Supplementary Fig.18.

Referee #5 (Remarks to the Author):

I have read the revised manuscript and the rebuttal and feel that the authors have done a very good job at addressing reviewer comments. My concerns have been addressed, and I would also note that the addition of the RBD binding data (response to reviewer 2) is a significant addition. This data

certainly helps to support the suggestion that many of the tested viruses do not use APN or ACE2 as receptor. I would recommend that the paper now be published.

Thank you for your comments. Of note, we have included the RBD binding data in Extended Data Fig.3, to highlight its importance.

Additional editorial requests:

1. Please reduce the length of the title to 75 characters (with spaces) or less, so that it fits on two lines in the final layout.

The title has been shortened to "Heart-nosed bat alphacoronaviruses use human CEACAM6 to enter cells".

2. Please add references to the abstract (if applicable).

Not applicable.

3. Please reduce the Abstract to 230 words or less. Currently there are 281 words.

The abstract has been shortened to 216 words.

4. The number of references should generally not exceed 60. Currently you have 83 references.

5. Please create a separate reference list for any methods references, making sure that the numbering continues from the main text references.

We have created a separate paragraph for the references included in the Method section. The main text contains 49 references, while 34 references are included in the additional section.

6. Please remove the main figures from the article file and re-supply them individually in an acceptable format such as EPS, AI, PS, PDF, PPT, PSD or XLS (for graphs) with editable vector files.

Main figures have been removed, and are provided as individual AI files.

7. Please ensure that the text size in all figures is at least 5 pt Arial.

This has been checked.

8. Please note that the legends for the main figures should not exceed 300 words. If it is not possible to reduce the length accordingly, please ensure that the final legends are as close as possible to 300 words.

All the legends are less than 300 words, with the exception of Figure 2. The diversity of experiments used and detailed methodology required does not allow to reduce the number of words any further.

9. Please reduce subheadings to 40 characters (with spaces) or less.

This has been done for the main text.

10. Please provide a supplementary information guide as a separate word document.

The guide has been created, and it includes all the supplementary figures and tables.

11. There are potential third party rights issues in the figures. Please check the sources of all illustrations and clarify whether permissions are needed to adapt or reproduce them. Please make sure to include the relevant details in third party rights table when you resubmit (more information below). If Biorender or a similar software has been used, please also ensure to provide relevant licenses. In particular please check: Figures 1, 2(a,j), 4(a,b), 5a and Supplementary Figures 31, 33.
12. Please make sure to provide a third party rights table (more information below) when you resubmit. If Biorender or similar software has been used, please also ensure to provide relevant licenses.

The third party rights document, with all the relevant information, has been completed.

13. Please provide an author contributions statement in the main text of the manuscript.

The statement has been included at the end of the main text.

14. There are 33 display items in the SI and no Extended Data. Please see the descriptions of Extended Data versus SI below. Data figures that are important for the main message of the paper should be Extended Data. We usually limit ED figures to 10, but in this case we can grant two extra (12 total). ED figures can be mutli-panelled as long as each figure fits on one page. Please make the most important SI figures ED figures. Any remaining figures that cannot fit in the 12 ED items can stay in the SI.

12 ED figures have been prepared from the 33 items, and are uploaded as individual .tiff images.

15. Please make sure that all data and weblinks are available.

This has been verified.

16. Please see the attachments for further guidance on formatting your paper and figures.

The two attached documents have been completed according to instructions.